# Optimizing Model-Agnostic Random Subspace Ensembles

## Abstract

This paper presents a model-agnostic ensemble approach for supervised learning. The proposed approach is based on a parametric version of Random Subspace, in which each base model is learned from a feature subset sampled according to a Bernoulli distribution. Parameter optimization is performed using gradient descent and is rendered tractable by using an importance sampling approach that circumvents frequent re-training of the base models after each gradient descent step. The degree of randomization in our parametric Random Subspace is thus automatically tuned through the optimization of the feature selection probabilities. This is an advantage over the standard Random Subspace approach, where the degree of randomization is controlled by a hyper-parameter. Furthermore, the optimized feature selection probabilities can be interpreted as feature importance scores. Our algorithm can also easily incorporate any differentiable regularization term to impose constraints on these importance scores.

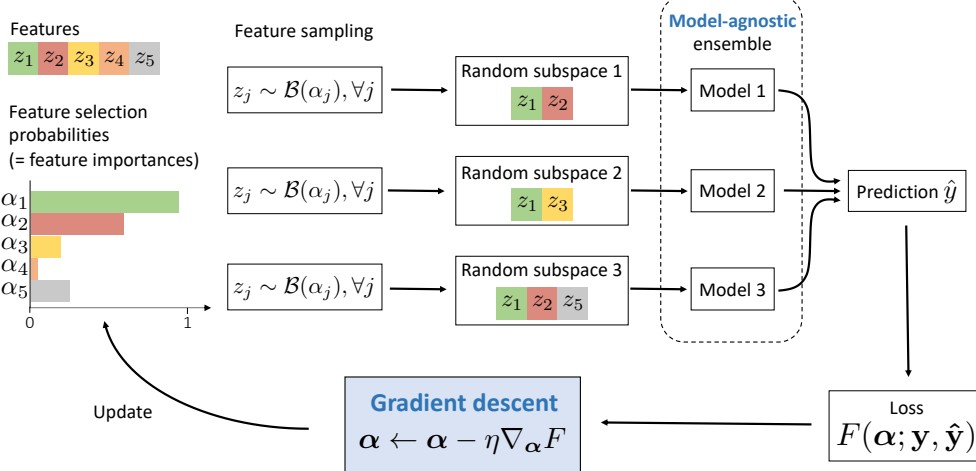

Figure 1: **Parametric Random Subspace (PRS).** A model-agnostic ensemble is built, in which each base model is learned from a feature subset sampled according to a Bernoulli distribution with parameters $\boldsymbol{\alpha}$. The training procedure consists in identifying the parameters $\boldsymbol{\alpha}$ that minimize any loss $F(\boldsymbol{\alpha}; \mathbf{y}, \hat{\mathbf{y}})$ (which may include a regularization term over $\boldsymbol{\alpha}$). This optimization problem is solved using gradient descent and importance sampling.

## 1 Introduction

In supervised learning, ensemble approaches are popular techniques to improve the performance of any learning algorithm. The most prominent ensemble methods include averaging ensembles like Bagging (Breiman, 1996b), Random Subspace (Ho, 1998), or Random Forest (Breiman, 2001), as well as boosting ensembles, such as Adaboost (Freund & Schapire, 1997) or gradient boosting (Friedman, 2001). Both Random Forest

and Random Subspace aggregate the predictions of base models that are randomized through a random feature selection mechanism (with additional sample bootstrapping, similarly as in Bagging, in the case of Random Forest). However, while Random Subspace is a model-agnostic approach, i.e. an approach that can be combined with any type of base models, Random Forest is designed specifically for the aggregation of decision trees. Indeed, in Random Subspace, feature randomization occurs at the level of the base model, before training the latter, and can thus be combined with any base model. On the other hand, the feature randomization used in Random Forest is designed specifically for decision trees: it occurs at the level of the tree node, where a feature subset is randomly sampled before selecting the best split. Note that while boosting is a model-agnostic approach, it is designed to aggregate weak models, and is hence typically used with shallow decision trees.

One advantage of decision trees is their interpretability. Their node splitting strategy is akin to an embedded feature selection mechanism that makes them robust to irrelevant features and feature importance scores can be furthermore easily derived from a trained tree model to quantitatively assess the selected features (Breiman et al., 1984b). These characteristics carry over when decision trees are used as based learners with the aforementioned ensemble methods, as importance scores can be averaged over all the trees in the ensemble, which furthermore increases their stability. This arguably has participated to the popularity of tree-based ensemble methods for the prediction of tabular data Grinsztajn et al. (2022). On the other hand, this interpretability through feature selection and ranking is obviously lost when model-agnostic ensemble methods are applied with other base learners that are not inherently interpretable.

In this paper, we propose a novel ensemble approach for supervised learning (Figure 1) that is fully model-agnostic, i.e., makes no assumption about the nature of the base models, and naturally embeds a feature selection mechanism and provides feature importance scores, irrespectively of the choice of the base model. The proposed approach is based on a parametric version of Random Subspace (denoted PRS), in which each base model is learned from a feature subset sampled according to a Bernoulli distribution. We formulate the training procedure as an optimization problem where the goal is to identify the parameters of the Bernoulli distribution that minimize the generalization error of the ensemble model, and we show that this optimization problem can be solved using gradient descent even when the base models are not differentiable. The optimization of the Bernoulli distribution is however intractable, as the computation of the exact output of the full ensemble model would require the training of one model for each possible feature subset. To render the parameter optimization tractable, we use Monte Carlo sampling to approximate the ensemble model output. We further use an importance sampling approach that circumvents frequent re-training of the base models after each update of the gradient descent.

The degree of randomization in our parametric Random Subspace is automatically tuned through the optimization of the feature selection probabilities. This is an advantage over the standard Random Subspace approach, where the degree of randomization is controlled by a hyper-parameter. Furthermore, the optimized feature selection probabilities can be interpreted as feature importance scores. Our algorithm can also easily incorporate any differentiable regularization term to impose constraints on these importance scores. We show the good performance of the proposed approach, both in terms of prediction and feature ranking, on simulated and real-world datasets. We also show that PRS can be successfully used for the reconstruction of gene regulatory networks.

## 2 Methods

We assume a standard supervised learning setting, where we have at our disposal a learning set containing $N$ input-output pairs $\{(\mathbf{x}_i, y_i)\}_{i=1}^N$ drawn from an unknown probability distribution. Let us denote by $M$ the number of input variables. The output $y$ can be either continuous (regression problem) or discrete (classification problem). Our goal is to train a model-agnostic predictive model, while deriving for each input variable a score that measures its importance for the output prediction.

To achieve this goal, we build upon the Random Subspace approach (RS, Ho, 1998). RS consists in learning an ensemble of predictive models, where each model is built from a randomly chosen subset of $K$ input variables (with $K < M$), sampled according to a uniform distribution. Here, instead of using a uniform distribution, we adopt a parametric distribution for the selection of the input features, and feature importance scores

are derived through the identification of the distribution parameters that yield the lowest generalization error. In the following, after introducing the parametric RS model (Section 2.1), we show how this model can be trained in a tractable way (Section 2.2) and we discuss our approach with respect to related works (Section 2.3).

## 2.1 The Parametric Random Subspace approach (PRS)

Let us denote by $\mathbf{z} = (z_1, \ldots, z_M)^\top \in \{0, 1\}^M$ a binary vector of length $M$ encoding a subset of selected input variables: $z_j = 1$ if the $j$-th variable is selected and $z_j = 0$ otherwise, $\forall j \in \{1, \ldots, M\}$. In the proposed PRS approach, each indicator variable $z_j$ is assumed to follow a Bernoulli distribution with parameter $\alpha_j$. The probability mass function for $\mathbf{z}$ is then given by:

$$p(\mathbf{z}|\boldsymbol{\alpha}) = \prod_{j=1}^{M} \alpha_j^{z_j} (1 - \alpha_j)^{(1-z_j)}, \tag{1}$$

where $\alpha_j \in [0, 1]$ is the probability of selecting the $j$-th variable and $\boldsymbol{\alpha} = (\alpha_1, \ldots, \alpha_M)^\top$. Let $\mathcal{Z} = \{\mathbf{z}^1, \mathbf{z}^2, \ldots, \mathbf{z}^{|\mathcal{Z}|}\}$ be the set of all the possible feature subsets, where $|\mathcal{Z}| = 2^M$ is the cardinality of $\mathcal{Z}$.

We assume an ensemble method that consists in averaging base models trained independently of each other using subsets of features drawn from $p(\mathbf{z}|\boldsymbol{\alpha})$. Let us denote by $\mathcal{F}$ some functional space corresponding to a given learning algorithm and by $\mathcal{F}_\mathbf{z} \subseteq \mathcal{F}$ the subset of functions from $\mathcal{F}$ that only depend on the variables indicated by $\mathbf{z}$. Let $f_{\mathbf{z}^t} \in \mathcal{F}_{\mathbf{z}^t}$ be the base model learned by this learning algorithm from the feature subset $\mathbf{z}^t$ ($\forall t \in \{1, \ldots, |\mathcal{Z}|\}$). Asymptotically, the prediction of the ensemble model for a given input $\mathbf{x}$ is given by:

$$\mathbb{E}[f_\mathbf{z}(\mathbf{x})]_{p(\mathbf{z}|\boldsymbol{\alpha})} = \sum_{t=1}^{|\mathcal{Z}|} p(\mathbf{z}^t|\boldsymbol{\alpha}) f_{\mathbf{z}^t}(\mathbf{x}). \tag{2}$$

For a fixed $\boldsymbol{\alpha}$, a practical approximation of $\mathbb{E}[f_\mathbf{z}(\mathbf{x})]_{p(\mathbf{z}|\boldsymbol{\alpha})}$ can be obtained by Monte-Carlo sampling, i.e. by drawing $T$ feature subsets from $p(\mathbf{z}|\boldsymbol{\alpha})$ and then training a model from each of these subsets, using the chosen learning algorithm (Figure 1). If all the $\alpha_j$'s are equal, the resulting ensemble method is very close to the standard RS approach, the only difference being that the number of selected features will be slightly randomized from one model to the next. In this work, we would like however to identify the parameters $\boldsymbol{\alpha}$ that yield the most accurate expected predictions $\mathbb{E}[f_\mathbf{z}(\mathbf{x})]_{p(\mathbf{z}|\boldsymbol{\alpha})}$ over our training set. Given a loss function $L$, the corresponding optimization problem can be formulated as follows:

$$\min_{\boldsymbol{\alpha} \in [0,1]^M} F(\boldsymbol{\alpha}),$$
$$\text{where } F(\boldsymbol{\alpha}) = \frac{1}{N} \sum_{i=1}^{N} L\left(y_i, \mathbb{E}[f_\mathbf{z}(\mathbf{x}_i)]_{p(\mathbf{z}|\boldsymbol{\alpha})}\right). \tag{3}$$

A nice advantage is that the selection probabilities $\boldsymbol{\alpha}$ after optimization can be interpreted as measures of variable importances: useless variables are expected to get low selection probabilities, while the most important ones are expected to get selection probabilities close to 1.

## 2.2 Training the PRS model

We propose to solve the optimization problem in Eq (3) using gradient descent. More specifically, since $\alpha_j$ must be between 0 and 1, $\forall j$, we use the projected gradient descent technique, where $\boldsymbol{\alpha}$ is projected into the space $[0, 1]^M$ after each step of the gradient descent. In the following, we first derive the analytical formulation of the gradient of the objective function. We then explain how to estimate this gradient by using Monte Carlo sampling and show how to incrementally update this gradient estimate using importance sampling. Precise pseudo-code of the algorithm is given in Appendix A and our Python implementation is available in the supplementary material[1].

---

[1] Our Python code and the datasets used in this paper will be available on GitHub, should the paper be accepted.

### 2.2.1 Computing the gradient

Assuming that the loss function $L$ is differentiable, the gradient of the objective function $F(\boldsymbol{\alpha})$ w.r.t. $\boldsymbol{\alpha}$ is:

$$\nabla_{\boldsymbol{\alpha}} F(\boldsymbol{\alpha}) = \frac{1}{N} \sum_{i=1}^{N} \frac{\mathrm{d}L}{\mathrm{d}\mathbb{E}[f_{\mathbf{z}}(\mathbf{x}_i)]_{p(\mathbf{z}|\boldsymbol{\alpha})}} \nabla_{\boldsymbol{\alpha}} \mathbb{E}[f_{\mathbf{z}}(\mathbf{x}_i)]_{p(\mathbf{z}|\boldsymbol{\alpha})}. \tag{4}$$

To compute the gradient $\nabla_{\boldsymbol{\alpha}} \mathbb{E}[f_{\mathbf{z}}(\mathbf{x}_i)]_{p(\mathbf{z}|\boldsymbol{\alpha})}$, we resort to the *score function* approach (Rubinstein & Shapiro, 1993), also known as the *REINFORCE* method (Williams, 1992) or the *likelihood-ratio* method (Glynn, 1990), which allows us to express the gradient of an expectation as an expectation itself (see Appendix B):

$$\nabla_{\boldsymbol{\alpha}} \mathbb{E}[f_{\mathbf{z}}(\mathbf{x}_i)]_{p(\mathbf{z}|\boldsymbol{\alpha})} = \mathbb{E}\left[f_{\mathbf{z}}(\mathbf{x}_i) \nabla_{\boldsymbol{\alpha}} \log p(\mathbf{z}|\boldsymbol{\alpha})\right]_{p(\mathbf{z}|\boldsymbol{\alpha})}. \tag{5}$$

A major advantage of the score function approach is that, in order to compute the gradient in Eq. (5), only the distribution $p(\mathbf{z}|\boldsymbol{\alpha})$ needs to be differentiable, and not the base model $f_{\mathbf{z}}$. By using the score function method with the Bernoulli distribution in Eq. (1), the $j$-th component of the gradient is given by (see Appendix B):

$$\frac{\partial \mathbb{E}[f_{\mathbf{z}}(\mathbf{x}_i)]_{p(\mathbf{z}|\boldsymbol{\alpha})}}{\partial \alpha_j} = f_{j,1}^{\boldsymbol{\alpha}}(\mathbf{x}_i) - f_{j,0}^{\boldsymbol{\alpha}}(\mathbf{x}_i) \tag{6}$$

where, for the simplicity of notations, we have defined:

$$f_{j,0}^{\boldsymbol{\alpha}}(\mathbf{x}_i) = \mathbb{E}[f_{\mathbf{z}}(\mathbf{x}_i)|z_j = 0]_{p(\mathbf{z}_{-j}|\boldsymbol{\alpha}_{-j})}, \tag{7}$$

$$f_{j,1}^{\boldsymbol{\alpha}}(\mathbf{x}_i) = \mathbb{E}[f_{\mathbf{z}}(\mathbf{x}_i)|z_j = 1]_{p(\mathbf{z}_{-j}|\boldsymbol{\alpha}_{-j})}, \tag{8}$$

with $\mathbf{z}_{-j} = \mathbf{z} \backslash z_j$, $\boldsymbol{\alpha}_{-j} = \boldsymbol{\alpha} \backslash \alpha_j$. $f_{j,0}^{\boldsymbol{\alpha}}$ (resp. $f_{j,1}^{\boldsymbol{\alpha}}$) is thus the expected output of a model that does not take (resp. takes) as input the $j$-th variable. We thus finally have:

$$\frac{\partial F}{\partial \alpha_j} = \frac{1}{N} \sum_{i=1}^{N} \frac{\mathrm{d}L}{\mathrm{d}\mathbb{E}[f_{\mathbf{z}}(\mathbf{x}_i)]_{p(\mathbf{z}|\boldsymbol{\alpha})}} \left(f_{j,1}^{\boldsymbol{\alpha}}(\mathbf{x}_i) - f_{j,0}^{\boldsymbol{\alpha}}(\mathbf{x}_i)\right). \tag{9}$$

The above derivative can be easily interpreted in the context of a gradient descent approach. For example, when $\frac{\mathrm{d}L}{\mathrm{d}\mathbb{E}[f_{\mathbf{z}}(\mathbf{x}_i)]_{p(\mathbf{z}|\boldsymbol{\alpha})}}$ is positive, the loss $L$ decreases with a lower model prediction $\mathbb{E}[f_{\mathbf{z}}(\mathbf{x}_i)]_{p(\mathbf{z}|\boldsymbol{\alpha})}$. This means that if $f_{j,0}^{\boldsymbol{\alpha}}(\mathbf{x}_i) < f_{j,1}^{\boldsymbol{\alpha}}(\mathbf{x}_i)$, the model without variable $j$ will give a lower loss than the model with variable $j$. In that case, the derivative $\frac{\partial F}{\partial \alpha_j}$ is positive and a gradient descent step (i.e. $\alpha_j \leftarrow \alpha_j - \eta \frac{\partial F}{\partial \alpha_j}$, where $\eta$ is the learning rate) will decrease the value of $\alpha_j$.

### 2.2.2 Estimating the gradient

Given the current selection probabilities $\boldsymbol{\alpha}$, the exact computation of the expectation in Eq. (5) is obviously intractable as it requires training $|\mathcal{Z}|$ models. An unbiased estimation can be obtained by Monte Carlo sampling, i.e. by averaging over $T$ subsets of features $\mathbf{z}^{(t)}$ sampled from $p(\mathbf{z}|\boldsymbol{\alpha})$:

$$\mathbb{E}\left[f_{\mathbf{z}}(\mathbf{x}_i) \nabla_{\boldsymbol{\alpha}} \log p(\mathbf{z}|\boldsymbol{\alpha})\right]_{p(\mathbf{z}|\boldsymbol{\alpha})} \simeq \frac{1}{T} \sum_{t=1}^{T} f_{\mathbf{z}^{(t)}}(\mathbf{x}_i) \nabla_{\boldsymbol{\alpha}} \log p(\mathbf{z}^{(t)}|\boldsymbol{\alpha}), \tag{10}$$

where $f_{\mathbf{z}^{(t)}}$ is the model trained using only as inputs the features in the subset $\mathbf{z}^{(t)}$.

It remains to be explained on which data the models $f_{\mathbf{z}^{(t)}}$ are trained. Using the same $N$ samples as the ones used to compute the gradient in Eq. (4) would lead to biased predictions $f_{\mathbf{z}}(\mathbf{x}_i)$ and hence to overfitting. We thus use a batch gradient descent approach, in which a subset of the training dataset (e.g. 10% of the samples) are used for computing the gradient, while the remaining samples are used for training the base models. Note that in the case where $\mathbf{z}^{(t)}$ is the empty set, which can happen when all the $\alpha_j$ parameters are very low, we set $f_{\mathbf{z}^{(t)}}$ to a constant model that always returns the mean value of the output in the training set (for regression problems) or the majority class (for classification problems).

Although the gradient estimator in Eq. (10) is unbiased, it is known to suffer from high variance, which can make the gradient descent optimization very unstable. One common solution to reduce this variance is to use the fact that, for any constant $b$, we have (see Appendix C.1):

$$\mathbb{E}\left[f_{\mathbf{z}}(\mathbf{x}_i)\nabla_{\boldsymbol{\alpha}}\log p(\mathbf{z}|\boldsymbol{\alpha})\right]_{p(\mathbf{z}|\boldsymbol{\alpha})} = \mathbb{E}\left[(f_{\mathbf{z}}(\mathbf{x}_i) - b)\nabla_{\boldsymbol{\alpha}}\log p(\mathbf{z}|\boldsymbol{\alpha})\right]_{p(\mathbf{z}|\boldsymbol{\alpha})}. \tag{11}$$

The constant $b$ is called a *baseline* and its value will affect the variance of the estimator. In our approach, we use the optimal value of $b$, i.e. the value that minimizes the variance, which is (see Appendix C.2):

$$b = \frac{\mathbb{E}[(\nabla_{\boldsymbol{\alpha}}\log p(\mathbf{z}|\boldsymbol{\alpha}))^2 f_{\mathbf{z}}(\mathbf{x}_i)]_{p(\mathbf{z}|\boldsymbol{\alpha})}}{\mathbb{E}[(\nabla_{\boldsymbol{\alpha}}\log p(\mathbf{z}|\boldsymbol{\alpha}))^2]_{p(\mathbf{z}|\boldsymbol{\alpha})}} \simeq \frac{\sum_{t=1}^{T}(\nabla_{\boldsymbol{\alpha}}\log p(\mathbf{z}^{(t)}|\boldsymbol{\alpha}))^2 f_{\mathbf{z}^{(t)}}(\mathbf{x}_i)}{\sum_{t=1}^{T}(\nabla_{\boldsymbol{\alpha}}\log p(\mathbf{z}^{(t)}|\boldsymbol{\alpha}))^2}. \tag{12}$$

Table S7 shows, on simulated problems, the merits of applying this variance reduction approach, as it typically results in a better performance (in particular for the regression problems), both in terms of prediction and feature ranking quality, and smaller feature subsets.

### 2.2.3 Updating the gradient

The above procedure allows us to estimate the gradient and to perform one gradient descent step. After this step, the distribution parameters $\boldsymbol{\alpha}$ are updated to $\boldsymbol{\beta} = \boldsymbol{\alpha} - \eta\nabla F$ and we must hence compute the gradient $\nabla_{\boldsymbol{\beta}}\mathbb{E}[f_{\mathbf{z}}(\mathbf{x}_i)]_{p(\mathbf{z}|\boldsymbol{\beta})}$ in order to do the next step. To be able to compute the approximation in Eq (10), new models $\{f_{\mathbf{z}^{(t)}}\}_{t=1}^{T}$ must thus in principle be learned by sampling each $\mathbf{z}^{(t)}$ from the new distribution $p(\mathbf{z}|\boldsymbol{\beta})$. This would result in a very computationally expensive algorithm where new models are learned after each parameter update.

In order to estimate the effect of a change in the feature selection probabilities $\boldsymbol{\alpha}$ without learning new models, we use the *importance sampling* approximation of the expectation. Given a new vector of feature selection probabilities $\boldsymbol{\beta} \neq \boldsymbol{\alpha}$, any expectation under $p(\mathbf{z}|\boldsymbol{\beta})$ can be approximated through $p(\mathbf{z}|\boldsymbol{\alpha})$. We have, for any input $\mathbf{x}_i$:

$$\mathbb{E}[(f_{\mathbf{z}}(\mathbf{x}_i) - b)\nabla_{\boldsymbol{\beta}}\log p(\mathbf{z}|\boldsymbol{\beta})]_{p(\mathbf{z}|\boldsymbol{\beta})} = \sum_{t=1}^{|\mathcal{Z}|}\frac{p(\mathbf{z}|\boldsymbol{\beta})}{p(\mathbf{z}|\boldsymbol{\alpha})}p(\mathbf{z}|\boldsymbol{\alpha})(f_{\mathbf{z}^t}(\mathbf{x}_i) - b)\nabla_{\boldsymbol{\beta}}\log p(\mathbf{z}|\boldsymbol{\beta}) \tag{13}$$

$$= \mathbb{E}\left[\frac{p(\mathbf{z}|\boldsymbol{\beta})}{p(\mathbf{z}|\boldsymbol{\alpha})}(f_{\mathbf{z}}(\mathbf{x}_i) - b)\nabla_{\boldsymbol{\beta}}\log p(\mathbf{z}|\boldsymbol{\beta})\right]_{p(\mathbf{z}|\boldsymbol{\alpha})} \tag{14}$$

$$\simeq \frac{1}{T}\sum_{t=1}^{T}\frac{p(\mathbf{z}^{(t)}|\boldsymbol{\beta})}{p(\mathbf{z}^{(t)}|\boldsymbol{\alpha})}(f_{\mathbf{z}^{(t)}}(\mathbf{x}_i) - b)\nabla_{\boldsymbol{\beta}}\log p(\mathbf{z}^{(t)}|\boldsymbol{\beta}), \tag{15}$$

where the feature subsets $\{\mathbf{z}^{(t)}\}_{t=1}^{T}$ in Eq (15) have been sampled from $p(\mathbf{z}|\boldsymbol{\alpha})$. This approximation can thus be computed for any $\boldsymbol{\beta}$ by using the models $\{f_{\mathbf{z}^{(t)}}\}_{t=1}^{T}$ obtained when the $\mathbf{z}^{(t)}$ were sampled from $p(\mathbf{z}|\boldsymbol{\alpha})$.

As shown by Eq.(15), the importance sampling approximation consists of a weighted average over $T$ feature subsets $\mathbf{z}^{(t)}$, using weights $w_t = \frac{p(\mathbf{z}^{(t)}|\boldsymbol{\beta})}{p(\mathbf{z}^{(t)}|\boldsymbol{\alpha})}$. When $\boldsymbol{\beta}$ becomes very different from $\boldsymbol{\alpha}$, some of the feature subsets will be hardly used in the average because they will have a very low weight $w_t$. The effective number of used feature subsets can be computed as (Doucet et al., 2001):

$$T_{eff} = \frac{\left(\sum_{t=1}^{T} w_t\right)^2}{\sum_{t=1}^{T} w_t^2}. \tag{16}$$

With imbalanced weights, the importance sampling approximation is equivalent to averaging over $T_{eff}$ feature subsets. When $T_{eff}$ is too low, the gradient estimation thus becomes unreliable. When this happens, we train $T$ new models $f_{\mathbf{z}^{(t)}}$ by sampling the feature subsets $\mathbf{z}^{(t)}$ from the current distribution $p(\mathbf{z}|\boldsymbol{\beta})$. In practice, new models are trained as soon as $T_{eff}$ drops below $0.9T$.

## 2.3 Discussion

The PRS algorithm has the advantage of being model-agnostic in that any supervised learning method can be used to fit the $f_{\mathbf{z}^{(t)}}$ models. Despite the use of gradient descent, no hypothesis of differentiability is required for the model family. The framework can also be easily adapted to any differentiable loss and regularization term.

**Computational complexity**  Once the models are trained, the computation of the gradient is linear with respect to the number $N$ of samples, the number $M$ of features and the number $T$ of base models in the ensemble. The costliest step of the algorithm is the construction of the base models. The complexity of the construction of the models depends on the type of model, but note that each model is grown only from a potentially small subset of features. Figure S4 in the appendix shows the computing times of PRS on simulated problems, for different values of $N$ and $M$.

**Regularization**  While we have not used any regularization term in (3), incorporating one is straight-forward. A natural regularization term to enforce sparsity could be simply the sum $\sum_{j=1}^{M} \alpha_j$, which can be nicely interpreted as $\mathbb{E}[||\mathbf{z}||_0]_{p(\mathbf{z}|\boldsymbol{\alpha})}$, i.e., the average size of the subsets drawn from $p(\mathbf{z}|\boldsymbol{\alpha})$. Adding this term to (3) with a regularization coefficient $\lambda$ would simply consists in adding $\lambda$ to the gradient in (4). We did not systematically include such regularization in our experiments below to reduce the number of hyper-parameters. Despite the lack of regularization, PRS has a natural propensity for selecting few features. Incorporating a useless feature $j$ will indeed often deteriorate the quality of the predictions and lead to a decrease of the corresponding $\alpha_j$. Note however that the sparsity of the resulting selection weights will depend on the robustness of the learning algorithm to the presence of irrelevant features. This will be illustrated in our experiments. Besides sparsity, we will also exploit more sophisticated regularization terms, for MNIST (where we will use a regularization term that enforces spatial smoothness) and the inference of gene regulatory networks (where we will enforce modular networks).

**Related works**  Our method has direct connections with the Random Subspace (RS) ensemble method (Ho, 1998). In addition to providing a feature ranking, it has the obvious added flexibility w.r.t. RS that the feature sampling distribution (and thus also the subspace size) is automatically adapted to improve performance. Another close work is the RaSE method (Tian & Feng, 2021), which iteratively samples a large population of feature subsets, trains models from them and selects the $T$ best ones according to a chosen criterion (e.g., the cross-validation prediction performance). The selection probability $\alpha_j$ of each feature is then updated as the proportion of times it appears in the $T$ best feature subsets. RaSE and PRS are thus similar in the sense that they both iteratively sample feature subsets from an explicit probability distribution with parameters $\boldsymbol{\alpha}$ (although the sampling distribution is different between the two approaches) and update the latter. One major difference is that RaSE implicitly minimizes the expected value of the loss function:

$$\min_{\boldsymbol{\alpha}} E\left[\frac{1}{N}\sum_{i=1}^{N} L(y_i, f_{\mathbf{z}}(\mathbf{x}_i))\right]_{p(\mathbf{z}|\boldsymbol{\alpha})}, \tag{17}$$

while we are trying to minimize the loss of the *ensemble* model $\mathbb{E}[f_{\mathbf{z}}(\mathbf{x})]_{p(\mathbf{z}|\boldsymbol{\alpha})}$ (see Eq.(3)). Both approaches also greatly differ in the optimization technique: RaSE iteratively updates the parameters $\boldsymbol{\alpha}$ from the best feature subsets in the current population, while PRS is based on gradient descent and importance sampling. Furthermore, as explained above, PRS allows the direct regularization of the parameters $\boldsymbol{\alpha}$, while such regularization is not possible in RaSE. Finally, RaSE samples the size of each feature subset from a uniform distribution whose upper bound is a hyper-parameter set by the user, while the subspace size is automatically adapted in our approach. Both approaches will be empirically compared in Section 3.

Our optimization procedure has also some links with variational optimization (VO, Staines & Barber, 2013). VO is a general technique for minimizing a function $G(\mathbf{z})$ that is non-differentiable or combinatorial. It is based on the bound:

$$\min_{\mathbf{z}\in\mathcal{Z}} G(\mathbf{z}) \leq E[G(\mathbf{z})]_{p(\mathbf{z}|\boldsymbol{\alpha})} = F(\boldsymbol{\alpha}), \tag{18}$$

Instead of minimizing $G$ with respect to $\mathbf{z}$, one can thus minimize the upper bound $F$ with respect to $\boldsymbol{\alpha}$. Replacing $G$ in Eq. (18) with the loss of an individual model $f_{\mathbf{z}}$ yields:

$$\min_{\mathbf{z} \in \mathcal{Z}} \frac{1}{N} \sum_{i=1}^{N} L(y_i, f_{\mathbf{z}}(\mathbf{x}_i)) \leq E \left[ \frac{1}{N} \sum_{i=1}^{N} L(y_i, f_{\mathbf{z}}(\mathbf{x}_i)) \right]_{p(\mathbf{z}|\boldsymbol{\alpha})}, \tag{19}$$

where the left-hand term is the definition of the global feature selection problem, which is combinatorial over the discrete values of $\mathbf{z}$. Instead of directly solving the feature selection problem, one can thus minimize an upper bound of it, by minimizing the expected value of the loss over the continuous $\boldsymbol{\alpha}$, e.g. using gradient descent. Like in VO, the formulation in (3) allows us to use gradient descent optimization despite the fact that the models $f_{\mathbf{z}}$ are not necessarily differentiable. Note however that our goal is not to solve the feature selection problem, but to train an ensemble and thus the function $F(\boldsymbol{\alpha})$ in Eq. (3), which is the loss of the ensemble, is *exactly* what we want to minimize (and not an upper bound).

Several works have used gradient descent to solve the feature selection problem in the left-hand term of Eq. (19), by using a continuous relaxation of the discrete variables $\mathbf{z}$ (Sheth & Fusi, 2020; Yamada et al., 2020; Donà & Gallinari, 2021; Balin et al., 2019; Yang et al., 2022). However, these methods are designed to be used with differentiable models (neural networks, polynomial models), so that both the feature selection and the model parameters can be updated in a single gradient descent step, while PRS is model-agnostic.

Note that while PRS is a model-agnostic ensemble method, it is not an explanation (or post-hoc) method, such as LIME (Ribeiro et al., 2016) or SHAP (Lundberg & Lee, 2017) for example. Methods such as LIME or SHAP are designed to highlight the features that a pre-trained black-box model uses to produce its predictions (locally or globally). They do not affect the predictive performance of the models they try to explain. PRS, on the other hand, produces an ensemble with hopefully improved predictive performance and interpretability with respect to (and whatever) the base learning algorithm it is combined with.

## 3 Results

We compare below PRS against several baselines and state-of-the-art methods on simulated (Section 3.1) and real (Section 3.2) problems. We then conduct two additional experiments, on MNIST (Section 3.3) and gene network inference (Section 3.4), to highlight the benefit of incorporating a problem-specific regularization term.

As base model $f_{\mathbf{z}}$, we used either a CART decision tree (Breiman et al., 1984a), a $k$-nearest neighbors (kNN) model (Altman, 1992) with $k = 5$ or a support vector machine (SVM, Boser et al., 1992) with a radial basis function kernel. All the hyper-parameters of these base models were set to the default values used in the scikit-learn library (Pedregosa et al., 2011)

We report the predictive performance with the $R^2$ score for regression problems and the accuracy for classification problems. For PRS a ranking of features can be obtained by sorting them by decreasing value of importances $\boldsymbol{\alpha}$. If the relevant variables are known, the feature ranking can be evaluated using the area under the precision-recall curve (AUPR). A perfect ranking (i.e. all the relevant features have a higher importance than the irrelevant ones) yields an AUPR of 1, while a random ranking has an AUPR close to the proportion of relevant features.

We compare PRS to the following methods: the standard Random Subspace (RS), Random Forest (RF), Gradient Boosting with Decision Trees (GBDT), and RaSE. Implementation details for all the methods are provided in Appendix D.

### 3.1 Simulated Problems

We simulated four problems, for which the relevant features are known (see Appendix E.1 for the detailed simulation protocol). Compared to single base models and RS, PRS yields higher prediction scores for all the base models (Figure 2). The improvement of performance over RS is larger in the case of kNN and SVM, compared to decision trees. This can be explained by the fact a decision tree, contrary to kNN and

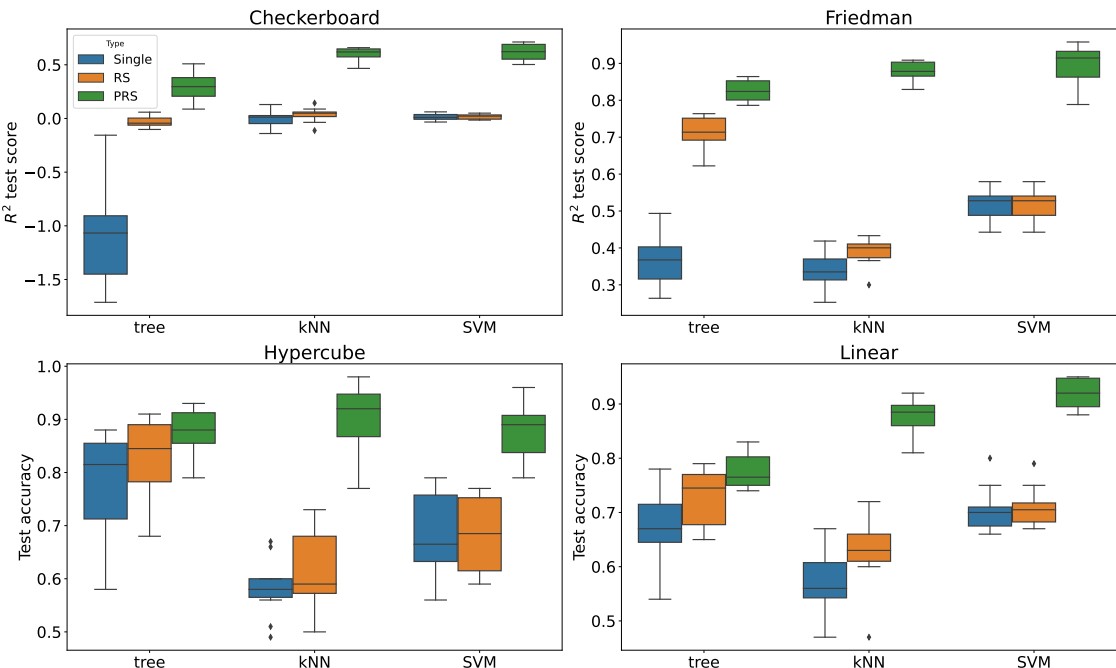

Figure 2: **Predictive performance for single models versus RS and PRS ensembles.** Performance values are $R^2$ scores for the Checkerboard and Friedman problems, and accuracies for the Hypercube and Linear problems. The boxplots summarize the values over 10 datasets.

SVM, has an inner feature selection mechanism and is hence able to maintain a good performance even in the presence of irrelevant features. Therefore, for a given irrelevant feature $j$, the difference between $f_{j,0}^{\boldsymbol{\beta}}$ and $f_{j,1}^{\boldsymbol{\beta}}$ (Eqs (7) and (8), respectively) will be lower in the case of trees, which can prevent the corresponding $\alpha_j$ to decrease towards zero during the gradient descent.

Note also that RS greatly improves over the single model only in the case of decision trees. The decision tree being a model that is prone to high variance, its prediction performance is indeed usually improved by using ensemble methods (Breiman, 1996a; 2001; Geurts et al., 2006). On the other hand, since kNN and SVM have a sufficiently low variance, their performance is not improved with a standard ensemble.

While the degree of randomization is controlled by the parameter $K$ (i.e. the number of randomly sampled features for each base model) in RS, it has the advantage to be automatically tuned in PRS. Table 1 indicates the sum $\sum_{j=1}^{M} \alpha_j$, which is equivalent to $\mathbb{E}[||\mathbf{z}||_0]_{p(\mathbf{z}|\boldsymbol{\alpha})}$, i.e. the average number of selected variables per base model. By comparing this average number to the optimal value of $K$ for RS, we can see that PRS effectively selects a much lower number of features, while no explicit constraint on sparsity is used during model training. The average number of selected variables remains however slightly higher than the actual number of relevant features, indicating that a certain degree of randomization is introduced during model construction.

Overall, PRS outperforms RF and GBDT both in terms of predictive performance and feature ranking (Table 2), the best performance being obtained with kNN and SVM. For some problems (e.g. Checkerboard), PRS is particularly better than RF and GBDT in the presence of a high number of irrelevant features (Figures S5 and S6).

Compared to RaSE, PRS yields an equivalent performance, with equivalent feature subset sizes (Table S8), while being much less computationally expensive. Indeed, $T \times B \times 10 = 500,000$ base models must be trained *at each iteration* of RaSE (see Appendix D.4), while in PRS the highest number of trained base models is 320,000 for the *whole run* of the algorithm (Table S9).

Table 1: **Number of features used per base model**, i.e. for RS: the number $K$ of randomly sampled features (optimized on the validation set), and for PRS: the sum $\sum_{j=1}^{M} \alpha_j$. Values are mean and standard deviation over 10 datasets.

| Model | | Checkerboard | Friedman | Hypercube | Linear |
|---|---|---|---|---|---|
| tree | RS | $53.50 \pm 52.13$ | $132.00 \pm 24.49$ | $133.20 \pm 63.76$ | $145.60 \pm 63.17$ |
| | PRS | $7.29 \pm 1.32$ | $7.82 \pm 0.75$ | $12.10 \pm 3.04$ | $12.68 \pm 2.95$ |
| kNN | RS | $97.10 \pm 86.99$ | $90.60 \pm 28.15$ | $61.70 \pm 47.24$ | $99.10 \pm 80.20$ |
| | PRS | $6.49 \pm 1.91$ | $5.49 \pm 0.47$ | $6.83 \pm 1.20$ | $10.12 \pm 2.30$ |
| SVM | RS | $111.00 \pm 126.96$ | $305.00 \pm 0.00$ | $131.80 \pm 95.29$ | $227.30 \pm 84.00$ |
| | PRS | $4.85 \pm 0.55$ | $7.28 \pm 1.31$ | $13.54 \pm 3.01$ | $19.97 \pm 1.90$ |

Table 2: **Comparison to RF and GBDT.** We report here the prediction score on the test set ($R^2$ score or accuracy) and the feature ranking quality (AUPR). Values are mean and standard deviation over 10 datasets. Highest scores are indicated in bold type.

| | | RF | GBDT | PRS - tree | PRS - kNN | PRS - SVM |
|---|---|---|---|---|---|---|
| Checkerboard | $R^2$ | $-0.03 \pm 0.05$ | $-0.09 \pm 0.10$ | $0.29 \pm 0.14$ | $0.60 \pm 0.06$ | $\mathbf{0.62 \pm 0.07}$ |
| | AUPR | $0.44 \pm 0.22$ | $0.40 \pm 0.24$ | $0.60 \pm 0.23$ | $0.92 \pm 0.14$ | $\mathbf{0.98 \pm 0.06}$ |
| Friedman | $R^2$ | $0.73 \pm 0.04$ | $0.86 \pm 0.03$ | $0.83 \pm 0.03$ | $0.88 \pm 0.03$ | $\mathbf{0.90 \pm 0.05}$ |
| | AUPR | $0.68 \pm 0.03$ | $0.87 \pm 0.04$ | $0.95 \pm 0.04$ | $\mathbf{1.00 \pm 0.00}$ | $0.98 \pm 0.05$ |
| Hypercube | Accuracy | $0.85 \pm 0.07$ | $0.86 \pm 0.06$ | $0.88 \pm 0.04$ | $\mathbf{0.90 \pm 0.06}$ | $0.88 \pm 0.05$ |
| | AUPR | $0.92 \pm 0.12$ | $0.90 \pm 0.10$ | $\mathbf{0.97 \pm 0.06}$ | $0.94 \pm 0.09$ | $0.90 \pm 0.14$ |
| Linear | Accuracy | $0.77 \pm 0.06$ | $0.85 \pm 0.04$ | $0.78 \pm 0.03$ | $0.88 \pm 0.03$ | $\mathbf{0.92 \pm 0.03}$ |
| | AUPR | $0.68 \pm 0.13$ | $0.70 \pm 0.15$ | $0.67 \pm 0.13$ | $0.73 \pm 0.12$ | $\mathbf{0.80 \pm 0.10}$ |

Finally, the efficiency of the importance sampling approach can be observed in Table S10. This table shows the performance and training times of PRS-SVM, for different thresholds on the effective number of models $T_{eff}$ as defined in Eq. (16). We recall that in PRS, new base models are trained only when $T_{eff}$ drops below the chosen threshold. Setting the threshold to $T$ corresponds to the case where we do not use the importance sampling approach and new models are trained at each epoch. This significantly increases the computing time, with no improvement in terms of prediction score and AUPR, compared to our default threshold $0.9T$. Lowering the threshold allows to decrease the training time, and there is a strong degradation of the performance only for small threshold values ($0.3T$ and $0.5T$).

## 3.2 Real-world problems

We compared the different approaches on benchmarks containing real-world datasets:

- The *tabular* benchmark of Grinsztajn et al. (2022). This benchmark contains 55 tabular datasets from various domains, split into four groups (regression or classification, with or without categorical features). Dataset sizes are indicated in Tables S11 and S12. For each dataset, we randomly choose 3,000 samples, that we split to compose the training, validation and test sets (1,000 samples each). The results of the different approaches are then averaged over 10 such random samplings.

- Biological, classification datasets from the *scikit-feature* repository (Li et al., 2018). These datasets (also tabular) have the particularity to have very few ($\sim 100$) samples for several thousands features. Among the biological datasets available in the repository, we filtered out datasets and classes in order to have only datasets with at least 30 samples per class. The final dataset sizes are indicated in Table S12. Given the small dataset sizes, we estimate the prediction accuracies on these datasets with 5-fold cross-validation, and for each fold we use 80% of the training set to train the models and the remaining 20% as validation set. Given the very high number of features in these datasets, we add in the objective function of PRS a regularization term that enforces sparsity (see Section 2.3),

Table 3: **Normalized prediction scores on the real benchmarks.** To aggregate the performance across the datasets of each benchmark, we first normalize the performance score ($R^2$ or accuracy) between 0 and 1 via an affine renormalization between the worse- and top-performing methods for each dataset. The normalized scores are then averaged over the different datasets and 10 data subsamplings (for the tabular datasets) or 5 cross-validation folds (for the scikit-feature datasets). For each benchmark, the highest performance is indicated in bold type.

| | Model | Tabular Regression | Tabular Classification | Scikit-feature Classification |
|---|---|---|---|---|
| tree | Single | $0.36 \pm 0.41$ | $0.17 \pm 0.22$ | $0.19 \pm 0.25$ |
| | RS | $0.82 \pm 0.18$ | $0.69 \pm 0.22$ | $0.67 \pm 0.26$ |
| | RaSE | $0.83 \pm 0.18$ | $0.70 \pm 0.25$ | $0.60 \pm 0.23$ |
| | PRS | $0.92 \pm 0.11$ | $0.80 \pm 0.16$ | $0.64 \pm 0.28$ |
| kNN | Single | $0.50 \pm 0.33$ | $0.15 \pm 0.20$ | $0.38 \pm 0.29$ |
| | RS | $0.65 \pm 0.26$ | $0.52 \pm 0.21$ | $0.35 \pm 0.26$ |
| | RaSE | $0.92 \pm 0.09$ | $0.67 \pm 0.24$ | $0.71 \pm 0.26$ |
| | PRS | $0.95 \pm 0.08$ | $0.79 \pm 0.20$ | $0.67 \pm 0.27$ |
| SVM | Single | $0.48 \pm 0.37$ | $0.52 \pm 0.24$ | $0.53 \pm 0.26$ |
| | RS | $0.49 \pm 0.38$ | $0.54 \pm 0.23$ | $0.49 \pm 0.26$ |
| | RaSE | $0.72 \pm 0.30$ | $0.63 \pm 0.25$ | $0.67 \pm 0.29$ |
| | PRS | $0.77 \pm 0.26$ | $0.56 \pm 0.28$ | $\mathbf{0.79 \pm 0.26}$ |
| | RF | $0.93 \pm 0.09$ | $\mathbf{0.84 \pm 0.20}$ | $0.61 \pm 0.28$ |
| | GBDT | $\mathbf{0.96 \pm 0.09}$ | $0.83 \pm 0.21$ | $0.60 \pm 0.22$ |

and we select the value of the regularization coefficient $\lambda$ (among $\{0.0001, 0.001, 0.01, 0.1\}$) that maximizes the accuracy on the validation set.

To aggregate the prediction performance across multiple datasets, we first normalize the performance score ($R^2$ or accuracy) between 0 and 1 via an affine renormalization between the worse- and top-performing methods for each dataset. The normalized scores are then averaged over the different datasets and the 10 data subsamplings (for the tabular datasets) or 5 cross-validation folds (for the scikit-feature datasets).

Table 3 shows the aggregated prediction scores, while the raw scores for each dataset can be found in Tables S13-S16. PRS always improves over RS, except on the scikit-feature benchmark with decision trees, and is also usually better than RaSE. PRS-SVM yields the highest performance on the scikit-feature benchmark, but RF and GBDT remain the best performers on the tabular benchmarks, with an equivalent performance of PRS-kNN on the regression datasets.

Overall, PRS and RaSE ensembles are sparser than RS, when comparing the (expected) number of selected features per base model (Tables S17-S19). In particular, RaSE returns very sparse models on the scikit-feature benchmark, as for these datasets the maximum feature subspace size is explicitly set to $\sqrt{N}$ (see Appendix D.4), where the number $N$ of samples is very small (between 100 and 200).

## 3.3 MNIST

We applied our method to classify images of handwritten digits 5's and 6's. The images were taken from the MNIST dataset (LeCun et al., 1998) and random noise was added to them to make the task more challenging (Figure 3). We treated the image pixels as individual features and we used the following objective function within PRS:

$$F(\boldsymbol{\alpha}) = \frac{1}{N} \sum_{i=1}^{N} L\left(y_i, \mathbb{E}[f_\mathbf{z}(\mathbf{x}_i)]_{p(\mathbf{z}|\boldsymbol{\alpha})}\right) + \lambda_1 \sum_{j=1}^{W} \sum_{k=1}^{H} \alpha_{j,k} + \lambda_2 \left( \sum_{j=2}^{H} |\alpha_{j,k} - \alpha_{j-1,k}| + \sum_{k=2}^{W} |\alpha_{j,k} - \alpha_{j,k-1}| \right), \quad (20)$$

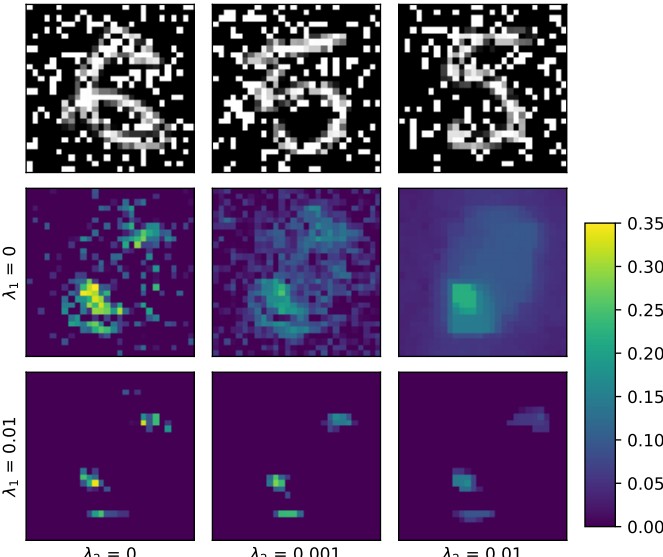

Figure 3: **PRS-kNN feature selection probabilities on MNIST.** The first row shows three exemples of (noisy) images from the dataset. The second and third rows show the values of the parameters $\boldsymbol{\alpha}$ for different values of the regularization coefficients $\lambda_1$ and $\lambda_2$. Increasing $\lambda_1$ enforces sparsity, while increasing $\lambda_2$ enforces spatial smoothness.

Table 4: **Test accuracies on MNIST.** The right-hand part of the table indicates the accuracies of PRS when the hyper-parameters $\lambda_1$ and $\lambda_2$ are optimized on the validation set.

|  |  | Without regularization |  |  | With regularization |  |  |
| --- | --- | --- | --- | --- | --- | --- | --- |
| RF | GBDT | PRS-tree | PRS-kNN | PRS-SVM | PRS-tree | PRS-kNN | PRS-SVM |
| 0.969 | **0.982** | 0.976 | 0.938 | 0.954 | **0.982** | 0.955 | 0.959 |

where $W$ and $H$ are respectively the width and height of the image, and $\alpha_{j,k}$ is the selection probability for the pixel in the $j$-row and $k$-th column. The second term is a regularization term that enforces sparsity, while the last term penalizes large differences between the $\alpha_{j,k}$ parameters corresponding to neighbouring pixels (Figure 3). Such regularization is known as the *fused lasso* (Tibshirani et al., 2005) and allows to account for the spatial structure of the features. Without any regularization ($\lambda_1 = 0, \lambda_2 = 0$), the pixels with strictly positive feature selection probabilities tend to be spread over the whole digit (Figure 3). As regularization is increased, they cluster around the bottom-left of the digit (as expected since this is where 5's and 6's differ in the images) and the prediction performance is improved (Table 4). The highest accuracies are obtained with PRS-tree and GBDT.

### 3.4 Gene network inference

An open problem in computational biology is the reconstruction of gene regulatory networks, which attempt to explain the joint variability in the expression levels of a group of genes through a sparse pattern of interactions. One approach to gene network reconstruction is the application of a feature selection approach that identifies the regulators of each target gene. Such approach is used by GENIE3, one of the current state-of-the-art network inference algorithms (Huynh-Thu et al., 2010). This method learns for each target gene a RF model predicting its expression from the expressions of all the candidate regulators, and identifies the regulators of that target gene through the RF-based feature importance scores. The PRS and RaSE approaches can be used in the same way for gene network inference, with however the advantage that the base models are not restricted to decision trees. Furthermore, while in GENIE3 the different models,

Table 5: **AUPRs obtained on the DREAM4 networks.** The highest AUPR is indicated in bold type for each network. *Random* indicates the AUPR of an approach that randomly ranks all the possible edges. The table also indicates the average subspace size for RaSE and PRS, i.e. for RaSE: the average subspace size over the $T \times G$ models, and for PRS: the sum $\sum_{j=1}^{M} \alpha_{j,g}$, averaged over the $G$ genes.

| | | AUPR | | | | | Subspace size | | | | |
| | | Net1 | Net2 | Net3 | Net4 | Net5 | Net1 | Net2 | Net3 | Net4 | Net5 |
|---|---|---|---|---|---|---|---|---|---|---|---|
| Random | | 0.02 | 0.02 | 0.02 | 0.02 | 0.02 | — | — | — | — | — |
| GENIE3 | RF | 0.17 | 0.15 | 0.25 | 0.23 | 0.22 | — | — | — | — | — |
| RaSE | tree | 0.08 | 0.07 | 0.16 | 0.15 | 0.12 | 5.93 | 5.90 | 6.19 | 6.12 | 5.98 |
| | kNN | 0.09 | 0.07 | 0.15 | 0.13 | 0.13 | 6.07 | 6.18 | 6.06 | 6.22 | 6.21 |
| | SVM | 0.08 | 0.07 | 0.13 | 0.11 | 0.10 | 5.34 | 5.82 | 5.00 | 5.38 | 5.57 |
| PRS | tree | 0.14 | 0.09 | 0.19 | 0.15 | 0.15 | 4.63 | 4.71 | 5.19 | 5.13 | 4.96 |
| $\lambda = 0$ | kNN | 0.15 | 0.11 | 0.21 | 0.19 | 0.21 | 5.14 | 4.68 | 5.84 | 5.39 | 5.11 |
| | SVM | 0.13 | 0.10 | 0.18 | 0.16 | 0.19 | 5.64 | 4.74 | 5.86 | 5.77 | 5.45 |
| PRS | tree | 0.09 | 0.11 | 0.17 | 0.18 | 0.11 | 0.28 | 0.27 | 0.45 | 0.41 | 0.33 |
| $\lambda > 0$ | kNN | 0.16 | **0.19** | 0.25 | **0.24** | 0.21 | 0.30 | 0.30 | 5.84 | 5.39 | 0.54 |
| | SVM | **0.18** | 0.16 | **0.27** | 0.19 | **0.24** | 0.50 | 0.51 | 1.60 | 1.13 | 0.93 |

corresponding to the different target genes, are learned independently of each other, PRS can be extended to introduce a global constraint on the topology of the network.

More specifically, we use a joint regularizer that enforces modular networks, a property often encountered in real gene regulatory networks. Let $G$ be the number of genes, among which there are $M$ candidate regulators, and let $\mathbf{x}_i \in \mathbb{R}^M$ and $\mathbf{y}_i \in \mathbb{R}^G$ be respectively the expression levels of the candidate regulators and of the $G$ target genes in the $i$-th sample ($i = 1, \ldots, N$). Our goal is to identify a $M \times G$ matrix $\boldsymbol{\alpha}$, where $\alpha_{j,g}$ is the weight of the regulatory link directed from the $j$-th candidate regulator to the $g$-th gene. In the context of PRS, we seek to identify the matrix $\boldsymbol{\alpha}$ that minimizes the following objective function:

$$\frac{1}{G} \frac{1}{N} \sum_{g=1}^{G} \sum_{i=1}^{N} \left( y_{i,g} - \mathbb{E}[f_{\mathbf{z}}(\mathbf{x}_i)]_{p(\mathbf{z}|\boldsymbol{\alpha}_{\cdot,g})} \right)^2 + \lambda \sum_{j=1}^{M} \sqrt{\sum_{g=1}^{G} \alpha_{j,g}^2}, \tag{21}$$

where $y_{i,g}$ is the expression of the $g$-th gene in the $i$-th sample and $\boldsymbol{\alpha}_{\cdot,g}$ denotes the $g$-th column of the matrix $\boldsymbol{\alpha}$. The second term in the above objective function is a joint regularizer (with a coefficient $\lambda$) that enforces structured sparsity, by enforcing the selection of as few rows as possible in $\boldsymbol{\alpha}$ (Jenatton et al., 2011). Using this joint regularizer will result in modular networks where only a few regulators control the expressions of the different genes.

We evaluate the ability of PRS to reconstruct the five 100-gene networks of the DREAM4 *Multifactorial Network* challenge (Marbach et al., 2010; 2012), for which GENIE3 was deemed the best performer. The DREAM4 networks are artificial networks for which the true regulatory links are known and an AUPR can thus be computed given a predicted ranking of links. To reconstruct each network, a simulated gene expression dataset with 100 samples was made available to the challenge participants.

The regularization coefficient $\lambda$ determines the number of used candidate regulators (Figure S7), and we selected the value of $\lambda$ that yields the lowest prediction error on the validation set. Adding the regularization term sometimes deteriorates the AUPR of PRS-tree, but it can greatly help PRS-kNN and PRS-SVM (Table 5). The two latter methods yield the highest AUPRs, while RaSE is the worse performer. The bad performance of RaSE compared to PRS could be explained by the lack of regularization, which leads to a higher number of used candidate regulators per base model (Table 5).

## 4 Conclusions

We proposed a model-agnostic ensemble method that aggregates base models independently trained on feature subsets sampled from a Bernoulli distribution. We show that the parameters of the latter distribution

can be trained using gradient descent even if the base models are not differentiable. The required iterative gradient computations can furthermore be performed efficiently by exploiting importance sampling. The resulting approach uniquely combines several interesting features: it is fully model-agnostic, it can use any combination of differentiable loss function and regularization term, and it provides variable importance scores. Experiments show that PRS almost always improves over standard RS and is competitive with respect to RF, GBDT and RaSE, both in terms of predictive performance and feature ranking quality. We also showed that an appropriate regularization strategy allows PRS to outperform the state-of-the-art GENIE3 in the inference of gene regulatory networks.

While we adopted an ensemble strategy, the same optimization technique, combining gradient descent and importance sampling, can be used to solve the feature selection problem as defined in (17) and addressed also by RaSE. It would be interesting to investigate this approach and compare it with the ensemble version explored in this paper. Note however that it would require to exploit a stronger learning algorithm, because it would not benefit from the ensemble averaging effect. Applying this technique, and its associated derivation of feature importance scores, on top of modern deep learning models would be also highly desirable given the challenge to explain these models. This would require however to develop specific strategies to reduce the non negligible computational burden that would arise when training multiple ensembles of deep, complex models. Finally, exploiting more complex feature subset distributions, beyond independent Bernoulli distributions, would be also very interesting but adapting the optimization strategy might not be trivial.

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

## A    Pseudo-code

---

**Algorithm S1** PRS training

---

1: **Input:** dataset $\mathcal{D} = \{(\mathbf{x}_i, y_i)\}_{i=1}^N$, number of models $T$, batch size $N_b$, number of epochs $n_{epochs}$.

2: **Output:** Feature selection probabilities $\boldsymbol{\alpha} = [\alpha_1, \alpha_2, \ldots, \alpha_M]^\top$, where $M$ is the number of input features, and a trained ensemble model.

3: **for** $j = 1$ **to** $M$ **do**

4:     $\alpha_j \leftarrow 5/T$

5: **end for**

6: **for** each batch $\mathcal{D}_b = \{(\mathbf{x}_{(i)}, y_{(i)})\}_{i=1}^{N_b} \subset \mathcal{D}$ **do**

7:     **for** $t = 1$ **to** $T$ **do**

8:         Draw a feature subset $\mathbf{z}^{(t)}$ from $p(\mathbf{z}|\boldsymbol{\alpha})$

9:         Learn a model $f_{\mathbf{z}^{(t)}}$ from $\mathbf{z}^{(t)}$ and $\mathcal{D}\backslash\mathcal{D}_b$

10:     **end for**

11: **end for**

12: $\boldsymbol{\beta} \leftarrow \boldsymbol{\alpha}, k \leftarrow 0$

13: **repeat**

14:     **for** each batch $\mathcal{D}_b = \{(\mathbf{x}_{(i)}, y_{(i)})\}_{i=1}^{N_b} \subset \mathcal{D}$ **do**

15:         **for** $i = 1$ **to** $N_b$ **do**

16:             $baseline = \dfrac{\sum_{t=1}^T (\nabla_{\boldsymbol{\beta}} \log p(\mathbf{z}^{(t)}|\boldsymbol{\beta}))^2 f_{\mathbf{z}^{(t)}}(\mathbf{x}_{(i)})}{\sum_{t=1}^T (\nabla_{\boldsymbol{\beta}} \log p(\mathbf{z}^{(t)}|\boldsymbol{\beta}))^2}$

17:             $\nabla_{\boldsymbol{\beta}} \mathbb{E}[f_{\mathbf{z}}(\mathbf{x}_{(i)})]_{p(\mathbf{z}|\boldsymbol{\beta})} \leftarrow \frac{1}{T} \sum_{t=1}^T \frac{p(\mathbf{z}^{(t)}|\boldsymbol{\beta})}{p(\mathbf{z}^{(t)}|\boldsymbol{\alpha})} (f_{\mathbf{z}^{(t)}}(\mathbf{x}_{(i)}) - baseline) \nabla_{\boldsymbol{\beta}} \log p(\mathbf{z}^{(t)}|\boldsymbol{\beta})$

18:         **end for**

19:         $\nabla_{\boldsymbol{\beta}} F \leftarrow \frac{1}{N_b} \sum_{i=1}^{N_b} \frac{\mathrm{d}L}{\mathrm{d}\mathbb{E}[f_{\mathbf{z}}(\mathbf{x}_{(i)})]_{p(\mathbf{z}|\boldsymbol{\beta})}} \nabla_{\boldsymbol{\beta}} \mathbb{E}[f_{\mathbf{z}}(\mathbf{x}_{(i)})]_{p(\mathbf{z}|\boldsymbol{\beta})}$

20:         $\boldsymbol{\beta} \leftarrow \mathrm{proj}(\boldsymbol{\beta} - \eta \nabla_{\boldsymbol{\beta}} F, [0, 1])$

21:     **end for**

22:     **for** $t = 1$ **to** $T$ **do**

23:         $w_t \leftarrow \frac{p(\mathbf{z}^{(t)}|\boldsymbol{\beta})}{p(\mathbf{z}^{(t)}|\boldsymbol{\alpha})}$

24:     **end for**

25:     $T_{eff} \leftarrow \dfrac{\left(\sum_{t=1}^T w_t\right)^2}{\sum_{t=1}^T w_t^2}$

26:     **if** $T_{eff} < 0.9T$ **then**

27:         $\boldsymbol{\alpha} \leftarrow \boldsymbol{\beta}$

28:         **for** each batch $\mathcal{D}_b = \{(\mathbf{x}_{(i)}, y_{(i)})\}_{i=1}^{N_b} \subset \mathcal{D}$ **do**

29:             **for** $t = 1$ **to** $T$ **do**

30:                 Draw a feature subset $\mathbf{z}^{(t)}$ from $p(\mathbf{z}|\boldsymbol{\alpha})$

31:                 Learn a model $f_{\mathbf{z}^{(t)}}$ from $\mathbf{z}^{(t)}$ and $\mathcal{D}\backslash\mathcal{D}_b$

32:             **end for**

33:         **end for**

34:     **end if**

35:     $k \leftarrow k + 1$

36: **until** $k = n_{epochs}$

37: $\boldsymbol{\alpha} \leftarrow \boldsymbol{\beta}$

38: **for** $t = 1$ **to** $T$ **do**

39:     Draw $\mathbf{z}^{(t)}$ from $p(\mathbf{z}|\boldsymbol{\alpha})$

40:     Learn a model $f_{\mathbf{z}^{(t)}}$ from $\mathcal{D}$ and $\mathbf{z}^{(t)}$

41: **end for**

42: **return** $\boldsymbol{\alpha}$ and $\{f_{\mathbf{z}^{(t)}}\}_{t=1}^T$

---

Algorithm S1 shows the pseudo-code for training a PRS model. Feature selection probabilities $\boldsymbol{\alpha}$ are first initialized to $\frac{5}{T}$ (lines 3-5). Given a batch $\mathcal{D}_b$, an ensemble of base models $f_{\mathbf{z}^{(t)}}$ are trained from $\mathcal{D}\backslash\mathcal{D}_b$, by drawing feature subsets from $p(\mathbf{z}|\boldsymbol{\alpha})$ (lines 7-10). The batch $\mathcal{D}_b$ is then used to estimate the gradient, using importance sampling approximation (lines 15-19), and the values of the feature selection probabilities are updated using projected gradient descent (line 20). When the effective number of feature subsets ($T_{eff}$) becomes too low, new models are trained (lines 26-34). Once the parameters $\boldsymbol{\alpha}$ are optimized, a final ensemble model is learned (lines 38-41).

## B   Computing the gradient

The score function method allows us to express the gradient of an expectation as an expectation itself. We have:

$$\nabla_{\boldsymbol{\alpha}} \mathbb{E}[f_{\mathbf{z}}(\mathbf{x})]_{p(\mathbf{z}|\boldsymbol{\alpha})} \quad = \quad \nabla_{\boldsymbol{\alpha}} \sum_{\mathbf{z}} p(\mathbf{z}|\boldsymbol{\alpha}) f_{\mathbf{z}}(\mathbf{x}) \tag{22}$$

$$= \quad \sum_{\mathbf{z}} f_{\mathbf{z}}(\mathbf{x}) \nabla_{\boldsymbol{\alpha}} p(\mathbf{z}|\boldsymbol{\alpha}) \tag{23}$$

$$= \quad \sum_{\mathbf{z}} f_{\mathbf{z}}(\mathbf{x}) p(\mathbf{z}|\boldsymbol{\alpha}) \nabla_{\boldsymbol{\alpha}} \log p(\mathbf{z}|\boldsymbol{\alpha}) \tag{24}$$

$$= \quad \mathbb{E}\left[f_{\mathbf{z}}(\mathbf{x}) \nabla_{\boldsymbol{\alpha}} \log p(\mathbf{z}|\boldsymbol{\alpha})\right]_{p(\mathbf{z}|\boldsymbol{\alpha})} \tag{25}$$

where to obtain Eq. (24), we used the equality $\nabla_{\boldsymbol{\alpha}} p(\mathbf{z}|\boldsymbol{\alpha}) = p(\mathbf{z}|\boldsymbol{\alpha}) \nabla_{\boldsymbol{\alpha}} \log p(\mathbf{z}|\boldsymbol{\alpha})$.

In the case of a Bernoulli distribution, we have:

$$p(\mathbf{z}|\boldsymbol{\alpha}) = \prod_{j=1}^{M} \alpha_j^{z_j} (1 - \alpha_j)^{(1-z_j)},$$

$$\log p(\mathbf{z}|\boldsymbol{\alpha}) = \sum_{j=1}^{M} z_j \log \alpha_j + (1 - z_j) \log(1 - \alpha_j).$$

By using Eq.(25), the $j$-th component of the gradient is given by:

$$\frac{\partial \mathbb{E}\left[f_{\mathbf{z}}(\mathbf{x})\right]_{p(\mathbf{z}|\boldsymbol{\alpha})}}{\partial \alpha_j} \quad = \quad \mathbb{E}\left[f_{\mathbf{z}}(\mathbf{x}) \frac{\partial}{\partial \alpha_j} \log p(\mathbf{z}|\boldsymbol{\alpha})\right]_{p(\mathbf{z}|\boldsymbol{\alpha})} \tag{26}$$

$$= \quad \mathbb{E}\left[f_{\mathbf{z}}(\mathbf{x}) \left(\frac{z_j}{\alpha_j} - \frac{1 - z_j}{1 - \alpha_j}\right)\right]_{p(\mathbf{z}|\boldsymbol{\alpha})} \tag{27}$$

$$= \quad \sum_{\mathbf{z}:z_j=1} f_{\mathbf{z}}(\mathbf{x}) \frac{p(\mathbf{z}|\boldsymbol{\alpha})}{\alpha_j} - \sum_{\mathbf{z}:z_j=0} f_{\mathbf{z}}(\mathbf{x}) \frac{p(\mathbf{z}|\boldsymbol{\alpha})}{1 - \alpha_j} \tag{28}$$

$$= \quad \sum_{\mathbf{z}:z_j=1} f_{\mathbf{z}}(\mathbf{x}) p(\mathbf{z}_{-j}|\boldsymbol{\alpha}_{-j}) - \sum_{\mathbf{z}:z_j=0} f_{\mathbf{z}}(\mathbf{x}) p(\mathbf{z}_{-j}|\boldsymbol{\alpha}_{-j}) \tag{29}$$

$$= \quad \mathbb{E}\left[f_{\mathbf{z}}(\mathbf{x})|z_j=1\right]_{p(\mathbf{z}_{-j}|\boldsymbol{\alpha}_{-j})} - \mathbb{E}\left[f_{\mathbf{z}}(\mathbf{x})|z_j=0\right]_{p(\mathbf{z}_{-j}|\boldsymbol{\alpha}_{-j})}, \tag{30}$$

where $\mathbf{z}_{-j} = \mathbf{z}\backslash z_j$ and $\boldsymbol{\alpha}_{-j} = \boldsymbol{\alpha}\backslash \alpha_j$.

## C   Estimating the gradient

### C.1   Estimation with baseline $b$

We have:

$$\mathbb{E}\left[\nabla_{\boldsymbol{\alpha}} \log p(\mathbf{z}|\boldsymbol{\alpha})\right]_{p(\mathbf{z}|\boldsymbol{\alpha})} = \sum_{\mathbf{z}} p(\mathbf{z}|\boldsymbol{\alpha}) \nabla_{\boldsymbol{\alpha}} \log p(\mathbf{z}|\boldsymbol{\alpha}) = \sum_{\mathbf{z}} \nabla_{\boldsymbol{\alpha}} p(\mathbf{z}|\boldsymbol{\alpha}) = \nabla_{\boldsymbol{\alpha}} \sum_{\mathbf{z}} p(\mathbf{z}|\boldsymbol{\alpha}) = \nabla_{\boldsymbol{\alpha}} 1 = 0. \tag{31}$$

Therefore, for any constant $b$, we have:

$$\mathbb{E}\left[(f_{\mathbf{z}}(\mathbf{x}) - b)\nabla_{\boldsymbol{\alpha}}\log p(\mathbf{z}|\boldsymbol{\alpha})\right]_{p(\mathbf{z}|\boldsymbol{\alpha})} = \mathbb{E}\left[f_{\mathbf{z}}(\mathbf{x})\nabla_{\boldsymbol{\alpha}}\log p(\mathbf{z}|\boldsymbol{\alpha})\right]_{p(\mathbf{z}|\boldsymbol{\alpha})} - b\mathbb{E}\left[\nabla_{\boldsymbol{\alpha}}\log p(\mathbf{z}|\boldsymbol{\alpha})\right]_{p(\mathbf{z}|\boldsymbol{\alpha})} \quad (32)$$

$$= \mathbb{E}\left[f_{\mathbf{z}}(\mathbf{x})\nabla_{\boldsymbol{\alpha}}\log p(\mathbf{z}|\boldsymbol{\alpha})\right]_{p(\mathbf{z}|\boldsymbol{\alpha})}. \quad (33)$$

## C.2 Optimal value of the baseline $b$

For readability, let us drop the subscript $p(\mathbf{z}|\boldsymbol{\alpha})$ in the expectations, i.e. $\mathbb{E}[\cdot] = \mathbb{E}[\cdot]_{p(\mathbf{z}|\boldsymbol{\alpha})}$, and let us define $h_{\mathbf{z}}(\boldsymbol{\alpha}) = \nabla_{\boldsymbol{\alpha}}\log p(\mathbf{z}|\boldsymbol{\alpha})$, with $\mathbb{E}[h_{\mathbf{z}}(\boldsymbol{\alpha})] = 0$. The gradient estimator is hence:

$$\mathbb{E}\left[(f_{\mathbf{z}}(\mathbf{x}) - b)h_{\mathbf{z}}(\boldsymbol{\alpha}))\right], \quad (34)$$

and its variance is given by:

$$V = \mathrm{var}\left[(f_{\mathbf{z}}(\mathbf{x}) - b)h_{\mathbf{z}}(\boldsymbol{\alpha}))\right] = \mathrm{var}[h_{\mathbf{z}}(\boldsymbol{\alpha})f_{\mathbf{z}}(\mathbf{x})] + b^2\mathrm{var}[h_{\mathbf{z}}(\boldsymbol{\alpha})] - 2b\,\mathrm{cov}[h_{\mathbf{z}}(\boldsymbol{\alpha})f_{\mathbf{z}}(\mathbf{x}), h_{\mathbf{z}}(\boldsymbol{\alpha})]. \quad (35)$$

The optimal value of the baseline $b$ is the one that minimizes the variance $V$, which is given by:

$$\frac{dV}{db} = 0 \quad (36)$$

$$\Leftrightarrow \quad 2b\,\mathrm{var}[h_{\mathbf{z}}(\boldsymbol{\alpha})] - 2\mathrm{cov}[h_{\mathbf{z}}(\boldsymbol{\alpha})f_{\mathbf{z}}(\mathbf{x}), h_{\mathbf{z}}(\boldsymbol{\alpha})] = 0 \quad (37)$$

$$\Leftrightarrow \quad b = \frac{\mathrm{cov}[h_{\mathbf{z}}(\boldsymbol{\alpha})f_{\mathbf{z}}(\mathbf{x}), h_{\mathbf{z}}(\boldsymbol{\alpha})]}{\mathrm{var}[h_{\mathbf{z}}(\boldsymbol{\alpha})]} \quad (38)$$

$$\Leftrightarrow \quad b = \frac{\mathbb{E}[h_{\mathbf{z}}^2(\boldsymbol{\alpha})f_{\mathbf{z}}(\mathbf{x})] - \mathbb{E}[h_{\mathbf{z}}(\boldsymbol{\alpha})f_{\mathbf{z}}(\mathbf{x})]\mathbb{E}[h_{\mathbf{z}}(\boldsymbol{\alpha})]}{\mathbb{E}[h_{\mathbf{z}}^2(\boldsymbol{\alpha})] - \mathbb{E}[h_{\mathbf{z}}(\boldsymbol{\alpha})]^2} \quad (39)$$

$$\Leftrightarrow \quad b = \frac{\mathbb{E}[h_{\mathbf{z}}^2(\boldsymbol{\alpha})f_{\mathbf{z}}(\mathbf{x})]}{\mathbb{E}[h_{\mathbf{z}}^2(\boldsymbol{\alpha})]}, \quad (40)$$

where we used the equality $\mathbb{E}[h_{\mathbf{z}}(\boldsymbol{\alpha})] = 0$ to obtain Eq. (40).

# D Implementation details

## D.1 Data pre-processing

Prior to training, we apply a one-hot encoding to the categorical features and all the features are then normalized to have zero mean and unit variance.

## D.2 PRS

In all our experiments, we use ensembles of $T = 100$ models and we initialize each $\alpha_j$ to 0.05, so that each feature is expected to be selected five times over the ensemble. We noticed that using lower initial $\alpha_j$ values prevents several features to be selected in the first iterations of the algorithm, hence resulting in convergence issues, while higher values result in larger computing times, as each base model must be trained using a larger number of features. The algorithm is run over 3,000 epochs with the Adam optimizer, and we select as optimal vector $\boldsymbol{\alpha}$ the one that yields the lowest value of the objective function on the validation set. For regression problems we use the mean square error as loss function, while for classification problems we use the cross-entropy. For the simulated, scikit-feature and DREAM4 datasets, the batch size is set to 10% of the samples of the training set, while the remaining 90% are used for training the base models. For the MNIST dataset, which is much larger, we use 50% of the samples as batch size and the remaining 50% for training. For the scikit-feature, MNIST and DREAM4 datasets, a grid-search strategy is used for tuning the value(s) of the regularization coefficient(s), by selecting the coefficient $\lambda$ (or the pair $(\lambda_1, \lambda_2)$) that minimizes the prediction error on the validation set. The tested values are the following:

- For scikit-feature: $\lambda = \{0.0001, 0.001, 0.01, 0.1\}$.

- For MNIST: $\lambda_1, \lambda_2 = \{0, 0.0001, 0.001, 0.01\}$.

- For DREAM4: $\lambda = \{0.001, 0.002, 0.003, 0.004, 0.005, 0.006, 0.007, 0.008, 0.009, 0.01\}$.

Regarding the predictions on a test set, the output of the PRS ensemble is computed as the average of the predictions of the different base models for regression problems, and as the majority class for classification problems.

### D.3 RS, RF and GBDT

Like for PRS, standard Random Subspace (RS), Random Forest (RF) and Gradient Boosting Decision Trees (GBDT) are all run with $T = 100$ models per ensemble. Given $M$ the total number of features, the following hyper-parameters are optimized on the validation set:

- For RS: the number $K$ of randomly sampled features for each base model. Tested values are $\{1, \frac{M}{100}, \frac{M}{50}, \frac{M}{20}, \frac{M}{10}, \frac{M}{5}, \frac{M}{3}, \frac{M}{2}, \sqrt{M}, M\}$.

- For RF: the number $K$ of randomly sampled features at each tree node. Tested values are $\{1, \frac{M}{100}, \frac{M}{50}, \frac{M}{20}, \frac{M}{10}, \frac{M}{5}, \frac{M}{3}, \frac{M}{2}, \sqrt{M}, M\}$.

- For GBDT: the maximum tree depth $d$. Tested values are $\{1, \ldots, 10\}$.

All the remaining hyper-parameters are set to the default values used in the scikit-learn library (Pedregosa et al., 2011). The feature rankings of RF and GBDT are computed using the standard Mean Decrease Impurity importance measure (Breiman et al., 1984a).

### D.4 RaSE

The RaSE approach (Tian & Feng, 2021) consists in iteratively sampling and evaluating a population of feature subsets. At each iteration, a probability distribution is identified from the best feature subsets and this distribution is used to sample new feature subsets. More specifically, given the current feature importances $\boldsymbol{\alpha}$, each iteration of RaSE consists of the following steps:

1. Sample $T \times B$ feature subsets: for $t$ from 1 to $T$ and for $b$ from 1 to $B$:
   - Sample the feature subset size $d$ from a uniform distribution $\mathcal{U}(1, D)$.
   - Sample the feature subset $\mathbf{z}^{t,b}$ of size $d$ from a multinomial distribution with parameters $d$ and $\tilde{\boldsymbol{\alpha}}$, where the selection probability $\tilde{\alpha}_j$ of the $j$-th feature is set as $\tilde{\alpha}_j = \alpha_j \mathbb{1}(\alpha_j > \frac{C_0}{\log M}) + \frac{C_0}{M}\mathbb{1}(\alpha_j \leq \frac{C_0}{\log M})$.

2. Evaluate each feature subset $\mathbf{z}^{t,b}$ by estimating, using 10-fold cross-validation, the prediction error of a model learned from this feature subset.

3. For $t$ from 1 to $T$, select the best subset $\mathbf{z}^{t,*}$ among $\{\mathbf{z}^{t,b}\}_{b=1}^{B}$, as the one with the lowest prediction error.

4. Set $\alpha_j$ as the fraction of these $T$ subsets where $z_j^{t,*} = 1$.

In all our experiments, we use $T = 100$, $B = 500$ and $C_0 = 0.1$. As done in (Tian & Feng, 2021), the maximum subset size is set as $D = \min(M, [\sqrt{N}])$, where $M$ is the number of features, $N$ is the number of samples in the training dataset and $[x]$ denotes the largest integer not larger than $x$. Like in PRS, each $\alpha_j$ is initialized to 0.05. We then run the algorithm over 10 iterations and we select as optimal vector $\boldsymbol{\alpha}$ the one that yields the lowest prediction error on the validation set. Note that the chosen number of iterations is very small because of the high computational complexity of RaSE (Tian & Feng, 2021 actually show results for at most 3 iterations).

# E    Simulated problems

## E.1    Simulation protocol

Table S6: **Simulated problems.** $M$ is the total number of features and $M_{rel}$ is the number of relevant ones.

| Problem | Type | $M$ | $M_{rel}$ |
|---|---|---|---|
| Checkerboard | Regression | 304 | 4 |
| Friedman | Regression | 305 | 5 |
| Hypercube | Classification | 305 | 5 |
| Linear | Classification | 310 | 10 |

We simulate four problems, where 300 irrelevant features are added to the relevant features. Let $M$ be the total number of features.

- *Checkerboard*: Checkerboard-like regression problem with strong correlation between features (Zhu et al., 2015). $\mathbf{x} \sim \mathcal{N}(\mathbf{0}_M, \Sigma_{M \times M})$, where $\Sigma_{i,j} = 0.9^{|i-j|}$. $y = 2x_1x_2 + 2x_3x_4 + \mathcal{N}(0,1)$.

- *Friedman*: Non-linear regression problem (Friedman, 1991). $y = 10sin(\pi x_1 x_2) + 20(x_3 - 0.5)^2 + 10x_4 + 5x_5 + 0.1\mathcal{N}(0,1)$. Like for the Checkerboard problem, we introduce a strong correlation between the features: $\mathbf{x} \sim \mathcal{N}(\mathbf{0}_M, \Sigma_{M \times M})$, where $\Sigma_{i,j} = s^2 0.9^{|i-j|}$. We use $s = \frac{0.5}{3}$, so that $\sim 99\%$ of the samples have values between 0 and 1.

- *Hypercube*: Non-linear, binary classification problem with 5 relevant features, generated with the *make_classification* function of the scikit-learn library (Pedregosa et al., 2011). In this problem, each class is associated with two vertices of a hypercube of dimension 5 and samples are generated in the neighbourhood of each vertex by using a normal distribution centred on the vertex (with $\Sigma = I$). Irrelevant features are each sampled from $\mathcal{N}(0,1)$.

- *Linear*: Linear, binary classification problem with 10 relevant features, generated by first simulating a linear regression problem with the *make_regression* function of the scikit-learn library and thresholding the output variable so that the two classes are balanced. The output before thresholding is: $y = \sum_{k=1}^{10} w_k x_k$, where $w_k \sim \mathcal{U}(0,100), k = 1, \dots, 10$ and $x_k \sim \mathcal{N}(0,1), k = 1, \dots, M$.

For each problem, we generate 10 datasets, each with 300 training samples, 100 validation samples and 100 test samples.

### E.2 Additional results

Table S7: **Performance of PRS with and without using the variance reduction technique.** The optimal value $b^*$ of the baseline is given by Eq. (12). Setting $b = 0$ amounts to removing the variance reduction method. We report here the prediction score on the test set ($R^2$ or accuracy), the feature ranking quality (AUPR) and the number of features used per base model (subspace size), i.e. the sum $\sum_{j=1}^{M} \alpha_j$. Values are mean and standard deviation over 10 datasets.

| | | tree | | kNN | | SVM | |
|---|---|---|---|---|---|---|---|
| | | $b = b^*$ | $b = 0$ | $b = b^*$ | $b = 0$ | $b = b^*$ | $b = 0$ |
| Checkerboard | $R^2$ | $0.29 \pm 0.14$ | $0.19 \pm 0.15$ | $0.60 \pm 0.06$ | $0.41 \pm 0.05$ | $0.62 \pm 0.07$ | $0.41 \pm 0.05$ |
| | AUPR | $0.60 \pm 0.23$ | $0.54 \pm 0.29$ | $0.92 \pm 0.14$ | $0.71 \pm 0.21$ | $0.98 \pm 0.06$ | $0.49 \pm 0.18$ |
| | Subspace size | $7.29 \pm 1.32$ | $13.71 \pm 1.61$ | $6.49 \pm 1.91$ | $31.49 \pm 2.87$ | $4.85 \pm 0.55$ | $24.73 \pm 2.16$ |
| Friedman | $R^2$ | $0.83 \pm 0.03$ | $0.77 \pm 0.04$ | $0.88 \pm 0.03$ | $0.72 \pm 0.03$ | $0.90 \pm 0.05$ | $0.78 \pm 0.05$ |
| | AUPR | $0.95 \pm 0.04$ | $0.81 \pm 0.07$ | $1.00 \pm 0.00$ | $0.71 \pm 0.08$ | $0.98 \pm 0.05$ | $0.86 \pm 0.09$ |
| | Subspace size | $7.82 \pm 0.75$ | $31.39 \pm 3.50$ | $5.49 \pm 0.47$ | $34.23 \pm 4.02$ | $7.28 \pm 1.31$ | $33.26 \pm 3.61$ |
| Hypercube | Accuracy | $0.88 \pm 0.04$ | $0.88 \pm 0.05$ | $0.90 \pm 0.06$ | $0.89 \pm 0.04$ | $0.88 \pm 0.05$ | $0.85 \pm 0.06$ |
| | AUPR | $0.97 \pm 0.06$ | $0.96 \pm 0.07$ | $0.94 \pm 0.09$ | $0.96 \pm 0.07$ | $0.90 \pm 0.14$ | $0.86 \pm 0.15$ |
| | Subspace size | $12.10 \pm 3.04$ | $22.90 \pm 2.75$ | $6.83 \pm 1.20$ | $20.14 \pm 2.36$ | $13.54 \pm 3.01$ | $23.49 \pm 5.07$ |
| Linear | Accuracy | $0.78 \pm 0.03$ | $0.78 \pm 0.04$ | $0.88 \pm 0.03$ | $0.88 \pm 0.03$ | $0.92 \pm 0.03$ | $0.93 \pm 0.03$ |
| | AUPR | $0.67 \pm 0.13$ | $0.69 \pm 0.12$ | $0.73 \pm 0.12$ | $0.73 \pm 0.11$ | $0.80 \pm 0.10$ | $0.82 \pm 0.11$ |
| | Subspace size | $12.68 \pm 2.95$ | $23.05 \pm 2.32$ | $10.12 \pm 2.30$ | $22.81 \pm 1.55$ | $19.97 \pm 1.90$ | $33.37 \pm 4.01$ |

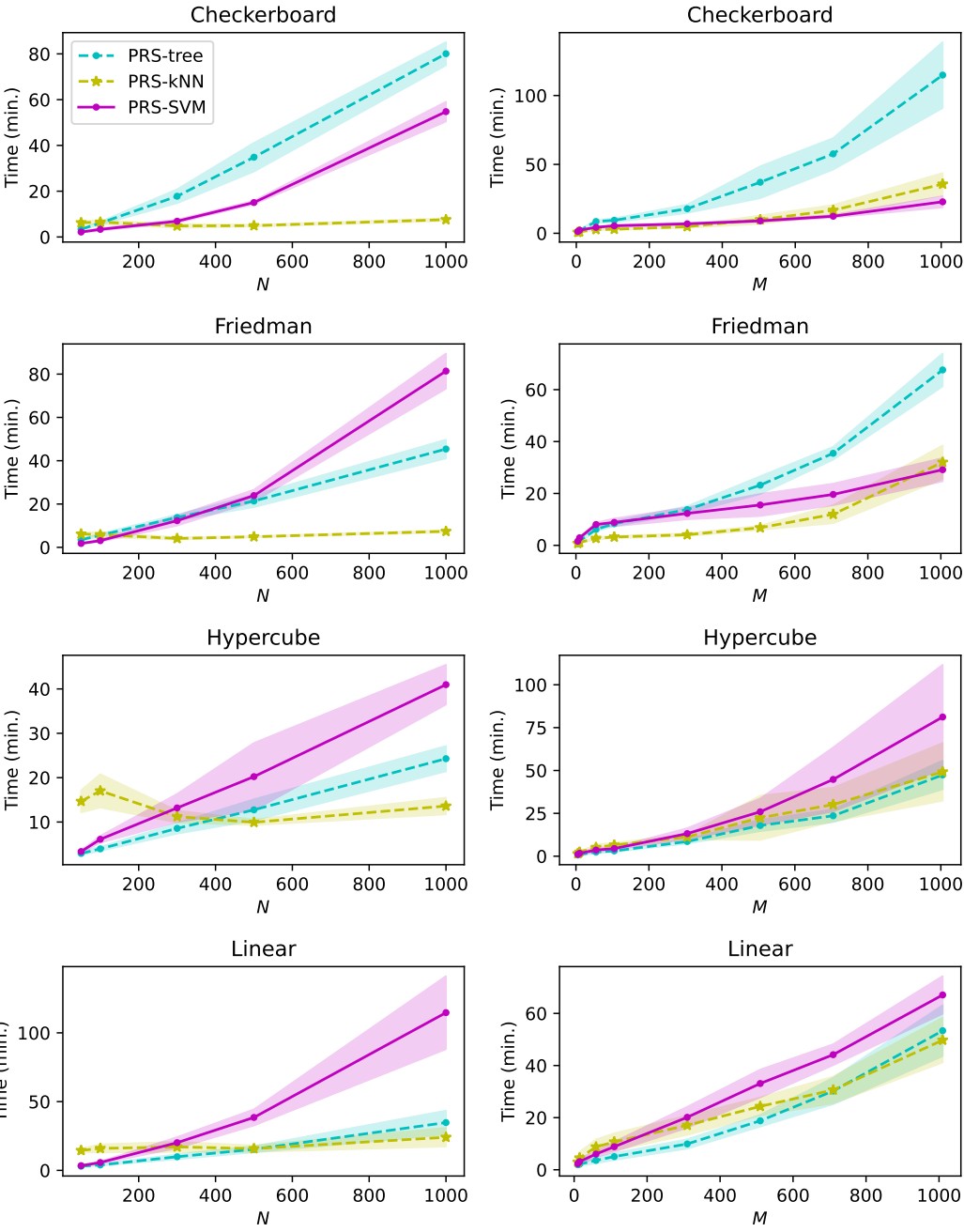

Figure S4: **Computing times of PRS (for training + testing).** Plain lines and shaded areas respectively indicate the mean and standard deviations over 10 datasets. The left-hand plots show the computing times for different training set sizes ($N$), with the number of irrelevant features set to 300. The right-hand plots show the computing times for different values of the number $M$ of features (we kept fixed the number of relevant features and increased the number of irrelevant features), with $N = 300$. The computing times were measured on AMD Epyc Rome CPUs at 2.9 GHz and 256GB of RAM.

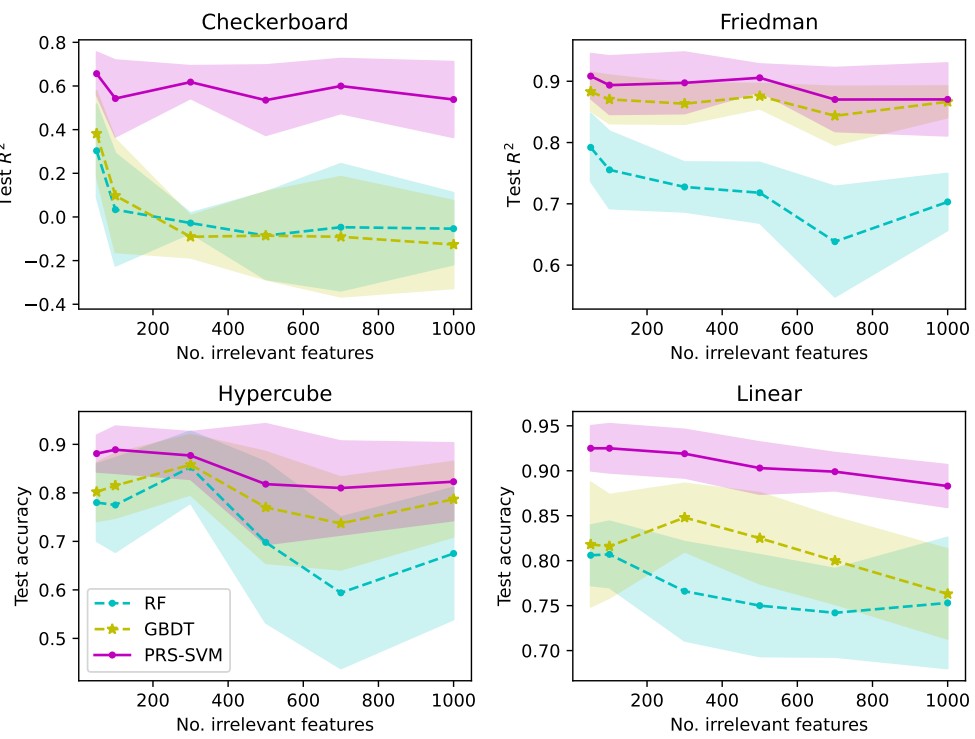

Figure S5: **Prediction score for an increasing number of irrelevant features.** Plain lines and shaded areas respectively indicate the mean and standard deviations over 10 datasets.

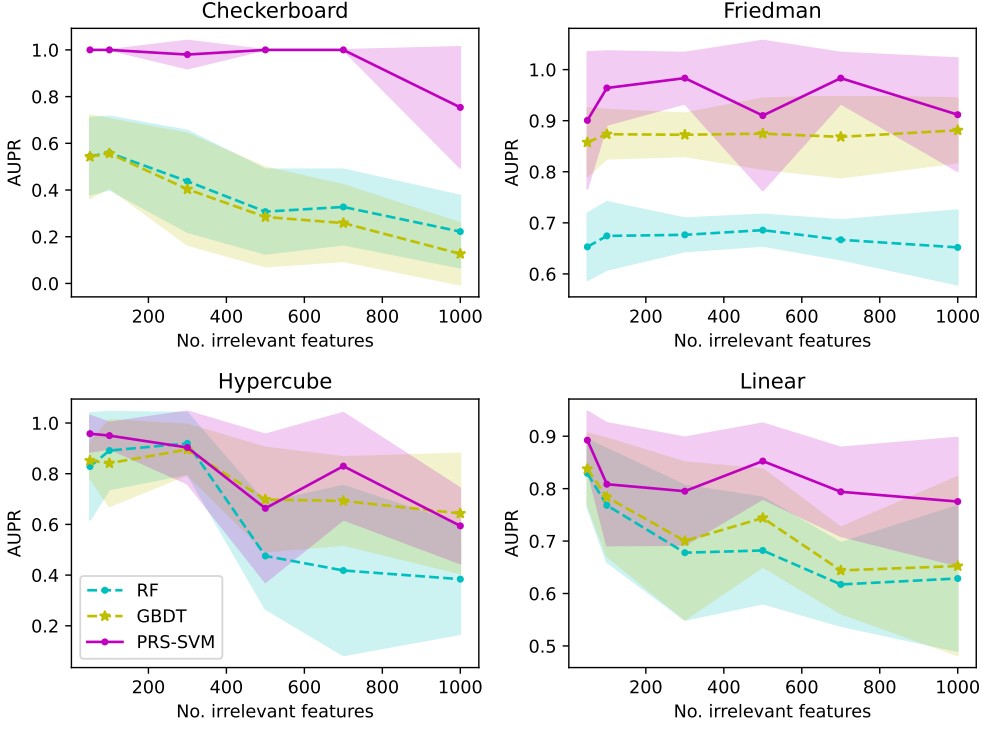

Figure S6: **Feature ranking AUPR for an increasing number of irrelevant features.** Plain lines and shaded areas respectively indicate the mean and standard deviations over 10 datasets.

Table S8: **Comparison of PRS and RaSE.** We report here the prediction score on the test set ($R^2$ or accuracy), the feature ranking quality (AUPR) and the number of features used per base model (subspace size), i.e. for PRS: the sum $\sum_{j=1}^{M} \alpha_j$, and for RaSE: the average feature subset size among the $T$ trained base models. Values are mean and standard deviation over 10 datasets.

| | | tree | | kNN | | SVM | |
| --- | --- | --- | --- | --- | --- | --- | --- |
| | | PRS | RaSE | PRS | RaSE | PRS | RaSE |
| Checkerboard | $R^2$ | $0.29 \pm 0.14$ | $0.25 \pm 0.15$ | $0.60 \pm 0.06$ | $0.61 \pm 0.05$ | $0.62 \pm 0.07$ | $0.62 \pm 0.07$ |
| | AUPR | $0.60 \pm 0.23$ | $0.77 \pm 0.24$ | $0.92 \pm 0.14$ | $1.00 \pm 0.00$ | $0.98 \pm 0.06$ | $1.00 \pm 0.00$ |
| | Subspace size | $7.29 \pm 1.32$ | $10.23 \pm 1.18$ | $6.49 \pm 1.91$ | $4.96 \pm 0.76$ | $4.85 \pm 0.55$ | $4.51 \pm 0.59$ |
| Friedman | $R^2$ | $0.83 \pm 0.03$ | $0.77 \pm 0.03$ | $0.88 \pm 0.03$ | $0.87 \pm 0.03$ | $0.90 \pm 0.05$ | $0.90 \pm 0.05$ |
| | AUPR | $0.95 \pm 0.04$ | $0.79 \pm 0.14$ | $1.00 \pm 0.00$ | $0.97 \pm 0.06$ | $0.98 \pm 0.05$ | $0.98 \pm 0.05$ |
| | Subspace size | $7.82 \pm 0.75$ | $10.40 \pm 2.87$ | $5.49 \pm 0.47$ | $5.67 \pm 0.51$ | $7.28 \pm 1.31$ | $6.51 \pm 0.97$ |
| Hypercube | Accuracy | $0.88 \pm 0.04$ | $0.86 \pm 0.07$ | $0.90 \pm 0.06$ | $0.92 \pm 0.04$ | $0.88 \pm 0.05$ | $0.90 \pm 0.05$ |
| | AUPR | $0.97 \pm 0.06$ | $0.89 \pm 0.13$ | $0.94 \pm 0.09$ | $0.94 \pm 0.09$ | $0.90 \pm 0.14$ | $0.92 \pm 0.12$ |
| | Subspace size | $12.10 \pm 3.04$ | $12.43 \pm 0.83$ | $6.83 \pm 1.20$ | $5.28 \pm 0.66$ | $13.54 \pm 3.01$ | $10.23 \pm 2.32$ |
| Linear | Accuracy | $0.78 \pm 0.03$ | $0.77 \pm 0.05$ | $0.88 \pm 0.03$ | $0.88 \pm 0.04$ | $0.92 \pm 0.03$ | $0.91 \pm 0.02$ |
| | AUPR | $0.67 \pm 0.13$ | $0.54 \pm 0.12$ | $0.73 \pm 0.12$ | $0.67 \pm 0.12$ | $0.80 \pm 0.10$ | $0.75 \pm 0.10$ |
| | Subspace size | $12.68 \pm 2.95$ | $12.66 \pm 0.42$ | $10.12 \pm 2.30$ | $10.85 \pm 1.92$ | $19.97 \pm 1.90$ | $15.06 \pm 0.47$ |

Table S9: **Number of trained base models during the PRS training (with 3000 epochs).** Values are mean and standard deviation over 10 datasets.

| | tree | kNN | SVM |
| --- | --- | --- | --- |
| Checkerboard | $308700 \pm 20095$ | $119700 \pm 24661$ | $76000 \pm 7238$ |
| Friedman | $323200 \pm 27282$ | $107000 \pm 18066$ | $130800 \pm 23949$ |
| Hypercube | $162600 \pm 24381$ | $112700 \pm 13535$ | $170900 \pm 41234$ |
| Linear | $214400 \pm 70949$ | $172300 \pm 37534$ | $248700 \pm 42159$ |

Table S10: **Performance of PRS-SVM for varying thresholds on $T_{eff}$.** The threshold $T$ corresponds to the case where new base models are trained at each epoch (no importance sampling). Values are mean and standard deviation over 10 datasets.

| Problem | $T_{eff}$ threshold | No. trained models | Training time (min.) | $R^2$/Accuracy | AUPR |
|---|---|---|---|---|---|
| Checkerboard | $T$ | $3001000 \pm 0$ | $223.17 \pm 2.98$ | $0.62 \pm 0.07$ | $0.98 \pm 0.06$ |
| | $0.9T$ | $76000 \pm 7238$ | $6.89 \pm 0.63$ | $0.62 \pm 0.07$ | $0.98 \pm 0.06$ |
| | $0.7T$ | $42300 \pm 2193$ | $4.24 \pm 0.18$ | $0.62 \pm 0.07$ | $0.98 \pm 0.06$ |
| | $0.5T$ | $31800 \pm 1989$ | $3.49 \pm 0.26$ | $0.62 \pm 0.08$ | $0.96 \pm 0.08$ |
| | $0.3T$ | $23400 \pm 1200$ | $2.88 \pm 0.14$ | $0.61 \pm 0.07$ | $0.96 \pm 0.08$ |
| Friedman | $T$ | $3001000 \pm 0$ | $260.85 \pm 9.57$ | $0.90 \pm 0.05$ | $0.98 \pm 0.05$ |
| | $0.9T$ | $130800 \pm 23949$ | $12.32 \pm 2.25$ | $0.90 \pm 0.05$ | $0.98 \pm 0.05$ |
| | $0.7T$ | $72400 \pm 8236$ | $7.57 \pm 1.01$ | $0.90 \pm 0.05$ | $0.98 \pm 0.05$ |
| | $0.5T$ | $53200 \pm 6720$ | $5.87 \pm 0.76$ | $0.90 \pm 0.05$ | $0.95 \pm 0.09$ |
| | $0.3T$ | $43000 \pm 6066$ | $5.15 \pm 0.73$ | $0.89 \pm 0.05$ | $0.89 \pm 0.14$ |
| Hypercube | $T$ | $3001000 \pm 0$ | $210.56 \pm 41.64$ | $0.88 \pm 0.05$ | $0.92 \pm 0.13$ |
| | $0.9T$ | $170900 \pm 41234$ | $13.17 \pm 3.15$ | $0.88 \pm 0.05$ | $0.90 \pm 0.14$ |
| | $0.7T$ | $66400 \pm 20967$ | $9.16 \pm 1.60$ | $0.87 \pm 0.05$ | $0.83 \pm 0.14$ |
| | $0.5T$ | $42200 \pm 11989$ | $9.12 \pm 1.19$ | $0.86 \pm 0.05$ | $0.76 \pm 0.26$ |
| | $0.3T$ | $18500 \pm 4944$ | $9.36 \pm 1.82$ | $0.83 \pm 0.09$ | $0.59 \pm 0.24$ |
| Linear | $T$ | $3001000 \pm 0$ | $230.52 \pm 23.52$ | $0.92 \pm 0.02$ | $0.80 \pm 0.11$ |
| | $0.9T$ | $248700 \pm 42159$ | $20.14 \pm 4.01$ | $0.92 \pm 0.03$ | $0.80 \pm 0.10$ |
| | $0.7T$ | $108600 \pm 34325$ | $12.07 \pm 1.42$ | $0.92 \pm 0.04$ | $0.76 \pm 0.08$ |
| | $0.5T$ | $65900 \pm 27351$ | $10.64 \pm 1.31$ | $0.90 \pm 0.03$ | $0.74 \pm 0.11$ |
| | $0.3T$ | $21600 \pm 4247$ | $11.85 \pm 2.26$ | $0.84 \pm 0.06$ | $0.54 \pm 0.20$ |

## F  Real-world datasets

Table S11: **Sizes of regression datasets.** For datasets with categorical features, the last column indicates the total number of features after one-hot encoding.

| tabular benchmark, regression, with only numerical features | | |
|---|---|---|
| Dataset | Samples | Features |
| cpu_act | 8192 | 21 |
| pol | 15000 | 26 |
| elevators | 16599 | 16 |
| isolet | 7797 | 613 |
| wine_quality | 6497 | 11 |
| Ailerons | 13750 | 33 |
| houses | 20640 | 8 |
| house_16H | 22784 | 16 |
| diamonds | 53940 | 6 |
| Brazilian_houses | 10692 | 8 |
| Bike_Sharing_Demand | 17379 | 6 |
| nyc-taxi-green-dec-2016 | 581835 | 9 |
| house_sales | 21613 | 15 |
| sulfur | 10081 | 6 |
| medical_charges | 163065 | 3 |
| MiamiHousing2016 | 13932 | 13 |
| superconduct | 21263 | 79 |
| california | 20640 | 8 |
| fifa | 18063 | 5 |
| year | 515345 | 90 |
| **tabular benchmark, regression, with both numerical and categorical features** | | |
| Dataset | Samples | Features |
| yprop_4_1 | 8885 | 82 |
| analcatdata_supreme | 4052 | 12 |
| visualizing_soil | 8641 | 5 |
| black_friday | 166821 | 23 |
| diamonds | 53940 | 26 |
| Mercedes_Benz_Greener_Manufacturing | 4209 | 735 |
| Brazilian_houses | 10692 | 17 |
| Bike_Sharing_Demand | 17379 | 20 |
| OnlineNewsPopularity | 39644 | 73 |
| nyc-taxi-green-dec-2016 | 581835 | 31 |
| house_sales | 21613 | 19 |
| particulate-matter-ukair-2017 | 394299 | 26 |
| SGEMM_GPU_kernel_performance | 241600 | 15 |

Table S12: **Sizes of classification datasets.** For datasets with categorical features, the last column indicates the total number of features after one-hot encoding.

| tabular benchmark, classification, with only numerical features | | | |
|---|---|---|---|
| Dataset | Classes | Samples | Features |
| credit | 2 | 16714 (8357 / 8357) | 10 |
| california | 2 | 20634 (10317 / 10317) | 8 |
| wine | 2 | 2554 (1277 / 1277) | 11 |
| electricity | 2 | 38474 (19237 / 19237) | 7 |
| covertype | 2 | 566602 (283301 / 283301) | 10 |
| pol | 2 | 10082 (5041 / 5041) | 26 |
| house_16H | 2 | 13488 (6744 / 6744) | 16 |
| kdd_ipums_la_97-small | 2 | 5188 (2594 / 2594) | 20 |
| MagicTelescope | 2 | 13376 (6688 / 6688) | 10 |
| bank-marketing | 2 | 10578 (5289 / 5289) | 7 |
| phoneme | 2 | 3172 (1586 / 1586) | 5 |
| MiniBooNE | 2 | 72998 (36499 / 36499) | 50 |
| Higgs | 2 | 940160 (470080 / 470080) | 24 |
| eye_movements | 2 | 7608 (3804 / 3804) | 20 |
| jannis | 2 | 57580 (28790 / 28790) | 54 |
| **tabular benchmark, classification, with both numerical and categorical features** | | | |
| Dataset | Classes | Samples | Features |
| electricity | 2 | 38474 (19237 / 19237) | 14 |
| eye_movements | 2 | 7608 (3804 / 3804) | 26 |
| covertype | 2 | 423680 (211840 / 211840) | 93 |
| rl | 2 | 4970 (2485 / 2485) | 38 |
| road-safety | 2 | 111762 (55881 / 55881) | 35 |
| compass | 2 | 16644 (8322 / 8322) | 59 |
| KDDCup09_upselling | 2 | 5128 (2564 / 2564) | 104 |
| **scikit-feature benchmark, classification, with only numerical features** | | | |
| Dataset | Classes | Samples | Features |
| arcene | 2 | 200 (112 / 88) | 10000 |
| CLL_SUB_111 | 2 | 100 (49 / 51) | 11340 |
| Prostate_GE | 2 | 102 (50 / 52) | 5966 |
| SMK_CAN_187 | 2 | 187 (90 / 97) | 19993 |
| TOX_171 | 4 | 171 (45 / 45 / 39 / 42) | 5748 |

Table S13: **Prediction performance of single models, RS, PRS, and RaSE, when the base model is the decision tree.** Values are $R^2$ scores for regression problems and accuracies for classification problems, averaged over 10 random data subsamplings for the tabular datasets and over 5 cross-validation folds for the scikit-feature datasets. For each dataset, the highest performance is indicated in bold type.

| | Single tree | RS-tree | PRS-tree | RaSE-tree |
|---|---|---|---|---|
| **tabular benchmark, regression, with only numerical features** | | | | |
| cpu_act | 0.95 ± 0.00 | 0.96 ± 0.00 | **0.98 ± 0.00** | 0.97 ± 0.00 |
| pol | 0.89 ± 0.02 | 0.91 ± 0.02 | **0.94 ± 0.01** | 0.94 ± 0.01 |
| elevators | 0.38 ± 0.05 | 0.60 ± 0.04 | **0.71 ± 0.02** | 0.70 ± 0.02 |
| isolet | 0.32 ± 0.04 | 0.74 ± 0.02 | **0.76 ± 0.02** | 0.75 ± 0.02 |
| wine_quality | -0.27 ± 0.11 | **0.35 ± 0.02** | 0.34 ± 0.02 | 0.16 ± 0.02 |
| Ailerons | 0.60 ± 0.04 | 0.71 ± 0.02 | **0.78 ± 0.02** | 0.73 ± 0.03 |
| houses | 0.50 ± 0.07 | 0.68 ± 0.02 | **0.74 ± 0.03** | 0.67 ± 0.02 |
| house_16H | 0.05 ± 0.30 | **0.48 ± 0.10** | 0.46 ± 0.07 | 0.41 ± 0.18 |
| diamonds | 0.89 ± 0.01 | 0.93 ± 0.00 | **0.93 ± 0.01** | 0.93 ± 0.00 |
| Brazilian_houses | 0.97 ± 0.02 | 0.98 ± 0.02 | **0.98 ± 0.02** | 0.98 ± 0.02 |
| Bike_Sharing_Demand | 0.33 ± 0.05 | 0.51 ± 0.02 | **0.59 ± 0.02** | 0.50 ± 0.04 |
| nyc-taxi-green-dec-2016 | -0.10 ± 0.14 | 0.29 ± 0.04 | **0.44 ± 0.05** | 0.39 ± 0.08 |
| house_sales | 0.64 ± 0.02 | 0.79 ± 0.01 | **0.83 ± 0.01** | 0.77 ± 0.02 |
| sulfur | 0.56 ± 0.13 | 0.64 ± 0.08 | **0.68 ± 0.08** | 0.67 ± 0.09 |
| medical_charges | 0.96 ± 0.01 | 0.96 ± 0.00 | **0.97 ± 0.00** | 0.97 ± 0.01 |
| MiamiHousing2016 | 0.70 ± 0.03 | 0.85 ± 0.01 | **0.87 ± 0.01** | 0.83 ± 0.01 |
| superconduct | 0.69 ± 0.03 | 0.84 ± 0.01 | **0.84 ± 0.01** | 0.84 ± 0.01 |
| california | 0.48 ± 0.07 | 0.71 ± 0.03 | **0.78 ± 0.02** | 0.71 ± 0.04 |
| fifa | 0.29 ± 0.06 | 0.44 ± 0.03 | **0.56 ± 0.03** | 0.43 ± 0.05 |
| year | -0.73 ± 0.09 | 0.13 ± 0.03 | 0.14 ± 0.04 | **0.15 ± 0.02** |
| **tabular benchmark, regression, with both numerical and categorical features** | | | | |
| yprop_4_1 | -0.90 ± 0.18 | **0.03 ± 0.02** | 0.02 ± 0.01 | 0.02 ± 0.01 |
| analcatdata_supreme | 0.96 ± 0.01 | 0.97 ± 0.01 | **0.98 ± 0.01** | 0.98 ± 0.01 |
| visualizing_soil | 1.00 ± 0.00 | 1.00 ± 0.00 | 1.00 ± 0.00 | **1.00 ± 0.00** |
| black_friday | 0.12 ± 0.09 | 0.44 ± 0.04 | **0.57 ± 0.03** | 0.56 ± 0.03 |
| diamonds | 0.95 ± 0.01 | 0.96 ± 0.00 | 0.97 ± 0.00 | **0.98 ± 0.00** |
| Mercedes_Benz_Greener_Manufacturing | 0.10 ± 0.13 | 0.44 ± 0.07 | 0.55 ± 0.05 | **0.56 ± 0.05** |
| Brazilian_houses | 0.97 ± 0.02 | 0.98 ± 0.02 | **0.98 ± 0.02** | 0.98 ± 0.02 |
| Bike_Sharing_Demand | 0.68 ± 0.05 | 0.73 ± 0.03 | **0.88 ± 0.02** | 0.86 ± 0.02 |
| OnlineNewsPopularity | -0.87 ± 0.08 | **0.08 ± 0.02** | 0.08 ± 0.03 | 0.04 ± 0.03 |
| nyc-taxi-green-dec-2016 | -0.10 ± 0.15 | 0.31 ± 0.04 | **0.45 ± 0.05** | 0.41 ± 0.08 |
| house_sales | 0.65 ± 0.02 | 0.79 ± 0.02 | **0.84 ± 0.01** | 0.78 ± 0.03 |
| particulate-matter-ukair-2017 | 0.28 ± 0.06 | 0.49 ± 0.02 | **0.56 ± 0.02** | 0.49 ± 0.03 |
| SGEMM_GPU_kernel_performance | 1.00 ± 0.00 | 1.00 ± 0.00 | **1.00 ± 0.00** | 1.00 ± 0.00 |
| **tabular benchmark, classification, with only numerical features** | | | | |
| credit | 0.69 ± 0.01 | 0.74 ± 0.02 | **0.75 ± 0.01** | 0.74 ± 0.01 |
| california | 0.78 ± 0.02 | 0.85 ± 0.01 | **0.86 ± 0.01** | 0.85 ± 0.02 |
| wine | 0.72 ± 0.02 | 0.78 ± 0.02 | **0.78 ± 0.02** | 0.75 ± 0.02 |
| electricity | 0.73 ± 0.02 | 0.77 ± 0.01 | **0.78 ± 0.01** | 0.77 ± 0.01 |
| covertype | 0.69 ± 0.02 | 0.72 ± 0.02 | **0.74 ± 0.02** | 0.73 ± 0.02 |
| pol | 0.95 ± 0.01 | 0.96 ± 0.01 | **0.96 ± 0.01** | 0.96 ± 0.01 |
| house_16H | 0.78 ± 0.01 | **0.86 ± 0.01** | 0.85 ± 0.02 | 0.82 ± 0.02 |
| kdd_ipums_la_97-small | 0.84 ± 0.01 | 0.87 ± 0.01 | 0.87 ± 0.01 | **0.87 ± 0.01** |
| MagicTelescope | 0.76 ± 0.02 | **0.81 ± 0.01** | 0.80 ± 0.01 | 0.79 ± 0.02 |
| bank-marketing | 0.70 ± 0.02 | 0.74 ± 0.01 | **0.75 ± 0.02** | 0.72 ± 0.02 |
| phoneme | 0.80 ± 0.02 | 0.80 ± 0.01 | **0.84 ± 0.01** | 0.82 ± 0.02 |
| MiniBooNE | 0.82 ± 0.01 | 0.90 ± 0.01 | **0.90 ± 0.01** | 0.89 ± 0.01 |
| Higgs | 0.58 ± 0.02 | 0.65 ± 0.01 | **0.65 ± 0.02** | 0.62 ± 0.02 |
| eye_movements | 0.53 ± 0.02 | 0.57 ± 0.01 | 0.63 ± 0.02 | **0.66 ± 0.02** |
| jannis | 0.65 ± 0.02 | 0.73 ± 0.02 | **0.73 ± 0.02** | 0.72 ± 0.01 |
| **tabular benchmark, classification, with both numerical and categorical features** | | | | |
| electricity | 0.73 ± 0.02 | 0.78 ± 0.01 | 0.78 ± 0.01 | **0.78 ± 0.02** |
| eye_movements | 0.52 ± 0.02 | 0.57 ± 0.01 | 0.63 ± 0.01 | **0.66 ± 0.02** |
| covertype | 0.70 ± 0.02 | **0.76 ± 0.01** | 0.75 ± 0.02 | 0.75 ± 0.02 |
| rl | 0.64 ± 0.02 | 0.71 ± 0.01 | **0.77 ± 0.01** | 0.76 ± 0.02 |
| road-safety | 0.65 ± 0.01 | 0.71 ± 0.02 | **0.71 ± 0.02** | 0.71 ± 0.01 |
| compass | 0.60 ± 0.02 | **0.67 ± 0.02** | 0.67 ± 0.02 | 0.66 ± 0.02 |
| KDDCup09_upselling | 0.75 ± 0.01 | 0.78 ± 0.01 | 0.79 ± 0.01 | **0.79 ± 0.01** |
| **scikit-feature benchmark, classification, with only numerical features** | | | | |
| arcene | 0.67 ± 0.05 | 0.80 ± 0.09 | 0.81 ± 0.05 | **0.82 ± 0.06** |
| CLL_SUB_111 | 0.62 ± 0.10 | **0.73 ± 0.08** | 0.70 ± 0.09 | 0.59 ± 0.11 |
| Prostate_GE | 0.75 ± 0.09 | 0.88 ± 0.08 | **0.88 ± 0.11** | 0.87 ± 0.08 |
| SMK_CAN_187 | 0.57 ± 0.08 | 0.66 ± 0.06 | 0.64 ± 0.07 | **0.66 ± 0.06** |
| TOX_171 | 0.56 ± 0.13 | **0.80 ± 0.06** | 0.75 ± 0.05 | 0.76 ± 0.08 |

Table S14: **Prediction performance of single models, RS, PRS, and RaSE, when the base model is the kNN.** Values are $R^2$ scores for regression problems and accuracies for classification problems, averaged over 10 random data subsamplings for the tabular datasets and over 5 cross-validation folds for the scikit-feature datasets. For each dataset, the highest performance is indicated in bold type.

| | Single kNN | RS-kNN | PRS-kNN | RaSE-kNN |
|---|---|---|---|---|
| **tabular benchmark, regression, with only numerical features** | | | | |
| cpu_act | 0.87 ± 0.03 | 0.87 ± 0.03 | **0.97 ± 0.00** | 0.96 ± 0.01 |
| pol | 0.87 ± 0.02 | 0.87 ± 0.02 | **0.94 ± 0.01** | 0.94 ± 0.01 |
| elevators | 0.54 ± 0.03 | 0.55 ± 0.03 | **0.73 ± 0.02** | 0.73 ± 0.02 |
| isolet | 0.68 ± 0.03 | 0.70 ± 0.03 | **0.85 ± 0.02** | 0.82 ± 0.02 |
| wine_quality | 0.25 ± 0.04 | 0.34 ± 0.02 | **0.34 ± 0.02** | 0.32 ± 0.04 |
| Ailerons | 0.64 ± 0.03 | 0.65 ± 0.03 | **0.80 ± 0.01** | 0.80 ± 0.01 |
| houses | 0.63 ± 0.02 | 0.63 ± 0.02 | **0.76 ± 0.02** | 0.73 ± 0.03 |
| house_16H | 0.41 ± 0.09 | 0.43 ± 0.10 | 0.42 ± 0.10 | **0.44 ± 0.10** |
| diamonds | 0.93 ± 0.01 | **0.94 ± 0.00** | 0.94 ± 0.01 | 0.93 ± 0.00 |
| Brazilian_houses | 0.95 ± 0.01 | 0.95 ± 0.01 | **0.98 ± 0.01** | 0.98 ± 0.01 |
| Bike_Sharing_Demand | 0.44 ± 0.05 | 0.51 ± 0.02 | **0.63 ± 0.02** | 0.59 ± 0.04 |
| nyc-taxi-green-dec-2016 | 0.21 ± 0.03 | 0.24 ± 0.02 | **0.46 ± 0.03** | 0.44 ± 0.03 |
| house_sales | 0.73 ± 0.01 | 0.75 ± 0.01 | **0.83 ± 0.01** | 0.83 ± 0.01 |
| sulfur | 0.59 ± 0.10 | 0.59 ± 0.10 | **0.69 ± 0.10** | 0.68 ± 0.12 |
| medical_charges | 0.96 ± 0.01 | 0.96 ± 0.01 | **0.98 ± 0.00** | 0.97 ± 0.00 |
| MiamiHousing2016 | 0.80 ± 0.01 | 0.81 ± 0.01 | **0.86 ± 0.01** | 0.85 ± 0.01 |
| superconduct | 0.77 ± 0.01 | 0.80 ± 0.01 | **0.82 ± 0.01** | 0.81 ± 0.01 |
| california | 0.67 ± 0.02 | 0.67 ± 0.02 | **0.77 ± 0.02** | 0.73 ± 0.03 |
| fifa | 0.36 ± 0.02 | 0.45 ± 0.02 | **0.61 ± 0.02** | 0.58 ± 0.02 |
| year | 0.03 ± 0.04 | 0.13 ± 0.02 | 0.18 ± 0.02 | **0.19 ± 0.02** |
| **tabular benchmark, regression, with both numerical and categorical features** | | | | |
| yprop_4_1 | -0.10 ± 0.04 | 0.04 ± 0.01 | **0.05 ± 0.01** | 0.05 ± 0.01 |
| analcatdata_supreme | 0.90 ± 0.02 | 0.90 ± 0.02 | **0.98 ± 0.01** | 0.98 ± 0.01 |
| visualizing_soil | 0.99 ± 0.00 | 0.99 ± 0.00 | **1.00 ± 0.00** | 1.00 ± 0.00 |
| black_friday | -0.04 ± 0.04 | 0.21 ± 0.02 | **0.55 ± 0.03** | 0.55 ± 0.03 |
| diamonds | 0.77 ± 0.01 | 0.91 ± 0.01 | 0.96 ± 0.01 | **0.97 ± 0.00** |
| Mercedes_Benz_Greener_Manufacturing | 0.32 ± 0.05 | 0.41 ± 0.05 | 0.54 ± 0.05 | **0.55 ± 0.05** |
| Brazilian_houses | 0.89 ± 0.02 | 0.93 ± 0.01 | **0.98 ± 0.01** | 0.98 ± 0.01 |
| Bike_Sharing_Demand | 0.39 ± 0.04 | 0.54 ± 0.03 | 0.86 ± 0.02 | **0.86 ± 0.01** |
| OnlineNewsPopularity | -0.10 ± 0.02 | 0.07 ± 0.02 | **0.10 ± 0.03** | 0.07 ± 0.03 |
| nyc-taxi-green-dec-2016 | 0.16 ± 0.03 | 0.24 ± 0.03 | **0.48 ± 0.03** | 0.47 ± 0.03 |
| house_sales | 0.72 ± 0.01 | 0.73 ± 0.02 | **0.84 ± 0.00** | 0.83 ± 0.00 |
| particulate-matter-ukair-2017 | 0.17 ± 0.04 | 0.39 ± 0.03 | 0.62 ± 0.02 | **0.62 ± 0.01** |
| SGEMM_GPU_kernel_performance | 0.87 ± 0.01 | 0.97 ± 0.01 | **1.00 ± 0.00** | 1.00 ± 0.00 |
| **tabular benchmark, classification, with only numerical features** | | | | |
| credit | 0.62 ± 0.04 | 0.75 ± 0.02 | **0.76 ± 0.01** | 0.75 ± 0.01 |
| california | 0.81 ± 0.01 | 0.83 ± 0.01 | **0.86 ± 0.01** | 0.84 ± 0.02 |
| wine | 0.73 ± 0.02 | 0.77 ± 0.02 | **0.77 ± 0.02** | 0.75 ± 0.02 |
| electricity | 0.73 ± 0.01 | 0.75 ± 0.02 | **0.77 ± 0.01** | 0.76 ± 0.01 |
| covertype | 0.70 ± 0.02 | 0.72 ± 0.02 | **0.76 ± 0.02** | 0.76 ± 0.02 |
| pol | 0.91 ± 0.01 | 0.94 ± 0.01 | **0.95 ± 0.01** | 0.95 ± 0.01 |
| house_16H | 0.81 ± 0.01 | 0.84 ± 0.01 | **0.85 ± 0.01** | 0.83 ± 0.01 |
| kdd_ipums_la_97-small | 0.81 ± 0.01 | 0.85 ± 0.01 | 0.87 ± 0.01 | **0.88 ± 0.01** |
| MagicTelescope | 0.78 ± 0.01 | 0.79 ± 0.01 | **0.81 ± 0.02** | 0.80 ± 0.02 |
| bank-marketing | 0.75 ± 0.01 | 0.77 ± 0.01 | **0.78 ± 0.01** | 0.76 ± 0.02 |
| phoneme | 0.83 ± 0.01 | 0.83 ± 0.01 | **0.84 ± 0.01** | 0.83 ± 0.01 |
| MiniBooNE | 0.84 ± 0.01 | 0.88 ± 0.01 | **0.89 ± 0.01** | 0.88 ± 0.02 |
| Higgs | 0.55 ± 0.01 | 0.62 ± 0.02 | **0.67 ± 0.01** | 0.66 ± 0.01 |
| eye_movements | 0.53 ± 0.02 | 0.55 ± 0.02 | **0.57 ± 0.02** | 0.55 ± 0.02 |
| jannis | 0.66 ± 0.02 | 0.70 ± 0.02 | **0.74 ± 0.01** | 0.73 ± 0.01 |
| **tabular benchmark, classification, with both numerical and categorical features** | | | | |
| electricity | 0.71 ± 0.02 | 0.76 ± 0.02 | **0.77 ± 0.02** | 0.77 ± 0.02 |
| eye_movements | 0.53 ± 0.02 | 0.56 ± 0.02 | **0.57 ± 0.02** | 0.55 ± 0.03 |
| covertype | 0.72 ± 0.02 | 0.74 ± 0.02 | **0.78 ± 0.02** | 0.78 ± 0.02 |
| rl | 0.60 ± 0.01 | 0.65 ± 0.02 | **0.73 ± 0.02** | 0.72 ± 0.02 |
| road-safety | 0.66 ± 0.01 | 0.70 ± 0.01 | **0.72 ± 0.01** | 0.71 ± 0.01 |
| compass | 0.60 ± 0.02 | 0.66 ± 0.01 | **0.68 ± 0.02** | 0.68 ± 0.02 |
| KDDCup09_upselling | 0.63 ± 0.02 | 0.68 ± 0.02 | 0.77 ± 0.01 | **0.79 ± 0.01** |
| **scikit-feature benchmark, classification, with only numerical features** | | | | |
| arcene | 0.80 ± 0.04 | 0.77 ± 0.02 | 0.81 ± 0.05 | **0.82 ± 0.06** |
| CLL_SUB_111 | 0.50 ± 0.09 | 0.48 ± 0.12 | 0.55 ± 0.16 | **0.72 ± 0.10** |
| Prostate_GE | 0.78 ± 0.04 | 0.79 ± 0.04 | 0.89 ± 0.07 | **0.91 ± 0.06** |
| SMK_CAN_187 | 0.63 ± 0.05 | 0.63 ± 0.05 | **0.66 ± 0.04** | 0.64 ± 0.09 |
| TOX_171 | 0.68 ± 0.12 | 0.71 ± 0.09 | **0.88 ± 0.06** | 0.79 ± 0.05 |

Table S15: **Prediction performance of single models, RS, PRS, and RaSE, when the base model is the SVM.** Values are $R^2$ scores for regression problems and accuracies for classification problems, averaged over 10 random data subsamplings for the tabular datasets and over 5 cross-validation folds for the scikit-feature datasets. For each dataset, the highest performance is indicated in bold type.

| | Single SVM | RS-SVM | PRS-SVM | RaSE-SVM |
|---|---|---|---|---|
| **tabular benchmark, regression, with only numerical features** | | | | |
| cpu_act | 0.38 ± 0.03 | 0.38 ± 0.03 | **0.74 ± 0.04** | 0.73 ± 0.04 |
| pol | 0.41 ± 0.09 | 0.41 ± 0.09 | 0.45 ± 0.21 | **0.76 ± 0.03** |
| elevators | -6.97 ± 1.86 | -6.97 ± 1.86 | **-0.00 ± 0.00** | -6.97 ± 1.86 |
| isolet | 0.47 ± 0.03 | 0.47 ± 0.03 | **0.71 ± 0.01** | 0.70 ± 0.02 |
| wine_quality | 0.34 ± 0.02 | 0.34 ± 0.02 | **0.35 ± 0.02** | 0.34 ± 0.02 |
| Ailerons | -4.33 ± 1.97 | -4.33 ± 1.97 | **-0.00 ± 0.00** | -4.33 ± 1.97 |
| houses | 0.73 ± 0.02 | 0.73 ± 0.02 | 0.74 ± 0.02 | **0.74 ± 0.03** |
| house_16H | 0.46 ± 0.12 | 0.46 ± 0.12 | **0.46 ± 0.11** | 0.46 ± 0.11 |
| diamonds | 0.94 ± 0.00 | 0.94 ± 0.00 | **0.94 ± 0.00** | 0.94 ± 0.00 |
| Brazilian_houses | 0.96 ± 0.01 | 0.96 ± 0.01 | **0.97 ± 0.01** | 0.97 ± 0.01 |
| Bike_Sharing_Demand | 0.16 ± 0.03 | 0.16 ± 0.03 | 0.24 ± 0.04 | **0.24 ± 0.03** |
| nyc-taxi-green-dec-2016 | 0.34 ± 0.04 | 0.34 ± 0.04 | 0.38 ± 0.04 | **0.39 ± 0.04** |
| house_sales | 0.77 ± 0.02 | 0.77 ± 0.01 | **0.83 ± 0.01** | 0.82 ± 0.01 |
| sulfur | -0.22 ± 0.40 | -0.17 ± 0.34 | **0.15 ± 0.05** | -0.02 ± 0.31 |
| medical_charges | 0.96 ± 0.01 | 0.96 ± 0.01 | 0.97 ± 0.01 | **0.97 ± 0.00** |
| MiamiHousing2016 | 0.86 ± 0.01 | 0.86 ± 0.01 | **0.87 ± 0.01** | 0.87 ± 0.01 |
| superconduct | 0.59 ± 0.03 | 0.59 ± 0.03 | 0.68 ± 0.02 | **0.69 ± 0.02** |
| california | 0.75 ± 0.02 | 0.75 ± 0.02 | **0.76 ± 0.02** | 0.76 ± 0.02 |
| fifa | 0.53 ± 0.02 | 0.53 ± 0.02 | **0.61 ± 0.02** | 0.60 ± 0.02 |
| year | 0.06 ± 0.03 | 0.06 ± 0.03 | **0.16 ± 0.04** | 0.16 ± 0.04 |
| **tabular benchmark, regression, with both numerical and categorical features** | | | | |
| yprop_4_1 | -0.19 ± 0.33 | -0.07 ± 0.06 | **-0.04 ± 0.07** | -0.07 ± 0.06 |
| analcatdata_supreme | 0.75 ± 0.01 | 0.75 ± 0.01 | **0.96 ± 0.01** | 0.96 ± 0.01 |
| visualizing_soil | 0.99 ± 0.00 | 0.99 ± 0.00 | 1.00 ± 0.00 | **1.00 ± 0.00** |
| black_friday | 0.12 ± 0.04 | 0.14 ± 0.03 | 0.48 ± 0.02 | **0.49 ± 0.02** |
| diamonds | 0.97 ± 0.00 | 0.97 ± 0.00 | 0.97 ± 0.00 | **0.98 ± 0.00** |
| Mercedes_Benz_Greener_Manufacturing | 0.34 ± 0.03 | 0.35 ± 0.04 | 0.53 ± 0.05 | **0.53 ± 0.05** |
| Brazilian_houses | 0.95 ± 0.02 | 0.95 ± 0.02 | **0.97 ± 0.01** | 0.97 ± 0.01 |
| Bike_Sharing_Demand | 0.08 ± 0.02 | 0.08 ± 0.02 | 0.24 ± 0.03 | **0.24 ± 0.03** |
| OnlineNewsPopularity | 0.04 ± 0.03 | 0.06 ± 0.03 | **0.08 ± 0.03** | 0.08 ± 0.03 |
| nyc-taxi-green-dec-2016 | 0.34 ± 0.05 | 0.34 ± 0.05 | 0.39 ± 0.03 | **0.40 ± 0.04** |
| house_sales | 0.77 ± 0.01 | 0.77 ± 0.01 | **0.82 ± 0.01** | 0.82 ± 0.01 |
| particulate-matter-ukair-2017 | 0.55 ± 0.03 | 0.55 ± 0.03 | 0.61 ± 0.01 | **0.61 ± 0.01** |
| SGEMM_GPU_kernel_performance | 0.96 ± 0.00 | 0.96 ± 0.00 | 0.99 ± 0.00 | **0.99 ± 0.00** |
| **tabular benchmark, classification, with only numerical features** | | | | |
| credit | 0.70 ± 0.03 | 0.71 ± 0.03 | 0.72 ± 0.03 | **0.73 ± 0.02** |
| california | 0.84 ± 0.01 | 0.84 ± 0.01 | **0.85 ± 0.01** | 0.85 ± 0.01 |
| wine | 0.77 ± 0.02 | **0.77 ± 0.02** | 0.75 ± 0.02 | 0.76 ± 0.02 |
| electricity | 0.75 ± 0.01 | 0.75 ± 0.01 | 0.75 ± 0.02 | **0.75 ± 0.01** |
| covertype | 0.74 ± 0.01 | 0.74 ± 0.01 | 0.75 ± 0.01 | **0.75 ± 0.02** |
| pol | 0.93 ± 0.01 | 0.94 ± 0.01 | 0.93 ± 0.01 | **0.94 ± 0.01** |
| house_16H | 0.84 ± 0.01 | 0.84 ± 0.01 | **0.85 ± 0.01** | 0.84 ± 0.01 |
| kdd_ipums_la_97-small | 0.84 ± 0.01 | 0.84 ± 0.01 | **0.84 ± 0.01** | 0.84 ± 0.01 |
| MagicTelescope | 0.82 ± 0.01 | 0.82 ± 0.01 | 0.81 ± 0.02 | **0.82 ± 0.02** |
| bank-marketing | 0.77 ± 0.01 | 0.77 ± 0.01 | 0.77 ± 0.01 | **0.78 ± 0.01** |
| phoneme | 0.83 ± 0.01 | **0.83 ± 0.01** | 0.79 ± 0.01 | 0.83 ± 0.01 |
| MiniBooNE | 0.83 ± 0.01 | 0.85 ± 0.01 | **0.87 ± 0.01** | 0.85 ± 0.02 |
| Higgs | 0.60 ± 0.01 | 0.62 ± 0.02 | 0.65 ± 0.02 | **0.65 ± 0.02** |
| eye_movements | 0.56 ± 0.01 | 0.56 ± 0.02 | 0.56 ± 0.01 | **0.57 ± 0.02** |
| jannis | 0.72 ± 0.01 | 0.72 ± 0.01 | 0.73 ± 0.01 | **0.73 ± 0.01** |
| **tabular benchmark, classification, with both numerical and categorical features** | | | | |
| electricity | 0.75 ± 0.02 | 0.75 ± 0.02 | 0.75 ± 0.01 | **0.76 ± 0.02** |
| eye_movements | 0.56 ± 0.01 | 0.56 ± 0.02 | **0.56 ± 0.02** | 0.56 ± 0.01 |
| covertype | 0.76 ± 0.01 | 0.76 ± 0.01 | 0.76 ± 0.01 | **0.77 ± 0.01** |
| rl | 0.62 ± 0.01 | 0.61 ± 0.01 | **0.63 ± 0.01** | 0.62 ± 0.01 |
| road-safety | 0.69 ± 0.01 | 0.69 ± 0.01 | 0.70 ± 0.01 | **0.71 ± 0.02** |
| compass | 0.66 ± 0.01 | 0.66 ± 0.01 | **0.68 ± 0.02** | 0.68 ± 0.02 |
| KDDCup09_upselling | 0.73 ± 0.01 | 0.73 ± 0.01 | 0.76 ± 0.02 | **0.77 ± 0.02** |
| **scikit-feature benchmark, classification, with only numerical features** | | | | |
| arcene | 0.73 ± 0.04 | 0.77 ± 0.05 | **0.80 ± 0.04** | 0.76 ± 0.04 |
| CLL_SUB_111 | 0.59 ± 0.07 | 0.56 ± 0.07 | 0.63 ± 0.13 | **0.65 ± 0.11** |
| Prostate_GE | 0.84 ± 0.10 | 0.82 ± 0.09 | 0.93 ± 0.07 | **0.93 ± 0.07** |
| SMK_CAN_187 | 0.68 ± 0.08 | 0.65 ± 0.08 | **0.70 ± 0.09** | 0.67 ± 0.08 |
| TOX_171 | 0.80 ± 0.10 | 0.80 ± 0.11 | **0.91 ± 0.04** | 0.82 ± 0.07 |

Table S16: **Comparison to RF and GDBT.** Values are $R^2$ scores for regression problems and accuracies for classification problems, averaged over 10 random data subsamplings for the tabular datasets and over 5 cross-validation folds for the scikit-feature datasets.

| | RF | GBDT | PRS-tree | PRS-kNN | PRS-SVM |
|---|---|---|---|---|---|
| **tabular benchmark, regression, with only numerical features** | | | | | |
| cpu_act | $0.98 \pm 0.00$ | $\mathbf{0.98 \pm 0.00}$ | $0.98 \pm 0.00$ | $0.97 \pm 0.00$ | $0.74 \pm 0.04$ |
| pol | $0.94 \pm 0.00$ | $0.94 \pm 0.01$ | $\mathbf{0.94 \pm 0.01}$ | $0.94 \pm 0.01$ | $0.45 \pm 0.21$ |
| elevators | $0.68 \pm 0.02$ | $\mathbf{0.74 \pm 0.02}$ | $0.71 \pm 0.02$ | $0.73 \pm 0.02$ | $-0.00 \pm 0.00$ |
| isolet | $0.71 \pm 0.02$ | $0.70 \pm 0.02$ | $0.76 \pm 0.02$ | $\mathbf{0.85 \pm 0.02}$ | $0.71 \pm 0.01$ |
| wine_quality | $\mathbf{0.37 \pm 0.02}$ | $0.32 \pm 0.03$ | $0.34 \pm 0.02$ | $0.34 \pm 0.02$ | $0.35 \pm 0.02$ |
| Ailerons | $0.80 \pm 0.02$ | $\mathbf{0.81 \pm 0.01}$ | $0.78 \pm 0.02$ | $0.80 \pm 0.01$ | $-0.00 \pm 0.00$ |
| houses | $0.74 \pm 0.02$ | $\mathbf{0.77 \pm 0.02}$ | $0.74 \pm 0.03$ | $0.76 \pm 0.02$ | $0.74 \pm 0.02$ |
| house_16H | $\mathbf{0.51 \pm 0.07}$ | $0.46 \pm 0.14$ | $0.46 \pm 0.07$ | $0.42 \pm 0.10$ | $0.46 \pm 0.11$ |
| diamonds | $0.94 \pm 0.00$ | $\mathbf{0.94 \pm 0.00}$ | $0.93 \pm 0.01$ | $0.94 \pm 0.01$ | $0.94 \pm 0.00$ |
| Brazilian_houses | $0.98 \pm 0.02$ | $0.98 \pm 0.02$ | $0.98 \pm 0.02$ | $\mathbf{0.98 \pm 0.01}$ | $0.97 \pm 0.01$ |
| Bike_Sharing_Demand | $0.63 \pm 0.02$ | $\mathbf{0.65 \pm 0.02}$ | $0.59 \pm 0.02$ | $0.63 \pm 0.02$ | $0.24 \pm 0.04$ |
| nyc-taxi-green-dec-2016 | $0.39 \pm 0.04$ | $0.41 \pm 0.05$ | $0.44 \pm 0.05$ | $\mathbf{0.46 \pm 0.03}$ | $0.38 \pm 0.04$ |
| house_sales | $0.83 \pm 0.01$ | $\mathbf{0.84 \pm 0.01}$ | $0.83 \pm 0.01$ | $0.83 \pm 0.01$ | $0.83 \pm 0.01$ |
| sulfur | $\mathbf{0.72 \pm 0.10}$ | $0.71 \pm 0.10$ | $0.68 \pm 0.08$ | $0.69 \pm 0.10$ | $0.15 \pm 0.05$ |
| medical_charges | $0.97 \pm 0.00$ | $\mathbf{0.98 \pm 0.00}$ | $0.97 \pm 0.00$ | $0.98 \pm 0.00$ | $0.97 \pm 0.01$ |
| MiamiHousing2016 | $0.86 \pm 0.01$ | $\mathbf{0.88 \pm 0.01}$ | $0.87 \pm 0.01$ | $0.86 \pm 0.01$ | $0.87 \pm 0.01$ |
| superconduct | $0.84 \pm 0.01$ | $0.84 \pm 0.01$ | $\mathbf{0.84 \pm 0.01}$ | $0.82 \pm 0.01$ | $0.68 \pm 0.02$ |
| california | $0.75 \pm 0.02$ | $0.77 \pm 0.02$ | $\mathbf{0.78 \pm 0.02}$ | $0.77 \pm 0.02$ | $0.76 \pm 0.02$ |
| fifa | $0.62 \pm 0.02$ | $\mathbf{0.64 \pm 0.02}$ | $0.56 \pm 0.03$ | $0.61 \pm 0.02$ | $0.61 \pm 0.02$ |
| year | $0.15 \pm 0.02$ | $0.16 \pm 0.04$ | $0.14 \pm 0.04$ | $\mathbf{0.18 \pm 0.02}$ | $0.16 \pm 0.04$ |
| **tabular benchmark, regression, with both numerical and categorical features** | | | | | |
| yprop_4_1 | $0.04 \pm 0.01$ | $0.04 \pm 0.01$ | $0.02 \pm 0.01$ | $\mathbf{0.05 \pm 0.01}$ | $-0.04 \pm 0.07$ |
| analcatdata_supreme | $0.98 \pm 0.01$ | $\mathbf{0.98 \pm 0.01}$ | $0.98 \pm 0.01$ | $0.98 \pm 0.01$ | $0.96 \pm 0.01$ |
| visualizing_soil | $1.00 \pm 0.00$ | $\mathbf{1.00 \pm 0.00}$ | $1.00 \pm 0.00$ | $1.00 \pm 0.00$ | $1.00 \pm 0.00$ |
| black_friday | $0.51 \pm 0.03$ | $0.55 \pm 0.03$ | $\mathbf{0.57 \pm 0.03}$ | $0.55 \pm 0.03$ | $0.48 \pm 0.02$ |
| diamonds | $0.97 \pm 0.00$ | $\mathbf{0.98 \pm 0.00}$ | $0.97 \pm 0.00$ | $0.96 \pm 0.01$ | $0.97 \pm 0.00$ |
| Mercedes_Benz_Greener_Manufacturing | $0.48 \pm 0.05$ | $0.55 \pm 0.05$ | $\mathbf{0.55 \pm 0.05}$ | $0.54 \pm 0.05$ | $0.53 \pm 0.05$ |
| Brazilian_houses | $0.98 \pm 0.02$ | $0.98 \pm 0.02$ | $0.98 \pm 0.02$ | $\mathbf{0.98 \pm 0.01}$ | $0.97 \pm 0.01$ |
| Bike_Sharing_Demand | $0.84 \pm 0.01$ | $\mathbf{0.89 \pm 0.01}$ | $0.88 \pm 0.02$ | $0.86 \pm 0.02$ | $0.24 \pm 0.03$ |
| OnlineNewsPopularity | $0.10 \pm 0.03$ | $0.08 \pm 0.03$ | $0.08 \pm 0.03$ | $\mathbf{0.10 \pm 0.03}$ | $0.08 \pm 0.03$ |
| nyc-taxi-green-dec-2016 | $0.41 \pm 0.03$ | $0.42 \pm 0.04$ | $0.45 \pm 0.05$ | $\mathbf{0.48 \pm 0.03}$ | $0.39 \pm 0.03$ |
| house_sales | $0.83 \pm 0.01$ | $\mathbf{0.85 \pm 0.01}$ | $0.84 \pm 0.01$ | $0.84 \pm 0.00$ | $0.82 \pm 0.01$ |
| particulate-matter-ukair-2017 | $0.61 \pm 0.02$ | $\mathbf{0.63 \pm 0.01}$ | $0.56 \pm 0.02$ | $0.62 \pm 0.02$ | $0.61 \pm 0.01$ |
| SGEMM_GPU_kernel_performance | $\mathbf{1.00 \pm 0.00}$ | $1.00 \pm 0.00$ | $1.00 \pm 0.00$ | $1.00 \pm 0.00$ | $0.99 \pm 0.00$ |
| **tabular benchmark, classification, with only numerical features** | | | | | |
| credit | $0.77 \pm 0.01$ | $\mathbf{0.77 \pm 0.01}$ | $0.75 \pm 0.01$ | $0.76 \pm 0.01$ | $0.72 \pm 0.03$ |
| california | $0.85 \pm 0.01$ | $0.86 \pm 0.01$ | $\mathbf{0.86 \pm 0.01}$ | $0.86 \pm 0.01$ | $0.85 \pm 0.01$ |
| wine | $\mathbf{0.79 \pm 0.02}$ | $0.78 \pm 0.02$ | $0.78 \pm 0.02$ | $0.77 \pm 0.02$ | $0.75 \pm 0.02$ |
| electricity | $\mathbf{0.79 \pm 0.01}$ | $0.79 \pm 0.02$ | $0.78 \pm 0.01$ | $0.77 \pm 0.01$ | $0.75 \pm 0.02$ |
| covertype | $0.75 \pm 0.02$ | $0.74 \pm 0.01$ | $0.74 \pm 0.02$ | $\mathbf{0.76 \pm 0.02}$ | $0.75 \pm 0.01$ |
| pol | $0.96 \pm 0.01$ | $\mathbf{0.97 \pm 0.01}$ | $0.96 \pm 0.01$ | $0.95 \pm 0.01$ | $0.93 \pm 0.01$ |
| house_16H | $0.86 \pm 0.01$ | $\mathbf{0.86 \pm 0.01}$ | $0.85 \pm 0.02$ | $0.85 \pm 0.01$ | $0.85 \pm 0.01$ |
| kdd_ipums_la_97-small | $0.88 \pm 0.01$ | $\mathbf{0.88 \pm 0.01}$ | $0.87 \pm 0.01$ | $0.87 \pm 0.01$ | $0.84 \pm 0.01$ |
| MagicTelescope | $\mathbf{0.83 \pm 0.01}$ | $0.82 \pm 0.01$ | $0.80 \pm 0.01$ | $0.81 \pm 0.02$ | $0.81 \pm 0.02$ |
| bank-marketing | $0.78 \pm 0.02$ | $\mathbf{0.78 \pm 0.01}$ | $0.75 \pm 0.02$ | $0.78 \pm 0.01$ | $0.77 \pm 0.01$ |
| phoneme | $\mathbf{0.86 \pm 0.01}$ | $0.85 \pm 0.01$ | $0.84 \pm 0.01$ | $0.84 \pm 0.01$ | $0.79 \pm 0.01$ |
| MiniBooNE | $0.90 \pm 0.01$ | $\mathbf{0.90 \pm 0.01}$ | $0.90 \pm 0.01$ | $0.89 \pm 0.01$ | $0.87 \pm 0.01$ |
| Higgs | $0.66 \pm 0.01$ | $0.66 \pm 0.01$ | $0.65 \pm 0.02$ | $\mathbf{0.67 \pm 0.01}$ | $0.65 \pm 0.02$ |
| eye_movements | $0.56 \pm 0.01$ | $0.56 \pm 0.01$ | $\mathbf{0.63 \pm 0.02}$ | $0.57 \pm 0.02$ | $0.56 \pm 0.01$ |
| jannis | $0.73 \pm 0.02$ | $0.73 \pm 0.01$ | $0.73 \pm 0.02$ | $\mathbf{0.74 \pm 0.01}$ | $0.73 \pm 0.01$ |
| **tabular benchmark, classification, with both numerical and categorical features** | | | | | |
| electricity | $0.79 \pm 0.01$ | $\mathbf{0.79 \pm 0.02}$ | $0.78 \pm 0.01$ | $0.77 \pm 0.02$ | $0.75 \pm 0.01$ |
| eye_movements | $0.58 \pm 0.02$ | $0.57 \pm 0.02$ | $\mathbf{0.63 \pm 0.01}$ | $0.57 \pm 0.02$ | $0.56 \pm 0.02$ |
| covertype | $0.78 \pm 0.01$ | $0.77 \pm 0.02$ | $0.75 \pm 0.02$ | $\mathbf{0.78 \pm 0.02}$ | $0.76 \pm 0.01$ |
| rl | $0.69 \pm 0.01$ | $0.71 \pm 0.01$ | $\mathbf{0.77 \pm 0.01}$ | $0.73 \pm 0.02$ | $0.63 \pm 0.01$ |
| road-safety | $0.72 \pm 0.01$ | $0.72 \pm 0.01$ | $0.71 \pm 0.02$ | $\mathbf{0.72 \pm 0.01}$ | $0.70 \pm 0.01$ |
| compass | $0.68 \pm 0.02$ | $\mathbf{0.69 \pm 0.02}$ | $0.67 \pm 0.02$ | $0.68 \pm 0.02$ | $0.68 \pm 0.02$ |
| KDDCup09_upselling | $0.79 \pm 0.01$ | $\mathbf{0.79 \pm 0.01}$ | $0.79 \pm 0.01$ | $0.77 \pm 0.01$ | $0.76 \pm 0.02$ |
| **scikit-feature benchmark, classification, with only numerical features** | | | | | |
| arcene | $0.78 \pm 0.05$ | $0.74 \pm 0.05$ | $0.81 \pm 0.05$ | $\mathbf{0.81 \pm 0.05}$ | $0.80 \pm 0.04$ |
| CLL_SUB_111 | $0.67 \pm 0.13$ | $0.70 \pm 0.08$ | $\mathbf{0.70 \pm 0.09}$ | $0.55 \pm 0.16$ | $0.63 \pm 0.13$ |
| Prostate_GE | $0.90 \pm 0.08$ | $0.85 \pm 0.10$ | $0.88 \pm 0.11$ | $0.89 \pm 0.07$ | $\mathbf{0.93 \pm 0.07}$ |
| SMK_CAN_187 | $0.65 \pm 0.10$ | $0.67 \pm 0.04$ | $0.64 \pm 0.07$ | $0.66 \pm 0.04$ | $\mathbf{0.70 \pm 0.09}$ |
| TOX_171 | $0.71 \pm 0.06$ | $0.77 \pm 0.07$ | $0.75 \pm 0.05$ | $0.88 \pm 0.06$ | $\mathbf{0.91 \pm 0.04}$ |

Table S17: **Number of features used per base model (tree)**, i.e. for RS: the number $K$ of randomly sampled features (optimized on the validation test), for PRS: the sum $\sum_{j=1}^{M} \alpha_j$, and for RaSE: the average feature subset size over the $T$ models. Values are means and standard deviations over 10 random data subsamplings for the tabular datasets and over 5 cross-validation folds for the scikit-feature datasets.

| | RS-tree | PRS-tree | RaSE-tree |
|---|---|---|---|
| **tabular benchmark, regression, with only numerical features** | | | |
| cpu_act | 11.10 ± 3.30 | **8.47 ± 0.57** | 14.92 ± 0.83 |
| pol | 26.00 ± 0.00 | **10.95 ± 0.64** | 18.01 ± 1.51 |
| elevators | 7.70 ± 0.90 | 6.03 ± 0.31 | **5.81 ± 0.47** |
| isolet | 189.90 ± 53.01 | 48.75 ± 3.35 | **27.84 ± 0.46** |
| wine_quality | 5.80 ± 0.60 | 5.77 ± 0.69 | **1.00 ± 0.00** |
| Ailerons | 16.00 ± 0.00 | **6.59 ± 0.52** | 17.95 ± 1.13 |
| houses | 4.00 ± 0.00 | **3.55 ± 0.17** | 4.50 ± 0.55 |
| house_16H | 7.30 ± 1.42 | **5.43 ± 1.58** | 10.72 ± 1.10 |
| diamonds | 2.50 ± 0.50 | 2.41 ± 0.18 | **1.00 ± 0.00** |
| Brazilian_houses | 8.00 ± 0.00 | 4.16 ± 0.31 | **3.04 ± 0.46** |
| Bike_Sharing_Demand | 3.00 ± 0.00 | 2.66 ± 0.36 | **1.09 ± 0.14** |
| nyc-taxi-green-dec-2016 | 3.70 ± 0.46 | **1.23 ± 0.20** | 1.55 ± 0.28 |
| house_sales | 8.00 ± 0.00 | **6.13 ± 0.33** | 9.95 ± 0.68 |
| sulfur | 4.50 ± 1.50 | **2.20 ± 0.60** | 3.86 ± 0.57 |
| medical_charges | 3.00 ± 0.00 | **1.78 ± 0.16** | 2.29 ± 0.19 |
| MiamiHousing2016 | **6.00 ± 0.00** | 6.58 ± 0.31 | 8.85 ± 0.32 |
| superconduct | 22.10 ± 10.14 | **8.91 ± 0.71** | 22.69 ± 0.68 |
| california | **4.00 ± 0.00** | 4.03 ± 0.20 | 4.20 ± 0.50 |
| fifa | 2.00 ± 0.00 | 2.09 ± 0.33 | **1.43 ± 0.50** |
| year | 39.30 ± 9.24 | **8.94 ± 1.88** | 23.27 ± 0.62 |
| **tabular benchmark, regression, with both numerical and categorical features** | | | |
| yprop_4_1 | 10.70 ± 6.10 | 4.80 ± 0.99 | **3.48 ± 1.26** |
| analcatdata_supreme | 12.00 ± 0.00 | **1.62 ± 0.27** | 2.65 ± 1.33 |
| visualizing_soil | 5.00 ± 0.00 | 4.19 ± 0.26 | **3.90 ± 0.16** |
| black_friday | 12.00 ± 0.00 | 2.58 ± 0.32 | **2.10 ± 0.66** |
| diamonds | 15.60 ± 5.20 | **11.94 ± 1.06** | 18.09 ± 0.98 |
| Mercedes_Benz_Greener_Manufacturing | 125.10 ± 33.45 | 34.65 ± 2.14 | **17.00 ± 3.31** |
| Brazilian_houses | 17.00 ± 0.00 | 5.01 ± 0.86 | **3.95 ± 0.65** |
| Bike_Sharing_Demand | 15.00 ± 5.00 | **7.91 ± 0.56** | 9.60 ± 1.12 |
| OnlineNewsPopularity | 16.00 ± 8.05 | **6.50 ± 1.13** | 7.47 ± 2.38 |
| nyc-taxi-green-dec-2016 | 15.40 ± 1.80 | **2.78 ± 0.38** | 6.39 ± 2.90 |
| house_sales | 10.00 ± 0.00 | **7.55 ± 0.70** | 12.65 ± 0.98 |
| particulate-matter-ukair-2017 | 13.00 ± 0.00 | 5.42 ± 1.14 | **1.81 ± 0.25** |
| SGEMM_GPU_kernel_performance | 15.00 ± 0.00 | **3.37 ± 0.37** | 6.14 ± 1.17 |
| **tabular benchmark, classification, with only numerical features** | | | |
| credit | 3.60 ± 0.92 | 4.42 ± 0.49 | **2.85 ± 0.26** |
| california | **3.70 ± 0.46** | 4.03 ± 0.21 | 3.85 ± 0.73 |
| wine | 5.40 ± 0.92 | **5.29 ± 0.44** | 7.71 ± 0.68 |
| electricity | 4.00 ± 0.00 | **3.11 ± 0.23** | 3.43 ± 0.71 |
| covertype | 4.80 ± 0.60 | **4.05 ± 0.19** | 6.13 ± 0.62 |
| pol | 14.30 ± 3.90 | **9.52 ± 0.71** | 19.54 ± 1.09 |
| house_16H | 7.70 ± 0.90 | **7.62 ± 0.58** | 11.43 ± 0.79 |
| kdd_ipums_la_97-small | 8.20 ± 1.99 | **3.42 ± 0.57** | 10.30 ± 1.60 |
| MagicTelescope | **5.00 ± 0.00** | 5.43 ± 0.29 | 7.35 ± 0.35 |
| bank-marketing | 3.60 ± 0.49 | **3.41 ± 0.15** | 5.27 ± 0.79 |
| phoneme | 3.50 ± 1.50 | **3.20 ± 0.13** | 4.46 ± 0.25 |
| MiniBooNE | 18.70 ± 4.61 | **11.94 ± 1.36** | 24.09 ± 0.62 |
| Higgs | 10.80 ± 1.83 | **6.08 ± 0.77** | 14.59 ± 1.96 |
| eye_movements | 3.10 ± 1.76 | 2.73 ± 0.30 | **2.17 ± 0.27** |
| jannis | 18.50 ± 6.95 | **9.05 ± 0.88** | 23.98 ± 1.17 |
| **tabular benchmark, classification, with both numerical and categorical features** | | | |
| electricity | 7.00 ± 0.00 | **5.02 ± 0.75** | 7.37 ± 0.99 |
| eye_movements | 5.40 ± 2.65 | 3.05 ± 0.61 | **2.40 ± 0.39** |
| covertype | 46.00 ± 0.00 | **14.09 ± 1.61** | 25.53 ± 0.76 |
| rl | 16.60 ± 2.94 | **5.70 ± 0.73** | 17.67 ± 3.00 |
| road-safety | 15.00 ± 3.00 | **5.62 ± 1.26** | 9.40 ± 3.73 |
| compass | 23.80 ± 7.07 | **6.67 ± 2.13** | 17.46 ± 3.32 |
| KDDCup09_upselling | 48.60 ± 6.80 | **6.51 ± 1.09** | 9.08 ± 2.33 |
| **scikit-feature benchmark, classification, with only numerical features** | | | |
| arcene | 2100.00 ± 3950.19 | 128.03 ± 192.70 | **8.23 ± 0.13** |
| CLL_SUB_111 | 6540.00 ± 824.08 | 407.63 ± 205.17 | **5.68 ± 0.60** |
| Prostate_GE | 1948.40 ± 2275.37 | 226.31 ± 113.33 | **5.19 ± 0.54** |
| SMK_CAN_187 | 5811.20 ± 7474.69 | 861.53 ± 326.82 | **7.27 ± 0.33** |
| TOX_171 | 256.40 ± 187.45 | 172.96 ± 133.50 | **7.99 ± 0.09** |

Table S18: **Number of features used per base model (kNN)**, i.e. for RS: the number $K$ of randomly sampled features (optimized on the validation test), for PRS: the sum $\sum_{j=1}^{M} \alpha_j$, and for RaSE: the average feature subset size over the $T$ models. Values are means and standard deviations over 10 random data subsamplings for the tabular datasets and over 5 cross-validation folds for the scikit-feature datasets.

| | RS-kNN | PRS-kNN | RaSE-kNN |
|---|---|---|---|
| **tabular benchmark, regression, with only numerical features** | | | |
| cpu_act | 15.50 ± 5.50 | **5.59 ± 0.54** | 6.00 ± 1.27 |
| pol | 26.00 ± 0.00 | **7.51 ± 0.60** | 10.45 ± 1.66 |
| elevators | 10.40 ± 3.67 | 5.68 ± 0.46 | **4.82 ± 0.48** |
| isolet | 295.80 ± 30.60 | 81.65 ± 6.86 | **29.86 ± 0.20** |
| wine_quality | 6.00 ± 0.00 | **5.76 ± 0.32** | 7.64 ± 0.66 |
| Ailerons | 27.90 ± 7.79 | 6.21 ± 0.37 | **5.96 ± 0.70** |
| houses | 8.00 ± 0.00 | **2.56 ± 0.07** | 3.45 ± 0.31 |
| house_16H | 8.80 ± 2.40 | **5.89 ± 1.06** | 10.89 ± 1.04 |
| diamonds | 2.90 ± 0.30 | **2.62 ± 0.19** | 2.85 ± 0.34 |
| Brazilian_houses | 5.50 ± 2.06 | 2.82 ± 0.33 | **2.73 ± 0.68** |
| Bike_Sharing_Demand | 3.00 ± 0.00 | 2.58 ± 0.08 | **2.40 ± 0.19** |
| nyc-taxi-green-dec-2016 | 3.90 ± 0.30 | **1.31 ± 0.21** | 1.43 ± 0.23 |
| house_sales | 9.40 ± 2.80 | **4.89 ± 0.31** | 6.48 ± 0.85 |
| sulfur | 5.70 ± 0.90 | **2.04 ± 0.34** | 2.36 ± 0.52 |
| medical_charges | 3.00 ± 0.00 | 1.56 ± 0.05 | **1.01 ± 0.01** |
| MiamiHousing2016 | **6.00 ± 0.00** | 6.66 ± 0.75 | 8.65 ± 1.28 |
| superconduct | 11.50 ± 3.69 | **8.55 ± 1.39** | 21.05 ± 2.52 |
| california | 7.60 ± 1.20 | **3.47 ± 0.13** | 5.08 ± 0.53 |
| fifa | **2.00 ± 0.00** | 2.01 ± 0.03 | 2.00 ± 0.00 |
| year | 30.60 ± 8.57 | **12.45 ± 1.74** | 23.37 ± 2.10 |
| **tabular benchmark, regression, with both numerical and categorical features** | | | |
| yprop_4_1 | 17.70 ± 6.80 | **7.73 ± 1.48** | 19.13 ± 3.31 |
| analcatdata_supreme | 12.00 ± 0.00 | **1.80 ± 0.33** | 2.15 ± 0.51 |
| visualizing_soil | 5.00 ± 0.00 | 3.85 ± 0.05 | **3.52 ± 0.00** |
| black_friday | 5.60 ± 1.80 | 2.25 ± 0.29 | **2.17 ± 0.42** |
| diamonds | 11.00 ± 2.00 | **7.66 ± 0.73** | 8.74 ± 0.98 |
| Mercedes_Benz_Greener_Manufacturing | 77.60 ± 25.63 | 37.59 ± 3.10 | **20.06 ± 1.44** |
| Brazilian_houses | 7.60 ± 0.80 | **2.88 ± 0.49** | 3.16 ± 0.80 |
| Bike_Sharing_Demand | 9.40 ± 1.20 | **5.87 ± 0.48** | 6.56 ± 0.15 |
| OnlineNewsPopularity | 13.50 ± 4.50 | **7.18 ± 1.15** | 20.27 ± 1.59 |
| nyc-taxi-green-dec-2016 | 14.20 ± 2.75 | **3.21 ± 0.71** | 4.79 ± 0.95 |
| house_sales | 11.80 ± 3.60 | **5.52 ± 0.70** | 7.55 ± 1.11 |
| particulate-matter-ukair-2017 | 11.40 ± 1.96 | **3.88 ± 0.61** | 4.46 ± 0.60 |
| SGEMM_GPU_kernel_performance | 8.00 ± 0.00 | 2.93 ± 0.12 | **2.17 ± 0.68** |
| **tabular benchmark, classification, with only numerical features** | | | |
| credit | **3.00 ± 0.00** | 3.48 ± 0.23 | 4.27 ± 0.37 |
| california | **3.30 ± 0.46** | 3.39 ± 0.15 | 4.41 ± 0.90 |
| wine | 4.40 ± 1.11 | **3.89 ± 0.15** | 7.00 ± 0.45 |
| electricity | 4.20 ± 0.98 | **2.72 ± 0.19** | 3.45 ± 0.77 |
| covertype | 4.40 ± 0.92 | **3.10 ± 0.16** | 4.07 ± 0.41 |
| pol | 13.00 ± 0.00 | **8.41 ± 0.30** | 13.84 ± 1.11 |
| house_16H | **5.30 ± 1.49** | 5.43 ± 0.39 | 10.61 ± 0.88 |
| kdd_ipums_la_97-small | 4.90 ± 1.92 | **2.46 ± 0.28** | 3.46 ± 0.50 |
| MagicTelescope | 5.50 ± 1.50 | **4.22 ± 0.22** | 5.93 ± 0.54 |
| bank-marketing | 3.80 ± 0.40 | **3.07 ± 0.19** | 3.83 ± 0.58 |
| phoneme | 3.80 ± 1.47 | **2.41 ± 0.28** | 4.03 ± 0.44 |
| MiniBooNE | 10.90 ± 4.18 | **7.71 ± 1.81** | 23.51 ± 2.22 |
| Higgs | 5.30 ± 0.90 | **4.03 ± 0.51** | 6.04 ± 0.94 |
| eye_movements | 4.10 ± 2.91 | **3.14 ± 0.18** | 8.26 ± 2.31 |
| jannis | 14.50 ± 8.80 | **5.89 ± 0.55** | 20.43 ± 1.81 |
| **tabular benchmark, classification, with both numerical and categorical features** | | | |
| electricity | 6.10 ± 1.14 | **4.36 ± 0.30** | 5.94 ± 1.64 |
| eye_movements | 5.00 ± 1.55 | **3.45 ± 0.40** | 11.90 ± 4.06 |
| covertype | 35.50 ± 6.87 | **10.46 ± 0.88** | 24.06 ± 1.50 |
| rl | 12.00 ± 2.00 | **4.65 ± 0.43** | 14.65 ± 2.18 |
| road-safety | 12.10 ± 3.65 | **5.88 ± 0.45** | 14.07 ± 1.52 |
| compass | 11.80 ± 4.60 | **6.37 ± 0.65** | 16.70 ± 3.34 |
| KDDCup09_upselling | 17.80 ± 10.15 | 5.82 ± 0.42 | **5.19 ± 0.59** |
| **scikit-feature benchmark, classification, with only numerical features** | | | |
| arcene | 4500.00 ± 4494.44 | 328.85 ± 103.85 | **9.26 ± 1.05** |
| CLL_SUB_111 | 109.20 ± 71.57 | 222.60 ± 103.72 | **5.52 ± 1.35** |
| Prostate_GE | 1259.80 ± 2353.18 | **4.77 ± 1.77** | 6.08 ± 0.68 |
| SMK_CAN_187 | 29.00 ± 56.00 | 743.66 ± 165.22 | **8.00 ± 0.79** |
| TOX_171 | 1206.80 ± 2270.71 | 111.80 ± 67.25 | **8.53 ± 0.64** |

Table S19: **Number of features used per base model (SVM)**, i.e. for RS: the number $K$ of randomly sampled features (optimized on the validation test), for PRS: the sum $\sum_{j=1}^{M} \alpha_j$, and for RaSE: the average feature subset size over the $T$ models. Values are means and standard deviations over 10 random data subsamplings for the tabular datasets and over 5 cross-validation folds for the scikit-feature datasets.

| | RS-SVM | PRS-SVM | RaSE-SVM |
|---|---|---|---|
| **tabular benchmark, regression, with only numerical features** | | | |
| cpu_act | 21.00 ± 0.00 | **1.87 ± 0.17** | 1.91 ± 0.13 |
| pol | 26.00 ± 0.00 | **1.61 ± 0.66** | 5.42 ± 0.27 |
| elevators | 1.00 ± 0.00 | **0.02 ± 0.02** | 8.68 ± 0.00 |
| isolet | 613.00 ± 0.00 | 42.45 ± 4.33 | **29.60 ± 0.29** |
| wine_quality | 8.50 ± 2.50 | **7.45 ± 0.59** | 9.15 ± 0.44 |
| Ailerons | 1.00 ± 0.00 | **0.00 ± 0.00** | 15.14 ± 0.00 |
| houses | 8.00 ± 0.00 | **5.64 ± 0.65** | 6.43 ± 0.68 |
| house_16H | 16.00 ± 0.00 | **9.01 ± 1.18** | 12.43 ± 1.61 |
| diamonds | 3.00 ± 0.00 | 3.08 ± 0.57 | **2.90 ± 0.69** |
| Brazilian_houses | 8.00 ± 0.00 | 3.12 ± 0.37 | **2.64 ± 0.50** |
| Bike_Sharing_Demand | 5.70 ± 0.90 | **1.85 ± 0.12** | 2.00 ± 0.00 |
| nyc-taxi-green-dec-2016 | 9.00 ± 0.00 | **1.42 ± 0.54** | 3.04 ± 0.45 |
| house_sales | 10.10 ± 3.21 | **5.63 ± 0.56** | 7.14 ± 0.89 |
| sulfur | 4.10 ± 1.58 | **0.30 ± 0.06** | 3.97 ± 1.25 |
| medical_charges | 3.00 ± 0.00 | 1.46 ± 0.18 | **1.02 ± 0.03** |
| MiamiHousing2016 | 13.00 ± 0.00 | **9.34 ± 0.39** | 10.59 ± 0.32 |
| superconduct | 79.00 ± 0.00 | **9.56 ± 2.15** | 12.14 ± 3.44 |
| california | 8.00 ± 0.00 | **4.89 ± 0.32** | 5.86 ± 0.76 |
| fifa | 5.00 ± 0.00 | **1.75 ± 0.13** | 2.00 ± 0.00 |
| year | 90.00 ± 0.00 | **12.06 ± 2.97** | 15.03 ± 3.77 |
| **tabular benchmark, regression, with both numerical and categorical features** | | | |
| yprop_4_1 | **1.80 ± 2.40** | 2.11 ± 2.07 | 14.49 ± 1.94 |
| analcatdata_supreme | 12.00 ± 0.00 | 1.31 ± 0.16 | **1.01 ± 0.01** |
| visualizing_soil | 5.00 ± 0.00 | **3.70 ± 0.17** | 4.00 ± 0.00 |
| black_friday | 13.40 ± 6.64 | **0.95 ± 0.08** | 1.00 ± 0.00 |
| diamonds | 26.00 ± 0.00 | **13.70 ± 1.57** | 18.41 ± 1.27 |
| Mercedes_Benz_Greener_Manufacturing | 149.70 ± 89.27 | 32.27 ± 6.69 | **21.54 ± 7.33** |
| Brazilian_houses | 17.00 ± 0.00 | 3.21 ± 0.38 | **3.03 ± 0.68** |
| Bike_Sharing_Demand | 12.20 ± 6.78 | **2.09 ± 0.32** | 2.51 ± 0.41 |
| OnlineNewsPopularity | 24.30 ± 4.73 | **11.50 ± 1.87** | 22.58 ± 2.08 |
| nyc-taxi-green-dec-2016 | 31.00 ± 0.00 | **3.03 ± 1.18** | 5.68 ± 1.46 |
| house_sales | 17.20 ± 3.60 | **6.25 ± 0.76** | 8.08 ± 0.85 |
| particulate-matter-ukair-2017 | 26.00 ± 0.00 | **3.66 ± 1.08** | 4.06 ± 1.09 |
| SGEMM_GPU_kernel_performance | 15.00 ± 0.00 | 2.49 ± 0.45 | **1.43 ± 0.25** |
| **tabular benchmark, classification, with only numerical features** | | | |
| credit | 4.40 ± 3.17 | **2.32 ± 0.21** | 3.93 ± 1.03 |
| california | 8.00 ± 0.00 | **3.43 ± 0.34** | 5.39 ± 0.72 |
| wine | 8.10 ± 2.98 | **2.48 ± 0.19** | 8.61 ± 0.80 |
| electricity | 6.70 ± 0.90 | **1.98 ± 0.20** | 5.08 ± 0.42 |
| covertype | 7.50 ± 2.50 | **2.13 ± 0.17** | 4.75 ± 0.42 |
| pol | 22.10 ± 5.96 | **5.07 ± 1.07** | 18.02 ± 2.24 |
| house_16H | 11.60 ± 4.54 | **4.86 ± 0.29** | 11.92 ± 1.16 |
| kdd_ipums_la_97-small | 9.20 ± 3.84 | **1.40 ± 0.13** | 8.18 ± 2.37 |
| MagicTelescope | 10.00 ± 0.00 | **3.33 ± 0.23** | 6.99 ± 0.52 |
| bank-marketing | 6.70 ± 0.90 | **2.65 ± 0.16** | 4.81 ± 0.49 |
| phoneme | 5.00 ± 0.00 | **1.67 ± 0.12** | 4.63 ± 0.35 |
| MiniBooNE | 14.00 ± 5.20 | **4.10 ± 0.60** | 14.13 ± 5.08 |
| Higgs | 5.00 ± 3.03 | **2.98 ± 0.31** | 8.71 ± 1.54 |
| eye_movements | 7.80 ± 4.92 | **2.09 ± 0.23** | 10.22 ± 3.36 |
| jannis | 30.80 ± 15.94 | **3.12 ± 0.41** | 23.65 ± 2.36 |
| **tabular benchmark, classification, with both numerical and categorical features** | | | |
| electricity | 10.80 ± 3.97 | **2.83 ± 0.43** | 8.30 ± 1.39 |
| eye_movements | 10.40 ± 8.52 | **2.01 ± 0.22** | 13.07 ± 2.96 |
| covertype | 93.00 ± 0.00 | **8.29 ± 1.27** | 26.15 ± 1.19 |
| rl | 24.10 ± 9.27 | **2.73 ± 0.65** | 18.99 ± 2.88 |
| road-safety | 16.80 ± 7.10 | **1.70 ± 0.21** | 14.78 ± 3.26 |
| compass | 39.80 ± 24.20 | **3.53 ± 0.27** | 19.33 ± 3.48 |
| KDDCup09_upselling | 98.80 ± 15.60 | **2.65 ± 0.35** | 7.48 ± 4.11 |
| **scikit-feature benchmark, classification, with only numerical features** | | | |
| arcene | 100.00 ± 0.00 | 188.33 ± 138.66 | **9.38 ± 0.66** |
| CLL_SUB_111 | 2333.40 ± 4503.59 | 113.33 ± 82.10 | **6.63 ± 0.54** |
| Prostate_GE | 1254.80 ± 2355.60 | 17.19 ± 12.82 | **5.94 ± 0.47** |
| SMK_CAN_187 | 4055.40 ± 7969.05 | 574.63 ± 377.67 | **8.07 ± 0.69** |
| TOX_171 | 76.00 ± 0.00 | 94.18 ± 53.14 | **9.35 ± 0.44** |

# G   Additional results on the DREAM4 networks

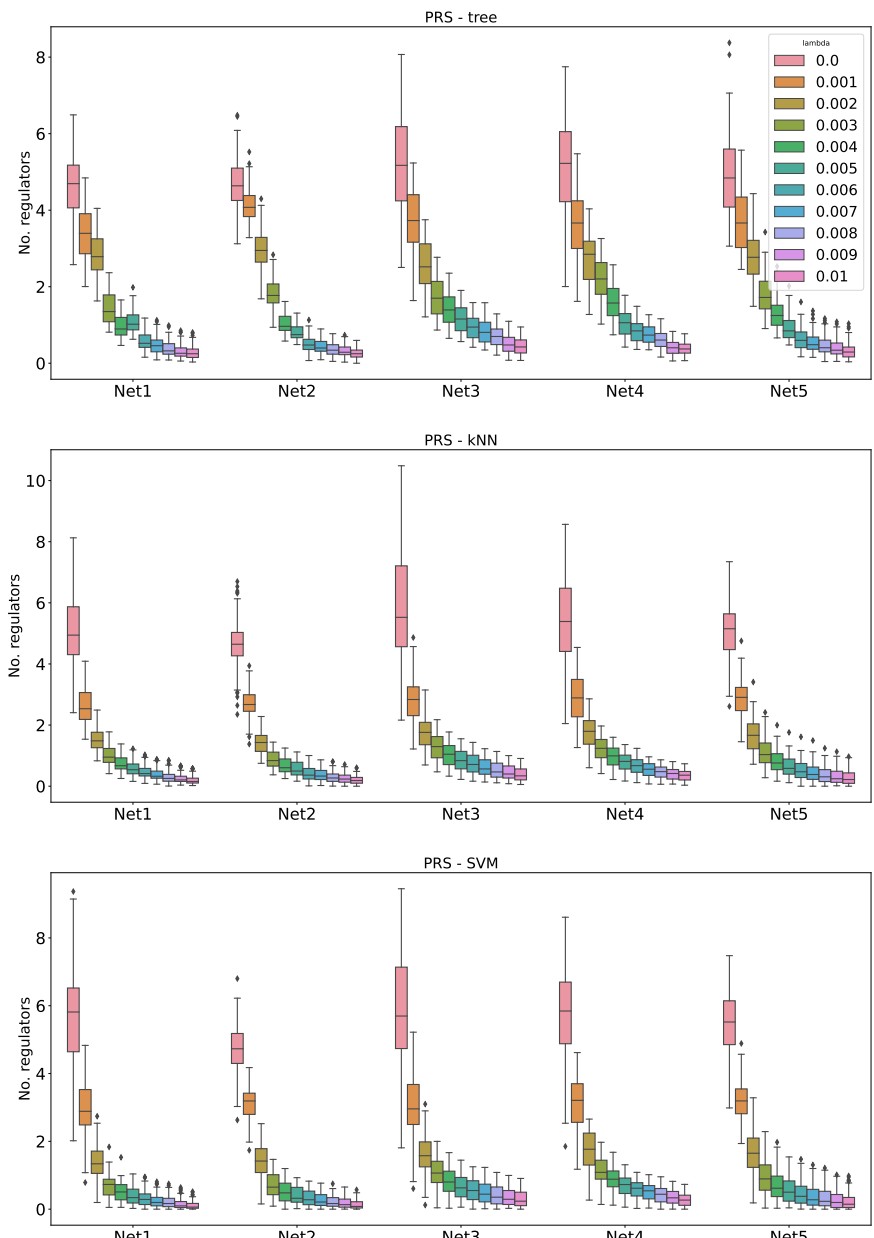

Figure S7: **Expected number of selected candidate regulators per base model in a PRS ensemble**, i.e. $\sum_{j=1}^{M} \alpha_{j,g}$, where $M$ is the number of candidate regulators. The boxplots summarize the values over the 100 target genes.

