# OpenReview forum: "Optimizing Model-Agnostic Random Subspace Ensembles"
_TMLR — Rejected by TMLR_

### Review · Reviewer_sJJj · 2022-11-30

**Summary Of Contributions:**

The paper shows how to optimize the parameters of a Bernoulli distribution governing random sampling in the Random Subspace technique. An importance sampling approach is proposed to avoid retraining the base model for each random subset of the features.



**Audience:**

Yes

**Broader Impact Concerns:**

If equipped with solid theoretical and empirical proof, the proposed idea of replacing an intractable bi-level optimization problem with a re-weighted Monte Carlo sampling approach may have an impact that goes beyond ensemble methods.  As it is, however, the paper does not seem to provide evidence of its main claim.



**Claims And Evidence:**

No

**Requested Changes:**

${\bf Main \ questions}$

1. Are the parameters of the base models pre-trained using all features as input? If so, are they set to zero when they are not selected?

2. How does Equation 5 show that the model should be retrained after $\alpha$ is updated?

3. What are the technical differences and advantages between optimizing the *expectation of the loss*, as in Equation 17, or the output of the ensemble model?

4. Even in the proposed approach, base models are retrained many times and a practical threshold of $0.9 T$ is introduced. According to Appendix E.2, the base model is retrained $\approx 100000 $in each simulation. Is this enough to ensure a consistent speed-up of the training process? It would be interesting to add a simulation to show the time complexity versus the performance of
different practical thresholds.

5. One of the claimed advantages over ${\tt RaSE}$ is the possibility of regularizing the model parameter $\alpha$. Isn't this equivalent to constraining the number of sampled features to be small? Why the number of iterations for ${\tt RaSE}$ is so small compared with the number of gradient updates for ${\tt PRS}$? Is this for matching the computational complexity of the two methods?

${\bf Other \ questions}$

1. In the abstract, the sentence starting with *While...* is not completely clear. For example, what does *it* refer to?

2. Could the feature-selection ability of the method be compared with existing feature-selection strategies?

3.  What is the difference between the proposed approach and the *mixture of experts* approach? Can the Bernoulli distribution be seen as a stochastic gate function?

4. Section 2 contains a review of existing Monte Carlo techniques. What is special in this case?

5. Is any sort of regularization applied to the base models?

6. The performance of ${\tt GBDT}$ is often as good as the performance of ${\tt PRS}$. Is this because ${\tt GBDT}$ is the only *optimized* method in the comparison?



**Strengths And Weaknesses:**

${\bf Strengths}$

The idea of optimizing the feature-selection distribution is interesting.

Understanding whether an important sampling strategy may approximate well a full bi-level optimization problem is a relevant problem. In certain setups, the empirical results look promising.

${\bf Weaknesses}$

The main contribution of the paper is to propose an importance sampling technique to speed up the optimization of the ensemble. It is not completely clear, however, whether the proposed important sampling technique can be justified theoretically. In practice, the model needs to be retrained after the probability distribution is updated a certain number of times. The paper quality would increase greatly if the authors could explain how this follows from Equation 5. In particular, what is the relationship between Equation 5 and the ideal objective function
$$E_{p(z|\alpha)}[\min_w L(f(w_z, z), {\cal D}_{train})] $$
where $w_z$ is the parameter of a model with input $z$?

On the empirical side, the author should have validated their main claim numerically, e.g. by comparing the proposed approach with a more expansive brute-force strategy where the model is retrained for each sample of the features.

The appendix contains a table showing how many times the model is retrained in each experiment. This is probably the most relevant table of the paper and should appear in the main text, with a detailed discussion of the computational gains associated with it.

The advantages of the method over single models seem to be more evident when the size of the available data is small. The author may run an experiment to show this explicitly, e.g. by plotting the performance gains over the base model versus the number of available training examples.

---

> ### Author Response · Authors · 2022-12-08
> **Response to Reviewer sJJj (part 1/2)**
>
> We thank you for your comments and suggestions.
>
> **Main contribution of the paper.** We would like first to clarify what we think is the main contribution of the paper. While we indeed propose an importance sampling technique to speed up the optimization of the ensemble, this only constitutes a secondary contribution. The main contribution is actually the formulation of the training of an ensemble as an optimization problem that can be solved with gradient descent, even when the base models are not differentiable. To our knowledge, no other approach formulates this ensemble training problem like we do.
>
> **Main questions**
>
> 1. **Pre-training.** No, the parameters of the base models are not pre-trained using all the features as inputs. Each base model is trained using only the subset of features that were selected (like in the standard Random Subspace approach).
>
> 2. **Base model retraining.** Eq. (5) computes the gradient as an expectation over the distribution $p(\mathbf{z} \vert \boldsymbol\alpha)$. In particular, the base model $f_\mathbf{z}$ is trained using the feature subset $\mathbf{z}$, which is sampled from the distribution $p(\mathbf{z} \vert \boldsymbol\alpha)$. When $\boldsymbol\alpha$ gets updated, the distribution $p(\mathbf{z} \vert \boldsymbol\alpha)$ changes and new models $f_\mathbf{z}$ must therefore be trained by sampling $\mathbf{z}$ from this new distribution.
>
> 3. **Expectation of the loss vs. loss of the expectation.** Optimizing the expectation of the loss (Eq. 17) and optimizing the loss of the expectation (Eq. 3) are not equivalent, as they do not solve the same problem. As explained in Section 2.3, Eq. (17) minimizes an upper bound to the combinatorial feature selection problem, as we have:
>
> $\min_\mathbf{z} \frac{1}{N} \sum_{i=1}^N L(y_i,f_{\mathbf{z}}(\mathbf{x}_i))$
>
> $\leq E_{p(\mathbf{z} \vert \boldsymbol\alpha)} [\frac{1}{N}\sum_{i=1}^N$ $L(y_i, f_\mathbf{z}(\mathbf{x}_i))],$
>
> where the left-hand term is the feature selection problem, which is combinatorial over the discrete values of $\mathbf{z}$. In other words, the above inequality states that the minimum of the loss function $F(\mathbf{z}) = \frac{1}{N} \sum_{i=1}^N L(y_i,f_{\mathbf{z}}(\mathbf{x}_i))$ is always less or equal than any average over different $\mathbf{z}$. Thus, instead of solving the feature selection problem over the discrete $\mathbf{z}$, one can minimize an upper bound by minimizing the expectation of the loss over the continuous $\boldsymbol\alpha$. In this paper however, our goal is not to solve the left-hand term in the above inequality, but to train an ensemble. The optimization problem in Eq. (3) is the minimization of the loss of the ensemble, which is exactly what we want to minimize (and not an upper bound of it).
>
> 4. **Practical threshold on the effective number of models.** We will add a table showing, on the simulated datasets, the performance and training times for different thresholds on the effective number of models. In particular, this table shows that the prediction errors and feature importance AUPRs do not increase much when we vary the threshold from $0.9T$ to $T$ (which is equivalent to training new base models at each epoch), while the computational burden is much higher. As an example, on the Hypercube problem, the average training time of PRS-SVM increases from 13 minutes to 3.5 hours, while the mean accuracy does not change and the mean AUPR only slightly increases from 0.90 $\pm$ 0.14 to 0.92 $\pm$ 0.13.
>
> 5. **RaSE.** A "regularization of the parameters $\boldsymbol\alpha$" could be any type of constraint put on $\boldsymbol\alpha$, not only feature sparsity. In the paper, we show two examples of regularization: 1) on the MNIST dataset, where we add a regularization term that enforces spatial smoothness, and 2) on the gene network inference problem, where we add a regularization term that enforces modular networks. RaSE does not allow to include such type of regularization.
>
> The number of iterations of RaSE is much lower than the number of epochs in PRS because of the computational complexity of RaSE (for which 500,000 base models must be trained at each iteration, given the used hyper-parameters). The original paper of RaSE actually only shows results obtained with at most 3 iterations, and we also observed in our experiments that the best validation loss was usually obtained within the very first few iterations. We thus believe than 10 iterations of RaSE is sufficient.
>
> 6. **Performance gain over single model versus the number of training samples.** Thank you for your suggestion. We ran experiments on the simulated datasets, in which we varied the training sample size, but we did not observe that the gain of PRS over the single model was systematically higher for smaller training sets (although the gain was positive in all cases).

---

> > ### Author Response · Authors · 2022-12-08
> > **Response to Reviewer sJJj (part 2/2)**
> >
> > **Other questions**
> >
> > 1. **Unclear sentence in the abstract.** *it* refers to the degree of randomization. We will rephrase the sentence as follows: "The degree of randomization in our parametric Random Subspace is thus automatically tuned through the optimization of the feature selection probabilities. This is an advantage over the standard Random Subspace approach, where the degree of randomization is controlled by a hyper-parameter."
> >
> > 2. **Comparison with other feature selection strategies.** The feature ranking ability of our approach can indeed be compared with existing feature ranking methods. The paper includes a comparison to RF and GBDT, which are standard feature ranking approaches.
> >
> > 3. **Difference with the mixture of experts (MoE) approach.** Both PRS and MoE combine the predictions of individual models. One fundamental difference however is that in MoE, the final prediction is a weighted average of the base model predictions, where the weights, given by the gate, depend on the input vector $x$. Such gate allows to take into account the complementarity between the experts and acts as a bias reduction technique. On the other hand, in PRS, base models are independently drawn from a similar distribution (defined by $p(\mathbf{z} \vert \boldsymbol\alpha)$), which does not depend on the inputs. Like bagging and RS, the main effect of PRS is thus to reduce variance. The number of experts in MoE is also fixed, while in our case, the number of models can be increased to get a better approximation of the expectation over $p(\mathbf{z} \vert \boldsymbol\alpha)$ (and thus reduce variance).
> >
> > 4. **Base model regularization.** For each type of base model, the regularization is entirely controlled by the hyper-parameters of the base model, which are set to the default values used in the *scikit-learn* library. With these settings:
> >
> > - The decision tree is not regularized.
> >
> > - The regularization of the kNN is controlled by the number of nearest neighbors, which is set to 5.
> >
> > - The regularization of the SVM is controlled by the (inverse) regularization parameter $C$, which is set to 1.
> >
> > 5. **Optimization of the compared approaches.** Like for GBDT, we also optimized the main parameter of the standard Random Subspace (i.e. the number of sampled features for each base model) and the main parameter of the Random Forests (i.e. the number of sampled features at each tree node).

---

> > > ### Comment · Reviewer_sJJj · 2022-12-27
> > > **a further comments after reading the author rebuttal**
> > >
> > > After reading the authors' rebuttal, I have a few questions and comments.
> > >
> > > 1. In the introduction, the authors write "we show that this  optimization problem can be solved using gradient descent even when the base  models are not differentiable." Is this a relevant point? Do similar methods require the differentiability of the base model?
> > >
> > > 2. One of the main advantages of the methods seems to be the possibility of tuning "the degree of randomization" automatically. In practice, the number of sampled features depends on alpha, which is learned by solving the optimization problem. Choosing a good regularization term for alpha may be as painful as tuning the hyper-parameter in standard RS approaches.
> > >
> > > 3. It would be interesting to see what is the shape of the Bernoulli distribution before and after the algorithm has converged. Is this different from the distribution obtained from RaSE? If so, it would be useful to compare the feature-importance scores learned through the two models. Also, does RaSE converge after 10 iterations?
> > >
> > > 4. Outperforming a single model or non-optimized Random Subspace methods may be an expected result because the proposed method has an additional free parameter. Why is Figure 2 relevant?
> > >
> > > 5. The only fair comparison in Table 2 seems to be the one between Random Forest, Gradient Boosting Decision Trees and the proposed method with decision trees as the based model. Why does it make sense to compare ensemble methods with different base models?
> > >
> > > 6. I agree with other reviewers that the method should be compared with other feature selection approaches and ensembles based on more flexible models such as NN.
> > >
> > > 7. How is the decision to use non-regularized trees, K=5, and C=1 (for SVM) motivated? Does this choice affect the overall performance of the corresponding ensembles?
> > >
> > > 8. Concerning the need of retraining the model, how quickly does the performance drop if the model is retrained after 1, 2,.., T epochs?
> > >
> > > 9. I am still confused about why the optimization of the model for each sampled subset of features is not included in the objective function.
> > >
> > > 10. The authors claim they are interested in training the ensemble and not minimizing the feature-selection objective. But feature selection is mentioned as one of the motivations for this work. Why is standard feature selection expected to come as a byproduct of training the ensemble? Can the gap be quantified in some simple setup?

---

> > > > ### Author Response · Authors · 2023-01-05
> > > > **Response to Reviewer sJJj's further comments (part 1/3)**
> > > >
> > > > 1. **Differentiability of the base model**: Gradient descent optimization indeed requires the base models to be differentiable. This optimization approach requires the computation of the gradient of the objective function, which is a function of the base model. As discussed in Section 2.3, there are other approaches that use gradient descent to solve a similar problem as ours, but all these approaches use differentiable base models in the form of neural networks. In our case, it is the introduction of the expectation in the objective function in Eq. (3) that allows us to optimize this objective function with gradient descent even when the base models are not differentiable.
> > > >
> > > > 2. **Degree of randomization**: Tuning the regularization term in PRS is indeed equivalent to tuning the hyper-parameter of RS, in terms of computational burden. However, as discussed in Section 2.3, even without any regularization, PRS has a natural propensity for selecting few features. In fact, we applied PRS on the simulated datasets and on the real *tabular* benchmark without any regularization. Thus, introducing a regularization term is not always necessary in PRS, while the number of selected features is an important hyper-parameter in RS.
> > > >
> > > > 3. **Comparison to RaSE**: For both PRS and RaSE, all the features have the same importance score at the initialization. On the simulated datasets, the distribution of the importance scores after convergence is roughly the same for the two methods: the importance values are concentrated on relevant features, with high values for the latter and very low values for the irrelevant features. We will add plots of these distributions in subsequent versions of the paper.
> > > >
> > > > The number of iterations of RaSE is much lower than the number of epochs in PRS because of the computational complexity of RaSE (for which 500,000 base models must be trained at each iteration, given the used hyper-parameters). The original paper of RaSE actually only shows results obtained with at most 3 iterations, and we also observed in our experiments that the best validation loss was usually obtained within the very first few iterations. We thus believe than 10 iterations of RaSE is sufficient.
> > > >
> > > > 4. **Figure 2**: We think that it is important to show that the approach that we propose outperforms the relevant baselines, even if these results are expected. Note that the Random Subspace method is **not** non-optimized: the number of randomly selected features is tuned on a validation set (see Appendix D.3). There is thus no additional free parameter in PRS with respect to RS. Furthermore, Figure 2 shows that the improvement of performance over RS is larger in the case of kNN and SVM, compared to decision trees, which we believe is a very interesting result.
> > > >
> > > > 5. **Table 2**: The goal is to show that our approach, when used with different types of base models, outperforms the baseline approaches based on decision trees. The comparison with RS-kNN and RS-SVM is done in Figure 2. RF and GBDT are two strong baseline methods on tabular data and this has been partly explained by their robustness against irrelevant features (see Grinsztajn *et al.* Why do tree-based models still outperform deep learning on typical tabular data? *NeurIPS 2022 Datasets and Benchmarks* ). We believe that it is interesting to show that PRS ensembles constructed with other base models that are not inherently robust against irrelevant features can reach similar or better performance.

---

> > > > > ### Author Response · Authors · 2023-01-05
> > > > > **Response to Reviewer sJJj's further comments (part 2/3)**
> > > > >
> > > > > 6. **Comparison to other feature selection approaches**: As discussed in our response to reviewer g9XZ (point 2), we do not consider our approach as a feature selection method, but rather as an ensemble method that produces an inherently interpretable ensemble model, in the sense that we know which features are the most important for the ensemble as a byproduct of its training. This characteristic is shared with ensemble methods applied with trees since trees naturally embed some feature selection and allow to derive feature importance scores in a more or less straightforward way. One of the main advantages of PRS is to bring this embedded feature selection and the derivation of feature importance scores to any base learner and it also shares this characteristic with RaSE. This is the reason why we mainly compare PRS with tree-based ensemble methods (Random forest and Gradient Boosting) and with RaSE. We nevertheless compare the selected features with the ground truth on the artificial problems and, indirectly, experiments in Section 3.4 show that PRS performs well against the plethora of feature selection methods that have been applied on the DREAM4 gene network inference problems. A comparison against other feature selection methods on the real datasets, for which the ground truth is unknown, would not be trivial (which metrics should we used for the evaluation?) and not very informative in our opinion.
> > > > >
> > > > > **MLP as base model**: See our answer to reviewer g9XZ (point 4). We don't include the MLP (or other neural network architectures) in our experiments, as the main purpose of our approach is to train ensembles of models that are non-differentiable. Moreover, the straightforward application of our method with MLP would be very expensive in terms of computing times. With differentiable models, a more efficient strategy would probably be to jointly optimize the feature subsets and the model parameters (rather than having an alternate approach like ours), either by using stochastic gradient estimators like the score function or the Straight-through estimator, or through a a continuous relaxation of the discrete variables $\mathbf{z}$ (as discussed at the end of Section 2.3). Some of the works mentioned at the end of Section 2.3 explore related ideas. We prefer to leave the extension of our ensemble framework to MLP or other differentiable models as future work.
> > > > >
> > > > > 7. **Base model hyper-parameters**: The non-regularized tree, $K=5$ and $C=1$ are simply the default hyper-parameter values used in the *scikit-learn* library. We stick to these default values to simplify the experiments. The hyper-parameters certainly affect the overall performance of the ensembles but since these hyper-parameters are common to all compared methods, we think this should not affect the relative ranking of the compared methods.
> > > > >
> > > > > 8. **Base model retraining**: The performance drops quite slowly. In Table S10 (of the revised version of the paper), dividing by 1000 the total number of trained models gives the number of epochs (over 3000 epochs) where new base models are trained (since we use 10 batches and ensembles of size 100, at each of these epochs, we train 100*10 base models). There is a strong degradation of the performance for $T_{eff}=0.5T$ and $T_{eff}=0.3T$. For these thresholds, the number of epochs where new models are trained is around 30-60 (depending of the problem), i.e. one every 50-100 epochs on average.
> > > > >
> > > > > 9. **Base model optimization**: The optimization of the model is implicitly included in the objective function. In the paragraph before Equation (2), we have defined $f_\mathbf{z}$ as the base model learned by the learning algorithm using as inputs only the variables in the subset $\mathbf{z}$. We don't include the optimization of the parameters of each base model in the global minimization in (3) at the same level as $\boldsymbol\alpha$, as we consider ensembles of independently sampled models, as in RS or RF. Including the parameters of the base models in the minimization in (3) would imply to choose a priori the number of models (as we cannot jointly optimize the parameters of an infinity of models) and would make the approach closer to a mixture of experts than to an ensemble method.

---

> > > > > > ### Author Response · Authors · 2023-01-05
> > > > > > **Response to Reviewer sJJj's further comments (part 3/3)**
> > > > > >
> > > > > > 10. **Feature selection as a byproduct**: (See also our answer to point 6). We do not want to solve explicitly the feature selection problem but to build an accurate ensemble by improving over RS. Feature selection comes as a byproduct of training the ensemble because one expects that irrelevant features will lead to small selection probabilities ($\alpha$ close to 0) while the most important ones are expected to get high selection probabilities ($\alpha$ close to 1), since this is how the performance of the ensemble will be optimized. Our experiments in 3.1, 3.3, and 3.4 confirms that this is indeed the case.
> > > > > >
> > > > > > **Can the gap be quantified in some simple setup?** We are not sure what gap you are referring to. If you mean the gap between the optimal $\boldsymbol\alpha$ according to (3) and according to (17), it is not an easy question. While we believe that exploiting the ideas explored in the paper to solve (17) should be possible, this would require some further work to adapt the algorithm for estimating the gradient. What we can say at this stage is that the optimal $\boldsymbol\alpha$ according to (17) should be composed of only 0 and 1 (corresponding to the optimal feature subset), while the optimal $\boldsymbol\alpha$ according to (3) contains non extreme values to keep some randomization within the ensemble.

---

### Review · Reviewer_YTUw · 2022-12-07

**Summary Of Contributions:**

The authors propose a new supervised ensemble approach. The new method uses relaxed Bernoulli variables to sample feature subsets for each base model. The framework is related to a random subspace ensemble, with a key feature that enables the model to learn the importance of different features by updating the parameters of the relaxed Bernoulli variables. The Bernoulli relaxation makes the method differentiable and enables adding a regularization (for example, for feature selection).  The scheme is agnostic to the type of based model, which is not necessarily differentiable. The authors evaluate their approach on several synthetic and real datasets and show its merits. The model seems useful both for ensemble purposes but also for identifying informative features.

**Audience:**

Yes

**Broader Impact Concerns:**

No concern was raised.

**Claims And Evidence:**

Yes

**Requested Changes:**

Please address my weakness above.
Why is equation 3 split?
Can the straight-through estimator be used as an alternative to REINFORCE?
In table 6, the AUPR does improve but still seems very low ~0.25, is this considered significant for identifying relevant genes? Also, what are the dimensions of this problem?

Other papers that use relaxed Bernoulli variables that are worth mentioning:

Balın, M. F., et al. (2019, May). Concrete autoencoders: Differentiable feature selection and reconstruction. In International conference on machine learning (pp. 444-453). PMLR.
Yang, J., et al. (2022, June). Locally Sparse Neural Networks for Tabular Biomedical Data. In International Conference on Machine Learning (pp. 25123-25153). PMLR.

**Strengths And Weaknesses:**

Strengths:

The paper is well-written, and overall I enjoyed reading the paper. The method is simple, seems natural, and requires generalization for random subspace ensemble.
The problem tackled by the authors is important and interesting for the community.
The authors have also presented a way to reduce the variance of the gradient estimate (section 2.2.2); this might be useful in high-dimensional settings.
The proposed solution does not require the base model to be differentiable; I see this as a nice advantage of the method and makes it suitable with any classifier.
The authors use synthetic data to demonstrate that the model can both improve test accuracy and identify informative features.
An evaluation with real data and an application are presented.

Weaknesses
I don't see an evaluation of the advantage of tuning b (section 2.2.2).
I see a short discussion about computational complexity; I think it is worth demonstrating this empirically concerning some base mode so that the reader can understand the time constraints.
Some of the results seem inferior to GBDT, I think this is fine, but I expect some discussion about when this method could excel. I assume that on some datasets, it does better than others. Maybe it has to do with the number of samples/features. My guess is that if there are many nuisances features, this method would outperform GBDT.

---

> ### Author Response · Authors · 2022-12-20
> **Response to Reviewer YTUw**
>
> We thank you for your feedback. We are glad that your enjoyed reading the paper and appreciated our method.
>
> Before replying to your comments, we first would like to point out that our method does not use a *relaxed* Bernoulli distribution, but the standard *discrete* Bernoulli distribution (one for each feature), from which we can  sample subsets of features (with each feature being either present or absent from the subset). The parameters $\boldsymbol\alpha$ of these distributions are however continuous and hence can be optimized using gradient descent.
>
> **Requested changes**
>
> 1. **Value of the baseline** *b*: We tune the value of the baseline *b* to minimize the variance of the gradient estimator. The optimal value is given in Eq. (12). In the revised version of the paper, we added a table showing, on the simulated datasets, the performance of PRS when we remove the variance reduction technique by setting the baseline *b* to 0. Setting *b*=0 typically results in a worse performance (in particular for the regression problems) and larger feature subsets.
>
> 2. **Computing times**: We added a figure showing the computing times of PRS on the simulated problems, for different training set sizes and different numbers of features.
>
> 3. **Comparison to RF and GBDT**: Thank you for your suggestion. We ran an additional experiment, where we increased the number of irrelevant features in the simulated problems. For example, on the Checkerboard problem, we indeed observe that the performance of RF and GBDT rapidly decreases with the number of irrelevant features, while the performance of PRS-SVM remains more stable. This behaviour is however less obvious on the other problems.
>
> 4. **Eq. (3)**: We split Eq. (3) in order to define the notation $F(\boldsymbol\alpha)$, that we use multiple times throughout the paper.
>
> 5. **Straight-through estimator as an alternative to the score function estimator**: We cannot use the Straight-through estimator in our case because we are in a setting where the base model $f_\mathbf{z}$ is not differentiable.  In the case of a Bernoulli distribution, the Straight-through estimator estimates $\nabla_{\boldsymbol\alpha} \mathbf{z} \simeq 1$ and estimates the gradient as: $\frac{\partial}{\partial \alpha_j} \mathbb{E} [f_\mathbf{z}] \simeq \frac{1}{T} \sum_{t=1}^T f'(\mathbf{z}^{(t)}) \frac{\partial \mathbf{z}^{(t)}}{\partial \alpha_j} = \frac{1}{T} \sum_{t=1}^T f'(\mathbf{z}^{(t)})$. This implies that the function $f_\mathbf{z}$ must be differentiable.
>
>
> 6. **Low AUPRs on the DREAM4 networks**: Despite being quite low, the AUPRs are still significantly higher than an approach that randomly ranks the regulatory links (AUPR around 0.02). The AUPRs of PRS are also slighly higher than the AUPRs of GENIE3, which was the best performing method of the DREAM4 challenge. As described in Section 3.4, each dataset comprises 100 samples and 100 genes.
>
> 7. **Other papers that use relaxed Bernoulli variables**: We added these two references when discussing this type of approach at the end of Section 2.3.

---

### Review · Reviewer_g9XZ · 2022-12-11

**Summary Of Contributions:**

This work proposes a model-agnostic interpretable ensemble method for a supervised learning task.  To choose a set of features among entire features, this method attempts to learn a Bernoulli distribution for selecting features.  Moreover, the feature selection probabilities are used to measure feature importances.  After choosing random features, any model is trained to construct an ensemble.  Finally, the authors provide some numerical results for diverse benchmarks including real-world problems.

**Audience:**

Yes

**Broader Impact Concerns:**

I do not have any broader impact concerns.

**Claims And Evidence:**

No

**Requested Changes:**

I think that this paper proposes an interesting approach for learnable model-agnostic feature selection, in order to construct an ensemble.

Regardless of the contributions of this work, I think that Table 1 and the corresponding description do not present the characteristics of ensemble methods appropriately.  Random forest has three primary components: average ensemble, bagging, and random feature selection.  All three components are model-agnostic, if we do not assume a decision tree is a base learner.  Similar to gradient boosting, we can easily define an ensemble model with bagging and random feature selection, with any base learners.  In addition, the description of feature importance is also problematic.  As described in the paper, if we choose a base learner as a decision tree for gradient boosting and random subspace, we can obtain the feature importance.  Moreover, [1] (a.k.a. LIME) has proposed to compute feature importance for any classifiers.  I think that if LIME is used, we can compute feature importance for any models.  To sum up, I think they should be revised more rigorously.  As my suggestions, I think Table 1 should be described by comparing existing methods in terms of separate components such as random feature selection, bagging, gradient boosting.  Along with the ensemble methods, feature selection methods such as LIME should be discussed elaborately.

Also, I am curious if another output aggregation technique can be investigated.  For example, voting, e.g., hard and soft voting, can be used to determine outputs.  Or, is there any particular reason why averaging is only considered?

I would like to see the results by more diverse base learners such as multi-layer perceptrons, which show the effectiveness of the proposed method.  Also, LIME can be compared to the proposed method.

I am also wondering whether $\boldsymbol \alpha$ is well-distributed or concentrated in a small number of particular features.

[1] Ribeiro, Marco Tulio, Sameer Singh, and Carlos Guestrin. ""Why should I trust you?" Explaining the predictions of any classifier." Proceedings of the 22nd ACM SIGKDD international conference on knowledge discovery and data mining. 2016.

**Strengths And Weaknesses:**

### Strengths

* It suggests an interesting and novel method.
* A paper is generally well-written, but it has some issues on presentation.

### Weaknesses

* I think that the motivation of this work should be revised.
* Some baseline method, e.g., multi-layer perceptron, should be added.

---

> ### Author Response · Authors · 2022-12-20
> **Response to Reviewer g9XZ**
>
> We thank you for your feedback.
>
> 1. **Description of the ensemble methods**: While Random Forest indeed includes a random feature selection, the way this feature selection is operated differs from model-agnostic approaches like Random Subspace. In Random Subspace, feature selection occurs at the level of the base model, with the same feature subset selected for the whole base model, and only the features in that subset can be used to train the base model. On the other hand, the feature selection mechanism used in Random Forest is designed specifically for decision trees (and is hence not model-agnostic): feature selection occurs at the level of the tree node, where a feature subset is randomly sampled before selecting the best split. We rewrote the introduction (and removed Table 1) to make our motivation clearer.
>
> 2. **LIME**: Unlike LIME and SHAP, PRS is an ensemble method and not an explanation (or post hoc) method, i.e. it can not be applied to explain the predictions of a pre-trained black-box model. We believe PRS can in fact be considered as a method that produces an inherently interpretable ensemble model, in the sense that we know which features are the most important for the ensemble as a byproduct of its training. Note also that PRS produces global feature importances, while LIME and SHAP produce local feature importances. Given these differences, we believe that comparing LIME/SHAP with PRS empirically would be irrelevant. We nevertheless now highlight the difference between PRS and model-agnostic explanation methods in Section 2.3.
>
> 3. **Output aggregation**: During training, the output of the ensemble is computed as the average of the predictions of the different base models in order to make the optimization with gradient descent feasible. However, regarding the predictions on a test set, the output of the PRS ensemble is computed as the majority class for classification problems (and the average for regression problems).
>
> 4. **MLP as base model**: We don't include the MLP (or other neural network architectures) in our experiments, as the main purpose of our approach is to train ensembles of models that are non-differentiable. Moreover, the straightforward application of our method with MLP would be very expensive in terms of computing times. With differentiable models, a more efficient strategy would probably be to jointly optimize the feature subsets and the model parameters (rather than having an alternate approach like ours), either by using stochastic gradient estimators like the score function or the Straight-through estimator, or through a a continuous relaxation of the discrete variables $\mathbf{z}$. Some of the works mentioned towards the end of Section 2.3 explore related ideas. We prefer to leave the extension of our ensemble framework to MLP or other differentiable models as future work.
>
> 5. **Values of $\boldsymbol\alpha$**: Ideally, the values of the optimized $\boldsymbol\alpha$ should be concentrated on the relevant features, with high values for the latter and zero values for the irrelevant features. This can be for example observed on the MNIST dataset (Figure 3). Even without any regularization ($\lambda_1 = \lambda_2 = 0$), the importance of many pixels is zero and a few of them have large values. The small expected values of the feature subset sizes (computed as the sum $\sum_j \alpha_j$) that are observed on the simulated and real-world datasets also suggest that many $\alpha_j$ have very low values.

---

### Author Response · Authors · 2022-12-20
**Revision**

We thank all the reviewers for their helpful comments and suggestions. We submitted a revised version of our paper, in which the modifications are highlighted in blue.

---

### Decision · Action_Editors · 2023-01-03

**Recommendation:** Reject

**Comment:**

This paper presents a variation of random subspace to construct an ensemble for supervised learning. The main idea is to parameterize the random subspace so that the corresponding parameters are learned by optimizing the ensemble loss. A critical trick is to use the importance sampling to bypass the re-training of base models at each gradient update of the feature selection probabilities.
In practice, it suggests to re-train the base models whenever the effective number of used feature subsets drops below 0.9T where T is the number of base models in the ensemble. In addition, the paper also claims that the optimized feature selection probabilities can be interpreted as feature importance scores. The idea seems to be sound and experiments demonstrate that the proposed method works better than the random subspace method.

While the paper contains an interesting idea, reviewers raised their concerns, some of which remained after the author response.
1. The paper explains the idea, presenting the algorithm. Experiments are provided to support that the method performs better than the random subspace method. One big missing component is an insight or justification why the method works better than the random subspace method. One reviewer suggested to present error bound for quantifying the performance gap. Even without theoretical error bound, it would be nice to present convincing justification so that readers can see any benefits brought by the method.
2. The importance sampling trick to bypass the re-training of base models is interesting. But it is not completely clear in the current manuscript. In fact, one reviewer also pointed out this issue, but unfortunately the rebuttal did not clarify it.
3. Most of reviewers expressed their concerns in experiments, which should be improved.
4. Since the paper claims that the optimized feature selection probabilities can be interpreted as feature importance scores, the paper should describe more details in this line of research. For instance, how optimized feature selection probabilities are related to existing explainable methods. Comparisons with them are also nice.

I do not think the minor revision can handle these issues.
Therefore, the paper is not recommended for acceptance in its current form.
I hope authors found the review comments informative and can improve their paper by addressing these carefully in future submissions.

**Audience:**

The parametric random subspace approach might be interesting for some individuals.

**Claims And Evidence:**

Two claims are made in this paper: (1) parametric random subspace improves the final performance compared to the existing random subspace; (2) the optimized feature selection probabilities in the parametric random subspace can be interpreted as feature importance scores. These are not convincing yet since the paper does not provide sufficient evidence.

---

> ### Author Response · Authors · 2023-01-05
> **Response to the action editor**
>
> We thank you for your consideration of our paper. We are however very much surprised, and obviously very disappointed, by the decision and how it is motivated. We would like to react to your list of concerns:
>
> 1. No reviewer has actually raised this issue (there were no concerns raised about why PRS might be better than RS and no suggestion to provide error bounds) and we thus did not have the opportunity to answer to this comment. The justification that PRS might work better than RS is very clear to us: the search space of PRS contains the search space of RS. Given this added flexibility, it is expected to work better on the learning sample at least. The fact that this leads to a better generalisation is supported through very extensive experiments in the paper. Note that in his further comments, reviewer sJJj commented that outperforming Random Subspace was an expected result.
>
> 2. We answer the only question raised by the reviewers about importance sampling in our response to reviewer sJJj (point 4) and added further experiments in the paper to highlight the benefit of importance sampling (Table S10). In his further comment, reviewer sJJj did not raise any further concern about this point. So, again, this concern comes as a complete surprise and we did not get the opportunity to address it. Could you explain what is still "not completely clear"?
>
> 3. We carried out experiments on 4 artificial problems, 60 publicly available real datasets, and two additional illustrative applications (MNIST and gene network inference). In response to the reviewers, we added five new tables and figures in our revision. We have explained why we don’t think adding experiments with neural networks and comparing against other feature selection methods (in addition to RF, GBDT and RaSE) would be relevant and we did not get any feedback as to why our arguments are not valid. Again, we don’t know which experiments are still missing.
>
> 4. This was addressed by point 2 in our response to reviewer g9XZ and this is addressed in our response to point 6 of the further comments of reviewer sJJj. We believe we compare ourselves to the most relevant works to ours: RF, GBDT and RaSE. It would have been very helpful for us to be more explicit about which comparison is expected, with which explainable method. The only explicit comparison asked by the reviewers was against LIME and we still believe this comparison would not be relevant (PRS is not an explanation method).
>
> Overall, we believe that concerns 1 to 3 are unfair and concern 4 is too vague to be addressed properly.  Also, we received this final decision while preparing our response to the new comments and questions raised by reviewer sJJj on December 27 (it took us some time to draft our answer because of the holiday period). It is very disappointing that our response to these comments (we just published it) could not be taken into account in your decision. I hope that you can understand our frustration.